# Physics-Driven Spatiotemporal Modeling for AI-Generated Video Detection

**Shuhai Zhang**[1 4 *]   **Zihao Lian**[1 *]   **Jiahao Yang**[1]   **Daiyuan Li**[1]   **Guoxuan Pang**[2]
**Feng Liu**[5]   **Bo Han**[7]   **Shutao Li**[6†]   **Mingkui Tan**[1 3†]

[1]South China University of Technology, [2]University of Science and Technology of China
[3]Key Laboratory of Big Data and Intelligent Robot, Ministry of Education, [4]Pazhou Lab
[5]University of Melbourne, [6]Hunan University, [7]Hong Kong Baptist University

## Abstract

AI-generated videos have achieved near-perfect visual realism (*e.g.*, Sora), urgently necessitating reliable detection mechanisms. However, detecting such videos faces significant challenges in modeling high-dimensional spatiotemporal dynamics and identifying subtle anomalies that violate physical laws. In this paper, we propose a physics-driven AI-generated video detection paradigm based on probability flow conservation principles. Specifically, we propose a statistic called *Normalized Spatiotemporal Gradient* (NSG), which quantifies the ratio of spatial probability gradients to temporal density changes, explicitly capturing deviations from natural video dynamics. Leveraging pre-trained diffusion models, we develop an NSG estimator through spatial gradients approximation and motion-aware temporal modeling without complex motion decomposition while preserving physical constraints. Building on this, we propose an NSG-based video detection method (NSG-VD) that computes the *Maximum Mean Discrepancy* (MMD) between NSG features of the test and real videos as a detection metric. Last, we derive an upper bound of NSG feature distances between real and generated videos, proving that generated videos exhibit amplified discrepancies due to distributional shifts. Extensive experiments confirm that NSG-VD outperforms state-of-the-art baselines by 16.00% in Recall and 10.75% in F1-Score, validating the superior performance of NSG-VD. The source code is available at https://github.com/ZSHsh98/NSG-VD.

## 1   Introduction

The rapid advancement of generative models [1, 2, 3, 4, 5, 6], particularly diffusion-based frameworks (*e.g.*, Sora [3]), has achieved unprecedented capabilities in synthesizing photorealistic video content. While these breakthroughs enable transformative applications in content creation for creative industries [7, 8, 9, 10], they simultaneously pose critical societal risks through malicious manipulation (*e.g.*, deepfake disinformation [11, 12, 13, 14, 15, 16, 17], synthetic media fraud [7, 18]). As AI-generated videos become increasingly realistic in both spatial and temporal domains, developing effective video detection methods becomes critically urgent for preserving societal trust in digital media.

A fundamental challenge for AI-generated video detection lies in modeling the spatiotemporal dynamics of video evolution. Intuitively, natural videos inherently obey physical laws like motion coherence and texture continuity [19, 20], while AI-generated videos often exhibit subtle yet systematic inconsistencies in spatiotemporal coherence [21]. This observation raises a crucial question:

*How can we model intrinsic spatiotemporal dynamics of natural videos to expose synthetic anomalies?*

---

*Equal contribution. Email: shuhaizhangshz@gmail.com, lianzihaolzh@gmail.com
†Corresponding author. Email: mingkuitan@scut.edu.cn, shutao_li@hnu.edu.cn

39th Conference on Neural Information Processing Systems (NeurIPS 2025).

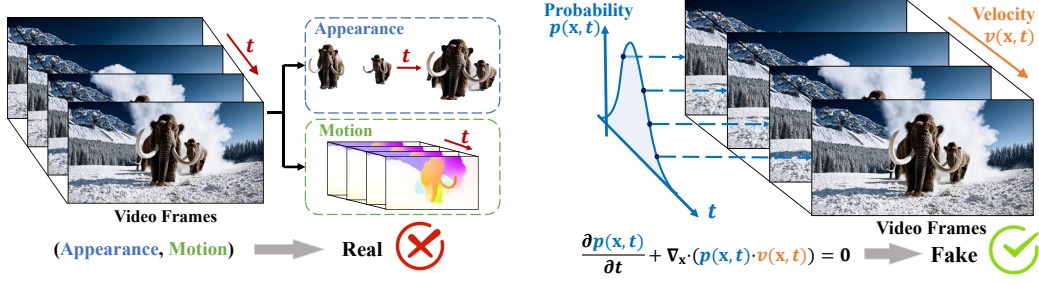

| (a) Traditional Spatiotemporal Modeling | (b) Physics-Driven Spatiotemporal Modeling (Ours) |

Figure 1: Comparisons of traditional and physics-driven paradigms for spatiotemporal modeling in AI-generated video detection. (a) Traditional methods [22, 23, 24] often rely on specific artifacts like appearance consistency and optical flow-based motion modeling, struggling with highly realistic content yet physically implausible (*e.g.*, Sora). (b) Our physics-driven approach explicitly models video dynamics via physics conservation laws, effectively identifying violations of physical laws.

Two critical difficulties arise: 1) Video content inherently contains complex *spatial domain correlations* (*e.g.*, texture structure) and *temporal domain dependencies* (*e.g.*, motion trajectories), requiring modeling frameworks that jointly capture both spatial structures and temporal dynamics characteristics. 2) AI-generated videos are rapidly approaching the perceptual quality of natural videos, with discrepancies that may become vanishingly *subtle* in both visual appearance and temporal evolution.

Existing AI-generated video detection methods primarily rely on local feature inconsistencies (*e.g.*, optical flow-based motion modeling [22], appearance consistency modeling [23]) or supervised learning with large-scale datasets [25, 24, 26, 27]. However, they often ignore physics-driven constraints governing spatiotemporal evolution inherent to natural videos. This limitation exhibits inherent vulnerabilities when confronting synthetic anomalies that violate physical laws, *e.g.*, non-physical motion patterns in Sora-generated videos [3] (Figure 1-a), leading to inferior performance.

In this paper, we propose a physics-driven paradigm based on probability flow conservation principles [28, 29]. By modeling video dynamics as fluid mechanics, we formulate video evolution through a probability flow velocity field governed by continuity equations (see Figure 1-b and Section 3.1). This reveals a key insight: *natural video dynamics preserve the product between the velocity field and the ratio of spatial probability gradients to temporal density changes*. Inspired by this, we introduce a **Normalized Spatiotemporal Gradient (NSG)** statistic, which quantifies the ratio of spatial probability gradients to temporal density changes. NSG captures fundamental discrepancies in how videos adhere to physical constraints while eliminating reliance on specific artifacts, enabling sensitive detection even when visual differences are imperceptible to humans or conventional models.

To enable practical estimation, we develop an NSG estimator leveraging pre-trained diffusion models' inherent gradient estimation ability [30, 31] in Section 3.2. By approximating spatial gradients with learned score functions (*i.e.*, the gradient of the log probability density) from the diffusion models and temporal derivatives through motion-aware temporal dynamics via a brightness constancy constraint [32], our method avoids explicit flow computation while preserving essential physical constraints. This estimator eliminates reliance on complex motion modeling by physics-inspired priors while maintaining sensitivity to subtle spatiotemporal inconsistencies inherent to synthetic content.

Building on this foundation, we propose an **NSG-based video detection method (NSG-VD)** in Section 3.3, which computes the *Maximum Mean Discrepancy* (MMD) [33, 34] between NSG features of real videos and the test video as the detection metric, as illustrated in Figure 2. We further theoretically derive an upper bound of the distance between NSG features of real and generated data in Section 3.4, showing this bound expands with increasing distribution shifts in generated videos. This implies that the MMD between NSGs of real videos tends to be smaller than that between real and generated videos, establishing the theoretical basis for the effectiveness of NSG-VD. Extensive experiments show that NSG-VD achieves $16.00\%$ higher Recall and $10.75\%$ higher F1-score than baselines, validating the superior performance of NSG-VD. Our contributions are summarized as:

- A physics-driven NSG statistic: We formulate the video evolution through a probability flow velocity field with a continuity equation and propose a novel statistic Normalized Spatiotemporal Gradient (NSG) that explicitly models spatiotemporal dynamics of videos. By quantifying the ratio

of spatial probability gradients to temporal density changes, NSG fundamentally captures violations of physical continuity in AI-generated videos without reliance on artifact-specific supervision.

- A diffusion-guided NSG estimation with physical priors: We develop an NSG estimator by spatial gradients approximation and motion-aware temporal dynamics modeling using pre-trained diffusion models. By avoiding explicit flow modeling and instead enforcing brightness constancy constraints, our method achieves effective NSG approximation without domain-specific motion modeling.

- An AI-generated video detection method with theoretical and empirical justifications: We propose an NSG-based video detection method (NSG-VD), which quantifies distributional shifts in NSG features using *Maximum Mean Discrepancy* (MMD). We derive an upper bound of NSG feature distances between real and generated videos, proving that generated videos exhibit amplified discrepancies under distribution shifts. Empirical results also show the superiority of our NSG-VD.

## 2 Related Work

**AI-Generated Video Detection.** Early generated video detection methods primarily focus on identifying synthetic facial videos. Yang et al. [35] and Amerini et al. [22] exploit auxiliary facial motion cues (landmark dynamics vs. optical flow) for deepfake detection. Gu et al. [36] separately model spatial and temporal inconsistencies, and introduce a vertical slicing feature fusion mechanism to establish a more comprehensive spatial-temporal representation. Wang et al. [23] propose an alternating-freezing strategy with spatiotemporal augmentation for facial consistency modeling. Xu et al. [24] transform consecutive frames into a predefined layout via masking/resizing to enable efficient spatiotemporal modeling. Peng et al. [37] integrate multi-feature fusion of facial perspectives, textures, and attributes. While most methods utilize facial priors, their reliance on domain-specific features limits their generalizability to more general AI-generated content detection.

With the rapid advancement in video generation, detecting general AI-generated content has become challenging. Bai et al. [38] fuse frame-level and optical flow predictions to detect spatial-temporal anomalies. To jointly capture spatiotemporal cues, Ma et al. [39] and Chen et al. [25] propose Transformer- and mamba-based frameworks to model spatiotemporal relationships in video frame features for detection. Song et al. [40] exploit the cross-modal perception and reasoning in vision-language large models to learn general forgery features. Despite this progress, these methods mainly focus on appearance inconsistencies, while overlooking the intrinsic spatiotemporal dynamics cues, thereby struggling to tackle visual cues from diverse video generative models.

**Diffusion Models.** Diffusion models [41, 30, 42, 31] have emerged as powerful probabilistic generative models, benefiting from their diffusion-denoising paradigm that perturb data into noise through Gaussian processes and reconstruct samples via iterative denoising. Intuitively, the high-quality and diverse generative capabilities of diffusion models come from their ability to capture and exploit the distributional characteristics of natural data, enabling effective discrimination between natural samples and outliers. Motivated by this, a growing body of research has leveraged diffusion models for the detection of adversarial [43, 44, 45] and generated samples [46, 47, 48], wherein the score model emerges as a powerful discriminative tool. Nevertheless, it remains challenging to simultaneously capture and integrate spatiotemporal features when relying solely on score models.

## 3 Modeling Spatiotemporal Dynamics for AI-Generated Video Detection

**AI-Generated Video Detection.** Let $\mathbb{P}$ be a Borel probability measure on a separable spatiotemporal metric space $\mathcal{X} \subset \mathbb{R}^{T \times d}$, where $T$ is the number of frames and $d$ is the spatial dimension. Given independent and identically distributed (i.i.d.) samples $S_{\mathbb{P}} = \{\mathbf{x}^{(i)}\}_{i=1}^{n}$ from the real video distribution $\mathbb{P}$, we aim to determine whether each sample $\mathbf{y}^{(j)}$ in $S_{\mathbb{Q}} = \{\tilde{\mathbf{y}}^{(j)}\}_{j=1}^{m}$ originates from $\mathbb{P}$.

**Challenges for Video Detection.** The complex spatiotemporal dynamics in high-dimensional video data often require modeling both spatial irregularities (*e.g.*, unnatural textures) and temporal inconsistencies (*e.g.*, implausible motions). Moreover, the diversity of generative paradigms (*e.g.*, diffusion models [1] and generative adversarial networks [49]) introduces heterogeneous distribution shifts that exhibit as subtle statistical inconsistencies rather than explicit artifacts. These challenges are further worsened by rapidly evolving generative techniques (*e.g.*, Sora [3]), which continuously produce novel spatiotemporal patterns surpassing existing detection mechanisms.

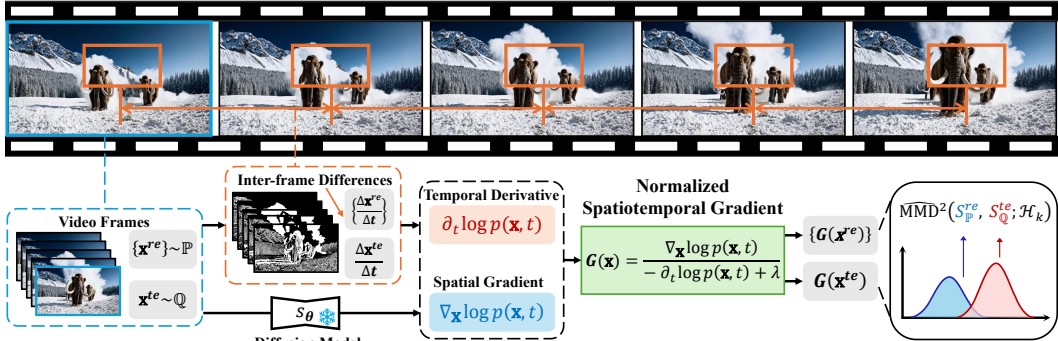

Figure 2: Overview of the proposed NSG-VD. Given a reference set of real videos $\{\mathbf{x}^{re}\}$ and a test video $\mathbf{x}^{te}$, we estimate their spatial gradients $\nabla_{\mathbf{x}} \log p(\mathbf{x}, t)$ and temporal derivatives $\partial_t \log p(\mathbf{x}, t)$ via a pre-trained diffusion model $s_\theta$, from which we derive their Normalized Spatiotemporal Gradients (NSGs) and calculate the MMD between NSG features of real and test videos as a detection metric.

**Method Overview.** To address these challenges, we propose a physics-driven method based on physical conservation principles to model *spatiotemporal dynamics* and introduce a novel statistic **Normalized Spatiotemporal Gradient (NSG)**, which quantifies the ratio of spatial probability gradients to temporal density changes, capturing subtle anomalies in videos (Section 3.1). Leveraging diffusion models, we develop an effective NSG estimator by spatial gradients approximation and motion-aware temporal dynamics modeling (Section 3.2). Building on this, we develop an **NSG-based video detection method (NSG-VD)**, which computes the *Maximum Mean Discrepancy* (MMD) between NSG features of the test video and real videos as a detection characteristic (Section 3.3), where its framework is shown in Figure 2. Last, we theoretically show that the MMD between NSGs of real videos tends to be smaller than that between real and generated videos (Section 3.4).

### 3.1 Modeling Spatiotemporal Dynamics via Normalized Spatiotemporal Gradient

Detecting AI-generated videos requires capturing both spatial irregularities and temporal inconsistencies in synthetic content. Inspired by conservation laws in physics (*e.g.*, mass or energy transport), we initially formulate the probability flow velocity field $\mathbf{v}(\mathbf{x}, t)$ to model the evolution of probability density $p(\mathbf{x}, t)$, which satisfies a continuity equation for global consistency across spatiotemporal domains. However, solving $\mathbf{v}$ faces challenges due to its underdetermined nature (Eqn. (4)). To address this, we propose **Normalized Spatiotemporal Gradient (NSG)** $\mathbf{g}(\mathbf{x}, t)$, a *dual* field statistic of $\mathbf{v}$ combining both spatial gradients and temporal dynamics of $p(\mathbf{x}, t)$, as defined in Eqn. (6).

**Probability Flow Velocity Field $\mathbf{v}(\mathbf{x}, t)$.** We begin to conceptualize the *probability flow* (also called *probability current*) as the movement of probability mass of $\mathbf{x}$ over time $t$ [50, 51]. To formalize this flow, we define the *probability flow density* $\mathbf{J}(\mathbf{x}, t)$, analogous to fluid mechanics [52]:

$$\mathbf{J}(\mathbf{x}, t) = p(\mathbf{x}, t) \cdot \mathbf{v}(\mathbf{x}, t), \tag{1}$$

where $p(\mathbf{x}, t)$ denotes the probability density and $\mathbf{v}(\mathbf{x}, t)$ represents the velocity field guiding the flow of probability mass. The conservation of probability mass [28, 29] implies the continuity equation:

$$\frac{\partial p(\mathbf{x}, t)}{\partial t} + \nabla_{\mathbf{x}} \cdot \mathbf{J}(\mathbf{x}, t) = 0, \tag{2}$$

where $\nabla_{\mathbf{x}} \cdot \mathbf{J} = \sum_i \frac{\partial \mathbf{J}_i}{\partial \mathbf{x}_i}$ denotes the divergence of the vector field $\mathbf{J}$ [51]. This is not a video-specific assumption but a universal mathematical formulation of probability mass conservation, which holds for any time-evolving probability density $p(\mathbf{x}, t)$ [53, 29]. Intuitively, this equation shows that the rate of change in probability density $\partial_t p$ at a point equals the difference between *inflow* (negative divergence) or *outflow* (positive divergence) of the probability flow $\mathbf{J}$. Substituting $\mathbf{J}(\mathbf{x}, t)$ into Eqn. (2), dividing by $p(\mathbf{x}, t)$, and applying the chain rule to $\log p(\mathbf{x}, t)$, yields:

$$\partial_t \log p(\mathbf{x}, t) + \nabla_{\mathbf{x}} \cdot \mathbf{v}(\mathbf{x}, t) + \mathbf{v}(\mathbf{x}, t) \cdot \nabla_{\mathbf{x}} \log p(\mathbf{x}, t) = 0. \tag{3}$$

This expression reveals how the velocity field $\mathbf{v}(\mathbf{x}, t)$ simultaneously encodes temporal evolution ($\partial_t \log p(\mathbf{x}, t)$) and spatial gradients ($\nabla_{\mathbf{x}} \log p(\mathbf{x}, t)$) of the probability distribution.

**Normalized Spatiotemporal Gradient $\mathbf{g}(\mathbf{x}, t)$ as Dual Field of $\mathbf{v}(\mathbf{x}, t)$.** To solve $\mathbf{v}(\mathbf{x}, t)$, we focus on the dominant components of Eqn. (3). Assuming that the divergence term $\nabla_{\mathbf{x}} \cdot \mathbf{v}$ is subdominant in smoothly varying distributions (*e.g.*, incompressible flow approximations [28, 54]), a condition commonly used in fluid dynamics [28] and quantum mechanics [55], Eqn. (3) simplifies to:

$$\mathbf{v}(\mathbf{x}, t) \cdot \nabla_{\mathbf{x}} \log p(\mathbf{x}, t) \approx -\partial_t \log p(\mathbf{x}, t). \tag{4}$$

Considering the non-uniqueness of solutions to $\mathbf{v}(\mathbf{x}, t)$ in Eqn. (4), we normalize both sides into

$$\mathbf{v}(\mathbf{x}, t) \cdot \frac{\nabla_{\mathbf{x}} \log p(\mathbf{x}, t)}{-\partial_t \log p(\mathbf{x}, t)} \approx 1. \tag{5}$$

**Definition 1.** *(Normalized Spatiotemporal Gradient (NSG).) The relation in Eqn. (5) reveals that natural video dynamics preserve the product between the velocity field and the ratio of spatial probability gradients to temporal density changes. We formalize this constrained ratio as the Normalized Spatiotemporal Gradient (NSG), defined as:*

$$\mathbf{g}(\mathbf{x}, t) = \frac{\nabla_{\mathbf{x}} \log p(\mathbf{x}, t)}{-\partial_t \log p(\mathbf{x}, t) + \lambda}. \tag{6}$$

Here, $\lambda > 0$ prevents numerical instability. Eqn. (5) and (6) imply that $\mathbf{g}(\mathbf{x}, t)$ acts as a *dual field* to $\mathbf{v}(\mathbf{x}, t)$, satisfying $\mathbf{v} \cdot \mathbf{g} \approx 1$. The formulation of $\mathbf{g}(\mathbf{x}, t)$ bypasses the ill-posed velocity $\mathbf{v}(\mathbf{x}, t)$ inversion problem while preserving the critical information about spatiotemporal gradient dynamics.

**Interpretation and Advantages**. The NSG statistic $\mathbf{g}(\mathbf{x}, t)$ quantifies the directional sensitivity of probability flow per unit temporal variation, driven by both spatial gradients ($\nabla_{\mathbf{x}} \log p(\mathbf{x}, t)$) and temporal derivatives ($\partial_t \log p(\mathbf{x}, t)$). This statistic captures both spatial irregularities (via $\nabla_{\mathbf{x}} \log p(\mathbf{x}, t)$) and temporal inconsistencies (via $\partial_t \log p(\mathbf{x}, t)$), enabling comprehensive analysis of video dynamics. Moreover, by modeling fundamental probability flow dynamics, NSG avoids dependencies on specific artifacts, making it suitable for detecting generated videos across diverse generation paradigms.

## 3.2 Estimating NSG with Diffusion Models

The NSG statistic $\mathbf{g}(\mathbf{x}, t)$ in Eqn. (6) requires estimating two key components: the spatial gradients $\nabla_{\mathbf{x}} \log p(\mathbf{x}, t)$ and the temporal derivatives $\partial_t \log p(\mathbf{x}, t)$. Using diffusion models' inherent gradient estimation ability [30, 31], we propose an effective estimator combining spatial gradients from pre-trained diffusion models with motion-aware temporal dynamics using Eqn. (8) and (9), yielding:

$$\mathbf{g}(\mathbf{x}, t) \approx \frac{\mathbf{s}_\theta(\mathbf{x}_t)}{\mathbf{s}_\theta(\mathbf{x}_t) \cdot \frac{\mathbf{x}_{t+\Delta t} - \mathbf{x}_t}{\Delta t} + \lambda}, \tag{7}$$

where $\mathbf{s}_\theta$ denotes the learned score function from diffusion models and $\mathbf{x}_t$ represents the $t$-th video frame. This estimator eliminates the need for explicit flow computation while preserving critical spatiotemporal dynamics through physics-inspired modeling. Below, we detail its derivation.

**Spatial Gradients Estimation.** Diffusion models [31, 56] explicitly learn a score network $\mathbf{s}_\theta$ through score matching [30] or denoising diffusion modeling [41] to approximate $\nabla_{\mathbf{x}} \log p(\mathbf{x}, t)$. For a given video $\mathbf{x}$ at $t$-th frame, the spatial gradient is estimated by:

$$\nabla_{\mathbf{x}} \log p(\mathbf{x}, t) \approx \mathbf{s}_\theta(\mathbf{x}_t), \tag{8}$$

where $\mathbf{x}_t$ is the $t$-th frame of the video $\mathbf{x}$. Here, we omit the diffusion timestep to make the notation clearer. This estimation allows direct computation of $\nabla_{\mathbf{x}} \log p(\mathbf{x}, t)$ in NSG via a single forward pass of the pre-trained diffusion model, eliminating the need for numerical differentiation.

**Temporal Derivatives Approximation.** To estimate $\partial_t \log p(\mathbf{x}, t)$, we exploit the temporal coherence of video sequences under the *brightness constancy assumption* [32], which posits that the probability density along motion trajectories remains constant. This leads to the following approximation:

**Proposition 1.** *Under the brightness constancy assumption $p(\mathbf{x} + \Delta \mathbf{x}, t + \Delta t) \approx p(\mathbf{x}, t)$ with small inter-frame motion ($\Delta t \to 0$) and inter-frame displacement ($\Delta \mathbf{x} \to 0$), we have*

$$\partial_t \log p(\mathbf{x}, t) \approx -\frac{\nabla_{\mathbf{x}} \log p(\mathbf{x}, t) \cdot \Delta \mathbf{x}}{\Delta t}. \tag{9}$$

## 3.3 Exploring NSG for Detecting AI-Generated Videos

To effectively distinguish AI-generated content from real videos, it is crucial to design metrics that capture subtle distributional discrepancies in high-dimensional spatiotemporal features. Recent studies show *Maximum Mean Discrepancy* (MMD) [33]—a non-parametric statistic for distribution alignment—has demonstrated remarkable capabilities in measuring distributional differences, *e.g.*, AI-text detection [48, 46] and adversarial samples detection [57, 58]. Building upon MMD's theoretical foundation and NSG's unique strength in modeling spatiotemporal dynamics, we propose an **NSG-based video detection method (NSG-VD)** that integrates MMD with the NSG feature representation.

**MMD Formulation with NSG Features.** We aggregate NSG features across $T$ frames in each video as $\mathbf{G}(\mathbf{x}) = \{\mathbf{g}(\mathbf{x}, t)\}_{t=1}^T$. Let $S_{\mathbb{P}}^{re} = \{\mathbf{x}^{(i)}\}_{i=1}^n$ denote a reference set of real videos and $S_{\mathbb{Q}}^{te} = \{\tilde{\mathbf{y}}\}$ represent a test video. The MMD [33] between $S_{\mathbb{P}}^{re}$ and $S_{\mathbb{Q}}^{te}$ in terms of NSG is computed as:

$$\widehat{\mathrm{MMD}}_b^2 \left[ S_{\mathbb{P}}^{re}, S_{\mathbb{Q}}^{te}; \mathcal{H}_k \right] = \frac{1}{n^2} \sum_{i,j=1}^n k\left( \mathbf{G}^{(i)}, \mathbf{G}^{(j)} \right) - \frac{2}{n} \sum_{i=1}^n k\left( \mathbf{G}^{(i)}, \mathbf{G}^{(\text{test})} \right) + k\left( \mathbf{G}^{(\text{test})}, \mathbf{G}^{(\text{test})} \right), \quad (10)$$

where $\mathbf{G}^{(i)} = \mathbf{G}(\mathbf{x}^{(i)})$ and $\mathbf{G}^{(\text{test})} = \mathbf{G}(\tilde{\mathbf{y}})$ are NSG features extracted from real and test videos. The kernel $k : \mathcal{G} \times \mathcal{G} \to \mathbb{R}$ maps NSG features to a reproducing kernel Hilbert space (RKHS) $\mathcal{H}_k$, such as the Gaussian kernel $k(\mathbf{a}, \mathbf{b}) = \exp\left( -\|\mathbf{a} - \mathbf{b}\|^2 / (2\sigma^2) \right)$. Note that while MMD is conventionally used for distribution-level comparisons, recent studies [48, 46, 58] validate its efficacy in single-sample detection by quantifying deviations from reference distributions. Crucially, while MMD provides a viable solution for distributional comparison, the core advantage of NSG-VD stems from the NSG itself modeling fundamental spatiotemporal dynamics (see details in Appendix E.3).

**Detection Protocol with MMD Metric.** Let $f(\tilde{\mathbf{y}}; S_{\mathbb{P}}, k_\omega, \tau) = \mathbb{I}\left( \widehat{\mathrm{MMD}}_b^2 > \tau \right)$, where $\mathbb{I}$ is the indicator function and $\tau$ is a threshold for the decision. Given a test video $\tilde{\mathbf{y}}$, we compute the MMD with NSG against a referenced real video set and give the decision:

$$f(\tilde{\mathbf{y}}) = \begin{cases} \text{Fake}, & \text{if } f(\tilde{\mathbf{y}}; S_{\mathbb{P}}, k_\omega, \tau) = 1, \\ \text{Real}, & \text{if } f(\tilde{\mathbf{y}}; S_{\mathbb{P}}, k_\omega, \tau) = 0. \end{cases} \quad (11)$$

**Optimization for NSG-VD.** To enhance discriminative power, we use a deep kernel [34] for MMD:

$$k_\omega(\mathbf{x}, \mathbf{y}) = [(1 - \epsilon)\kappa\left( \phi_{\mathbf{G}}(\mathbf{x}), \phi_{\mathbf{G}}(\mathbf{y}) \right) + \epsilon] \cdot \Phi\left( \mathbf{G}(\mathbf{x}), \mathbf{G}(\mathbf{y}) \right), \quad (12)$$

where $\phi_{\mathbf{G}}(\mathbf{x}) = \phi(\mathbf{G}(\mathbf{x}))$ is a deep neural network, $\kappa$ and $\Phi$ are Gaussian kernels with bandwidths $\sigma_\phi$ and $\sigma_\Phi$, and $\epsilon \in (0, 1)$. The kernel parameters $\omega = \{\epsilon, \phi, \sigma_\phi, \sigma_\Phi\}$ will be optimized by Eqn. (13) to maximize the detection ability. Considering the multiple-population scenarios across diverse video distributions [48], we adopt a *multi-population aware optimization* for the kernel training:

$$k_\omega^* = \arg\max_{k_\omega} \frac{\widehat{\mathrm{MPP}}_u(S_{\mathbb{P}}^{tr}, S_{\mathbb{Q}}^{tr}; k_\omega)}{\sqrt{\hat{\sigma}^2(S_{\mathbb{P}}^{tr}, S_{\mathbb{Q}}^{tr}; k_\omega) + \lambda}}, \quad \hat{\sigma}^2 = \frac{4}{N^3} \sum_{i=1}^N \left( \sum_{j=1}^N H_{ij}^* \right)^2 - \frac{4}{N^4} \left( \sum_{i=1}^N \sum_{j=1}^N H_{ij}^* \right)^2, \quad (13)$$

where $S_{\mathbb{P}}^{tr}$ and $S_{\mathbb{Q}}^{tr}$ denote the training real and generated videos, respectively, $\widehat{\mathrm{MPP}}_u(S_{\mathbb{P}}^{tr}, S_{\mathbb{Q}}^{tr}; k_\omega) = \frac{1}{N(N-1)} \sum_{i \neq j} H_{ij}^*$ and $H_{ij}^* = k_\omega(\mathbf{x}_i, \mathbf{x}_j) - k_\omega(\mathbf{x}_i, \mathbf{y}_j) - k_\omega(\mathbf{y}_i, \mathbf{x}_j)$.

## 3.4 Theoretical Guarantees for NSG-VD

The effectiveness of NSG-VD relies on ensuring the MMD between NSG features of real videos is smaller than that between real and generated videos. To formalize this, we analyze the MMD formulation in Eqn. (10), where the key discriminative information lies in the cross-term $k\left( \mathbf{G}^{(i)}, \mathbf{G}^{(\text{test})} \right)$ since the first and third terms remain invariant for fixed reference sets. Under the Gaussian kernel, this cross-term is dominated by the exponential squared distance between NSG features. Note that analyzing $\frac{\nabla_{\mathbf{x}} \log p(\mathbf{x}, t)}{-\partial_t \log p(\mathbf{x}, t) + \lambda}$ under practical distributions can be very difficult and infeasible, we adopt a common practice [59, 60] by assuming Gaussian-distributed data to derive theoretical insights. Below, we first characterize the NSG statistics for real and generated videos under Gaussian assumptions.

**Proposition 2.** *Let the real video distribution be* $p(\mathbf{x}, t) = \mathcal{N}(\mathbf{0}, \sigma(t)^2 \mathbf{I}_d)$ *and the generated video distribution be* $q(\mathbf{y}, t) = \mathcal{N}(\boldsymbol{\mu}, \sigma(t)^2 \mathbf{I}_d)$, *respectively, where* $\mathbf{I}_d \in \mathbb{R}^{d \times d}$ *is an identity matrix and* $\boldsymbol{\mu} \neq \mathbf{0} \in \mathbb{R}^d$ *is the distribution shift and* $\sigma(t) \neq 0$, *the NSG* $\mathbf{g}(\mathbf{x}, t)$ *and* $\mathbf{g}(\mathbf{y}, t)$ *satisfy:*

$$\mathbf{g}(\mathbf{x}, t) = -\frac{\mathbf{x}/\sigma(t)^2}{D_r(\mathbf{x})}, \quad -\frac{\mathbf{x}}{\sigma(t)^2} \sim \mathcal{N}\Big(\mathbf{0}, \sigma(t)^2 \mathbf{I}_d\Big), \ D_r(\mathbf{x}) \sim \lambda + \frac{d\dot{\sigma}(t)}{\sigma(t)} - \frac{\dot{\sigma}(t)}{\sigma(t)} \chi^2(d);$$

$$\mathbf{g}(\mathbf{y}, t) = -\frac{\mathbf{y}/\sigma(t)^2}{D_f(\mathbf{y})}, \quad -\frac{\mathbf{y}}{\sigma(t)^2} \sim \mathcal{N}\Big(-\frac{\boldsymbol{\mu}}{\sigma(t)}, \sigma(t)^2 \mathbf{I}_d\Big), \ D_f(\mathbf{y}) \sim \lambda + \frac{d\dot{\sigma}(t)}{\sigma(t)} - \frac{\dot{\sigma}(t)}{\sigma(t)} \chi^2(d, \varphi),$$

*where* $D_r(\mathbf{x}) = \lambda + \frac{d\dot{\sigma}(t)}{\sigma(t)} - \frac{\|\mathbf{x}\|^2 \dot{\sigma}(t)}{\sigma(t)^3}$, $D_f(\mathbf{y}) = \lambda + \frac{d\dot{\sigma}(t)}{\sigma(t)} - \frac{\|\mathbf{y}\|^2 \dot{\sigma}(t)}{\sigma(t)^3}$, $\dot{\sigma}(t) \triangleq \frac{d}{dt}\sigma(t)$, *and* $\varphi = \frac{\|\boldsymbol{\mu}\|^2}{\sigma(t)^2}$, $\chi^2(d)$ *is the central chi-squared distribution with d degrees of freedom and* $\chi^2(d, \varphi)$ *is the noncentral chi-squared distribution with noncentrality parameter* $\varphi$ *and d degrees of freedom* [61].

Proposition 2 reveals that the distribution shift $\boldsymbol{\mu}$ in generated videos introduces deviations in both the numerator and denominator of the NSG, *i.e.*, *noncentral* Gaussian and chi-squared distributions. To quantify this deviation, we derive an upper bound on the squared distance between NSGs:

**Theorem 1.** *Let the real video distribution be* $\mathbf{x} \sim \mathcal{N}(\mathbf{0}, \sigma(t)^2 \mathbf{I}_d)$ *and the generated video distribution be* $\mathbf{y} \sim \mathcal{N}(\boldsymbol{\mu}, \sigma(t)^2 \mathbf{I}_d)$, *respectively, where* $\mathbf{I}_d \in \mathbb{R}^{d \times d}$ *is an identity matrix and* $\boldsymbol{\mu} \neq \mathbf{0} \in \mathbb{R}^d$ *is the distribution shift. Given* $\mathbf{G}(\mathbf{x}) = \{\mathbf{g}(\mathbf{x}, t)\}_{t=1}^T$, *denote* $\varphi = \|\boldsymbol{\mu}\|^2 / \sigma(t)^2$ *and assume* $|-\partial_t \log p(\mathbf{x}, t) + \lambda| \geq C > 0$ *and* $|-\partial_t \log p(\mathbf{y}, t) + \lambda| \geq C > 0$, *with probability at least* $1 - \delta$, *we have*

$$\|\mathbf{G}(\mathbf{x}) - \mathbf{G}(\mathbf{y})\|^2 \leq \mathcal{O}\left(\frac{T}{C^4 \sigma(t)^2}\left[\varphi d + d^2 + \varphi + \log \frac{T}{\delta} \cdot (\varphi + d) + \log^2 \frac{T}{\delta}\right]\right).$$

Theorem 1 reveals that the bound of the squared distance between NSG features of real and fake data will be smaller if the distribution shift term $\varphi = \|\boldsymbol{\mu}\|^2 / \sigma(t)^2$ is closer to zero for a given $\delta$. This formalizes the intuition that small distribution shifts produce small geometric distortions in NSG space, while significant deviations in synthetic content lead to large separations from real data. Under the Gaussian kernel, this implies that the real data have a larger $k(\mathbf{G}(\mathbf{x}), \mathbf{G}(\mathbf{y}))$ than the fake data since the distribution shift term $\varphi = 0$ for real data. Therefore, when substituted into Eqn. (10), the MMD between NSG features of real videos is smaller than that between real and generated videos.

## 4 Experiments

**Datasets.** We evaluate our methods on the GenVideo benchmark [25], a large-scale dataset for AI-generated video detection that includes diverse real-world videos and synthetic content from multiple generative models. We use Kinetics-400 [62] as the real video source, SEINE [63] or Pika [64] as the AI-generated videos for training. The test set comprises MSR-VTT [65] and 10 diverse AI-generated datasets from different generation paradigms. More details are in Appendix C.1.

**Evaluation Metrics.** We evaluate the performance of video detection on Recall, Accuracy, F1-score [66] and AUROC [67] metrics. More details are provided in Appendix C.2. We use **bold** numbers to indicate the best results and underlined numbers to denote the second-best results in tables.

**Baselines**. We compare our NSG-VD with following baselines: TALL [24], NPR [27], STIL [36], and Demamba [25]. These baselines are implemented based on the codebase provided by Demamba [25].

### 4.1 Comparisons on Standard Evaluation

We start by comparing our NSG-VD with baselines using $10,000$ real videos from Kinetics-400 and $10,000$ generated videos from Pika (Table 1) and SENIE (Table 2) for training, respectively.

**Results on Trained with Kinetics-400 and Pika.** From Table 1, existing methods exhibit critical limitations. For instance, Demamba struggles with generative paradigms like HotShot ($40.60\%$ Recall) and Sora ($48.21\%$ Recall), while NPR shows unstable performance with Accuracy ranging from $57.20\%$ to $98.20\%$. TALL fails on synthetic outliers (*e.g.*, $25.00\%$ Recall on Sora) and STIL collapses completely on critical cases (*e.g.*, $1.40\%$ Recall on HotShot and $1.79\%$ Recall on Sora), revealing limitations of their inherent dependencies on generator-specific artifacts.

Table 1: Comparisons with baselines on *a standard evaluation* (%), where we train all models with $10,000$ real and generated videos from Kinetics-400 and Pika, respectively.

| Method | Metric | Model Scope | Morph Studio | Moon Valley | HotShot | Show1 | Gen2 | Crafter | Lavie | Sora | Wild Scrape | Avg. |
|---|---|---|---|---|---|---|---|---|---|---|---|---|
| DeMamba | Recall | 87.00 | 93.60 | 98.80 | 40.60 | 48.40 | 98.00 | 88.40 | 59.00 | 48.21 | 58.20 | 72.02 |
| | Accuracy | 91.70 | 95.00 | 97.60 | 68.50 | 72.40 | 97.20 | 92.40 | 77.70 | 72.32 | 77.30 | 84.21 |
| | F1 | 91.29 | 94.93 | 97.63 | 56.31 | 63.68 | 97.22 | 92.08 | 72.57 | 63.53 | 71.94 | 80.12 |
| | AUROC | 98.04 | 98.82 | 99.68 | 87.84 | 90.12 | 99.46 | 97.81 | 91.32 | 88.36 | 87.38 | 93.88 |
| NPR | Recall | 61.20 | 80.00 | 98.00 | 16.00 | 33.00 | 91.20 | 80.60 | 34.60 | 35.71 | 43.20 | 57.35 |
| | Accuracy | 79.80 | 89.20 | 98.20 | 57.20 | 65.70 | 94.80 | 89.50 | 66.50 | 67.86 | 70.80 | 77.96 |
| | F1 | 75.18 | 88.11 | 98.20 | 27.21 | 49.03 | 94.61 | 88.47 | 50.81 | 52.63 | 59.67 | 68.39 |
| | AUROC | 93.05 | 97.18 | 99.66 | 82.97 | 90.50 | 99.13 | 97.87 | 87.54 | 90.47 | 91.84 | 93.02 |
| TALL | Recall | 51.20 | 65.20 | 93.40 | 32.00 | 61.60 | 94.80 | 81.80 | 49.20 | 25.00 | 53.60 | 60.78 |
| | Accuracy | 75.10 | 82.10 | 96.20 | 65.50 | 80.30 | 96.90 | 90.40 | 74.10 | 61.61 | 76.30 | 79.85 |
| | F1 | 67.28 | 78.46 | 96.09 | 48.12 | 75.77 | 96.83 | 89.50 | 65.51 | 39.44 | 69.34 | 72.63 |
| | AUROC | 95.82 | 97.14 | 99.73 | 92.55 | 97.36 | 99.79 | 99.09 | 94.84 | 86.67 | 93.75 | 95.67 |
| STIL | Recall | 73.80 | 70.80 | 43.40 | 1.40 | 2.00 | 45.00 | 13.20 | 7.20 | 1.79 | 11.60 | 27.02 |
| | Accuracy | 86.90 | 85.40 | 71.70 | 50.70 | 51.00 | 72.50 | 56.60 | 53.60 | 50.89 | 55.80 | 63.51 |
| | F1 | 84.93 | 82.90 | 60.53 | 2.76 | 3.92 | 62.07 | 23.32 | 13.43 | 3.51 | 20.79 | 35.82 |
| | AUROC | 96.43 | 97.77 | 99.34 | 86.66 | 90.56 | 98.88 | 97.04 | 88.16 | 92.57 | 87.52 | 93.49 |
| NSG-VD (Ours) | Recall | 68.33 | 98.33 | 100.00 | 92.50 | 87.50 | 80.00 | 98.33 | 94.17 | 78.57 | 82.50 | **88.02** |
| | Accuracy | 81.67 | 98.33 | 96.67 | 91.67 | 90.83 | 88.33 | 95.83 | 94.17 | 88.39 | 88.75 | **91.46** |
| | F1 | 78.85 | 98.33 | 96.77 | 91.74 | 90.52 | 87.27 | 95.93 | 94.17 | 87.13 | 88.00 | **90.87** |
| | AUROC | 92.26 | 98.66 | 98.15 | 94.45 | 96.38 | 94.83 | 98.16 | 97.41 | 96.40 | 94.73 | **96.14** |

Table 2: Comparisons with baselines on *a standard evaluation* (%), where we train all models with $10,000$ real and generated videos from Kinetics-400 and SEINE, respectively.

| Method | Metric | Model Scope | Morph Studio | Moon Valley | HotShot | Show1 | Gen2 | Crafter | Lavie | Sora | Wild Scrape | Avg. |
|---|---|---|---|---|---|---|---|---|---|---|---|---|
| DeMamba | Recall | 47.40 | 87.80 | 88.20 | 77.40 | 75.00 | 85.60 | 91.60 | 68.60 | 42.86 | 48.00 | 71.25 |
| | Accuracy | 72.80 | 93.00 | 93.20 | 87.80 | 86.60 | 91.90 | 94.90 | 83.40 | 68.75 | 73.10 | 84.54 |
| | F1 | 63.54 | 92.62 | 92.84 | 86.38 | 84.84 | 91.36 | 94.73 | 80.52 | 57.83 | 64.09 | 80.87 |
| | AUROC | 88.29 | 98.39 | 98.76 | 97.84 | 96.89 | 98.76 | 99.35 | 96.87 | 80.93 | 88.11 | 94.42 |
| NPR | Recall | 46.40 | 76.40 | 69.80 | 63.80 | 56.00 | 75.00 | 83.80 | 58.80 | 35.71 | 27.40 | 59.31 |
| | Accuracy | 71.40 | 86.40 | 83.10 | 80.10 | 76.20 | 85.70 | 90.10 | 77.60 | 66.96 | 61.90 | 77.95 |
| | F1 | 61.87 | 84.89 | 80.51 | 76.22 | 70.18 | 83.99 | 89.43 | 72.41 | 51.95 | 41.83 | 71.33 |
| | AUROC | 85.73 | 96.01 | 93.79 | 91.44 | 89.96 | 95.13 | 96.87 | 89.46 | 84.15 | 76.66 | 89.92 |
| TALL | Recall | 58.60 | 75.00 | 79.40 | 60.20 | 62.00 | 77.80 | 88.20 | 43.80 | 33.93 | 35.80 | 61.47 |
| | Accuracy | 78.80 | 87.00 | 89.20 | 79.60 | 80.50 | 88.40 | 93.60 | 71.40 | 66.07 | 67.40 | 80.20 |
| | F1 | 73.43 | 85.23 | 88.03 | 74.69 | 76.07 | 87.02 | 93.23 | 60.50 | 50.00 | 52.34 | 74.05 |
| | AUROC | 97.10 | 98.12 | 98.63 | 96.37 | 96.45 | 97.76 | 99.38 | 94.80 | 83.35 | 89.45 | 95.14 |
| STIL | Recall | 28.60 | 57.40 | 78.40 | 46.80 | 18.80 | 66.40 | 69.00 | 24.80 | 14.29 | 19.00 | 42.35 |
| | Accuracy | 64.20 | 78.60 | 89.10 | 73.30 | 59.30 | 83.10 | 84.40 | 62.30 | 57.14 | 59.40 | 71.08 |
| | F1 | 44.41 | 72.84 | 87.79 | 63.67 | 31.60 | 79.71 | 81.56 | 39.68 | 25.00 | 31.88 | 55.81 |
| | AUROC | 95.53 | 97.91 | 99.40 | 96.49 | 92.79 | 98.06 | 98.86 | 91.00 | 92.79 | 86.58 | 94.94 |
| NSG-VD (Ours) | Recall | 91.67 | 100.00 | 100.00 | 100.00 | 100.00 | 98.33 | 100.00 | 97.50 | 94.64 | 89.17 | **97.13** |
| | Accuracy | 82.50 | 88.33 | 89.58 | 84.58 | 86.25 | 87.08 | 86.67 | 87.92 | 89.29 | 78.33 | **86.05** |
| | F1 | 83.97 | 89.55 | 90.57 | 86.64 | 87.91 | 88.39 | 88.24 | 88.97 | 89.83 | 80.45 | **87.45** |
| | AUROC | 90.67 | 97.62 | 98.38 | 95.88 | 96.69 | 97.87 | 97.64 | 95.09 | 96.14 | 88.65 | **95.46** |

In contrast, our NSG-VD achieves state-of-the-art performance across all metrics, significantly outperforming baselines despite not being pre-trained on large-scale videos. Remarkably, NSG-VD demonstrates exceptional reliability on challenging closed-source generators like Sora ($78.57\%$ Recall vs. $48.21\%$ for Demamba) and emerging paradigms like HotShot ($92.50\%$ Recall vs. $40.60\%$ for Demamba), and maintains reliability across other diverse domains (*e.g.*, MorphStudio, MoonValley). Notably, our NSG-VD achieves $16.00\% \uparrow$ average Recall and $10.75\% \uparrow$ F1-score over Demamba, and $55.05\% \uparrow$ F1-score over STIL. These results confirm its generalization across both open-source and closed-source generated models, highlighting the advantages of physics-driven modeling.

**Results on Trained with Kinetics-400 and SENIE.** As shown in Table 2, our NSG-VD achieves superior detection performance across all metrics compared to baselines. Notably, it attains near-perfect Recall ($\geq 98.33\%$) on models like MoonValley, HotShot and Show1, while maintaining balanced performance across diverse domains (*e.g.*, ModelScope, WildScrape). These results are consistent with the results on Pika in Table 1, further demonstrating the effectiveness of our proposed method. In contrast, existing baselines exhibit pronounced limitations under this setting. Demamba's performance is more constrained ($\leq 85.60\%$ Recall on most models), and NPR's F1-score varies widely ($41.83\% \sim 89.43\%$). TALL shows instability on models like Sora ($33.93\%$ Recall), while

Table 3: Comparisons with baselines under *data-imbalanced scenarios* (%), where we train all models with $10,000$ real and $1,000$ generated videos from Kinetics-400 and SEINE, respectively.

| Method | Metric | Model Scope | Morph Studio | Moon Valley | HotShot | Show1 | Gen2 | Crafter | Lavie | Sora | Wild Scrape | Avg. |
|---|---|---|---|---|---|---|---|---|---|---|---|---|
| DeMamba | Recall | 56.80 | 80.40 | 82.60 | 65.60 | 63.80 | 78.20 | 83.00 | 53.40 | 33.93 | 43.20 | 64.09 |
| | Accuracy | 78.10 | 89.90 | 91.00 | 82.50 | 81.60 | 88.80 | 91.20 | 76.40 | 65.18 | 71.30 | 81.60 |
| | F1 | 72.17 | 88.84 | 90.17 | 78.94 | 77.62 | 87.47 | 90.41 | 69.35 | 49.35 | 60.08 | 76.44 |
| | AUROC | 93.01 | 98.17 | 98.90 | 96.42 | 95.36 | 98.38 | 98.74 | 95.50 | 86.51 | 87.49 | 94.85 |
| NPR | Recall | 25.40 | 52.20 | 42.40 | 26.00 | 21.40 | 48.20 | 66.60 | 22.00 | 10.71 | 12.20 | 32.71 |
| | Accuracy | 62.40 | 75.80 | 70.90 | 62.70 | 60.40 | 73.80 | 83.00 | 60.70 | 55.36 | 55.80 | 66.09 |
| | F1 | 40.32 | 68.32 | 59.30 | 41.07 | 35.08 | 64.78 | 79.67 | 35.89 | 19.35 | 21.63 | 46.54 |
| | AUROC | 83.64 | 94.85 | 92.44 | 86.68 | 83.77 | 94.33 | 95.77 | 84.34 | 84.60 | 70.52 | 87.10 |
| TALL | Recall | 28.20 | 45.20 | 41.20 | 26.20 | 33.80 | 60.20 | 60.20 | 22.60 | 25.00 | 18.20 | 36.08 |
| | Accuracy | 64.00 | 72.50 | 70.50 | 63.00 | 66.80 | 80.00 | 80.00 | 61.20 | 62.50 | 59.00 | 67.95 |
| | F1 | 43.93 | 62.17 | 58.27 | 41.46 | 50.45 | 75.06 | 75.06 | 36.81 | 40.00 | 30.74 | 51.40 |
| | AUROC | 93.34 | 94.56 | 94.25 | 91.64 | 91.63 | 94.99 | 97.60 | 91.46 | 84.92 | 85.20 | 91.96 |
| STIL | Recall | 25.80 | 64.80 | 68.40 | 46.20 | 29.20 | 67.20 | 70.00 | 44.40 | 26.79 | 25.00 | 46.78 |
| | Accuracy | 62.70 | 82.20 | 84.00 | 72.90 | 64.40 | 83.40 | 84.80 | 72.00 | 63.39 | 62.30 | 73.21 |
| | F1 | 40.89 | 78.45 | 81.04 | 63.03 | 45.06 | 80.19 | 82.16 | 61.33 | 42.25 | 39.87 | 61.43 |
| | AUROC | 85.14 | 95.74 | 96.87 | 89.46 | 83.22 | 96.09 | 96.23 | 90.36 | 89.89 | 78.99 | 90.24 |
| NSG-VD (Ours) | Recall | 85.83 | 99.17 | 100.00 | 99.17 | 97.50 | 95.83 | 99.17 | 91.67 | 82.14 | 81.67 | **93.21** |
| | Accuracy | 84.58 | 92.50 | 93.75 | 89.58 | 89.58 | 90.83 | 92.50 | 90.00 | 86.61 | 81.67 | **89.16** |
| | F1 | 84.77 | 92.97 | 94.12 | 90.49 | 90.35 | 91.27 | 92.97 | 90.16 | 85.98 | 81.67 | **89.48** |
| | AUROC | 90.76 | 98.18 | 98.18 | 95.03 | 95.48 | 96.97 | 96.53 | 95.11 | 95.73 | 87.13 | **94.91** |

STIL fails entirely on critical cases (*e.g.*, $19.00\%$ Recall on WildScrape). These failures highlight the fragility of artifact-based approaches in capturing subtle spatiotemporal inconsistencies.

Quantitatively, NSG-VD surpasses Demamba by $25.88\% \uparrow$ in average Recall ($97.13\%$ vs. $71.25\%$) and NPR by $16.12\% \uparrow$ in average F1-score ($87.45\%$ vs. $71.33\%$). On closed-source models like Sora, it achieves $94.64\%$ Recall—nearly twice Demamba's ($42.86\%$) and sextuple STIL's ($14.29\%$). This improvement highlights NSG-VD's sensitivity to synthetic anomalies, especially in near-photorealistic videos (*e.g.*, Sora), where subtle spatiotemporal inconsistencies are amplified by the NSG but not effectively captured by baselines, indicating reliable detection across diverse generation paradigms.

## 4.2 Comparisons on Challenging Data-Imbalanced Scenarios

In real-world scenarios, natural videos are often abundant and accessible, while collecting sufficient AI-generated videos remains challenging due to rapidly evolving generation techniques. To thoroughly assess reliability under these conditions, we train all models using $10,000$ Kinetics-400 real videos and only $1,000$ SENIE-generated videos. As shown in Table 3, all baselines exhibit significant limitations. Demamba fails catastrophically on challenging generators like Sora ($33.93\%$ Recall) and WildScrape ($43.20\%$ Recall), while NPR exhibits fluctuations in Accuracy ($55.36\% \sim 83.00\%$). TALL fails completely on emerging paradigms like WildScrape ($18.20\%$ Recall) and Lavie ($22.60\%$ Recall), and STIL shows highly variable performance, *e.g.*, $25.00\% \sim 70.00\%$ Recall. Such instability indicates over-reliance on synthetic data volume or sensitivity to superficial artifacts.

In contrast, NSG-VD achieves strong generalization across 10 diverse generations. Notably, NSG-VD attains superior performance on critical test cases: $82.14\%$ Recall on Sora (vs. $10.71\% \sim 33.93\%$ for baselines) and $81.67\%$ Recall on WildScrape (vs. $12.20\% \sim 43.20\%$). Critically, NSG-VD achieves $29.12\% \uparrow$ higher average Recall than Demamba and $38.08\% \uparrow$ higher F1-score than TALL. These results confirm NSG-VD's reliable generalization from limited synthetic data without compromising discriminative power, demonstrating that adherence to universal physical principles outperforms domain-specific feature reliance even when synthetic training data is severely constrained.

## 4.3 Impact of Spatial Gradients and Temporal Derivatives for NSG-VD

To investigate the impact of the spatial gradients $\nabla_{\mathbf{x}} \log p(\mathbf{x}, t)$ and temporal derivatives $\partial_t \log p(\mathbf{x}, t)$ for our NSG-VD, we evaluate these components as independent detection statistics for AI-generated video detection. To this end, we train these separate models with $10,000$ real videos from Kinetics-400 and generated videos from Pika. From Table 4, the spatial

Table 4: Impact of spatial gradients and temporal derivatives on average metrics (%).

| Method | Recall | Accuracy | F1 | AUROC |
|---|---|---|---|---|
| Spatial Gradients | 87.99 | 82.84 | 83.40 | 91.85 |
| Temporal Derivatives | 60.35 | 71.09 | 66.97 | 78.95 |
| NSG-VD (Ours) | **88.02** | **91.46** | **90.87** | **96.14** |

gradient achieves moderate performance (*e.g.*, $87.99\%$ Recall, $83.40\%$ F1-score), suggesting its ability to capture spatial anomalies, which may arise from its sensitivity to localized variations in texture or geometry. The temporal derivative, however, shows limited detection power (*e.g.*, $60.35\%$ Recall, $66.97\%$ F1-score), likely due to its sensitivity to transient noise in dynamic modeling. In contrast, our NSG-VD integrating both components achieves significantly enhanced performance (*e.g.*, $88.02\%$ Recall, $90.87\%$ F1-score). This demonstrates that the interplay between spatial gradients and temporal derivatives formalized via physical conservation principles is critical for video detection.

### 4.4 Impact of Decision Threshold for NSG-VD

We evaluate the decision threshold $\tau$ in Eqn. (11) for NSG-VD by testing $\tau \in [0.4, 1.3]$ under the same settings as Table 1. As shown in Figure 3, our NSG-VD maintains remarkably stable performance across a wide range of $\tau$ values without requiring fine-grained tuning. Specifically, NSG-VD consistently shows high detection performance as $\tau \in [0.7, 1.1]$ for average Recall, Accuracy and F1-Score across diverse generators. These results indicate that NSG features create a clear separation between real and fake distributions. We set $\tau = 1.0$ as the default throughout all settings.

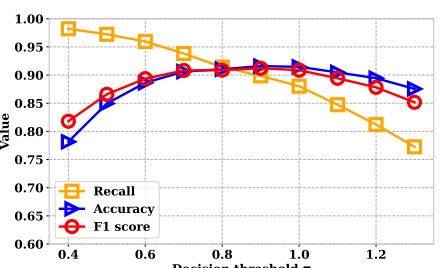

Figure 3: Impact of decision threshold.

## 5 Conclusion

In this paper, we propose a physics-driven AI-generated video detection paradigm by modeling spatiotemporal dynamics through the Normalized Spatiotemporal Gradient (NSG), a novel statistic based on probability flow conservation principles. Leveraging pre-trained diffusion models, we propose an NSG-based video detection method (NSG-VD). Theoretical analyses and extensive experiments validate the superiority of our NSG-VD in detecting advanced generated videos.

## Acknowledgements

This work was partially supported by the Joint Funds of the National Natural Science Foundation of China (Grant No.U24A20327), RGC Young Collaborative Research Grant No. C2005-24Y, RGC General Research Fund No. 12200725, and NSFC General Program No. 62376235.

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

# APPENDIX

## Contents

# A  Theoretical Analysis

## A.1  Basic Theorems and Corollaries Related to Statistics

We start to provide some basic theoretical results, laying the foundation for establishing the bounds of the statistics in Appendix A.6 and A.7.

**Theorem 2.** *Let $X \sim \chi^2(d, \varphi)$ follow a **noncentral** chi-squared distribution with $d$ degrees of freedom and noncentrality parameter $\varphi$. For any $t > 0$, the following tail bounds hold:*

$$P\left\{X - (d + \varphi) \geq 2\sqrt{(d + 2\varphi)t} + 2t\right\} \leq e^{-t},$$

$$P\left\{X - (d + \varphi) \leq -2\sqrt{(d + 2\varphi)t}\right\} \leq e^{-t}.$$

*Proof.* The moment-generating function of $X$ satisfies

$$E[e^{sX}] = \frac{e^{\frac{\varphi s}{1-2s}}}{(1-2s)^{d/2}} \quad (s < 1/2).$$

The log-moment generating function of $X - (d + \varphi)$ is:

$$\begin{aligned}
\log \mathbb{E}[e^{s(X-(d+\varphi))}] &= \log \mathbb{E}[e^{sX}] - s(d + \varphi) \\
&= -\frac{d}{2}\log(1-2s) + \frac{\varphi s}{1-2s} - s(d + \varphi).
\end{aligned} \tag{14}$$

For $0 < s < 1/2$, we have

$$-s - \frac{1}{2}\log(1-2s) \leq \frac{s^2}{1-2s}, \tag{15}$$

which holds because the function $\psi(s) = -s - \frac{1}{2}\log(1-2s) - \frac{s^2}{1-2s}$ satisfies $\psi'(s) = -1 + \frac{1}{1-2s} - \frac{2s-s^2}{(1-2s)^2} = -\frac{2s^2}{(1-2s)^2} \leq 0$, implying $\max_{0<s<1/2} \psi(s) < \psi(0^+) = 0$, *i.e.*, $\psi(s) \leq 0$.

Substituting Eqn. (15) into Eqn.(14), we get

$$\log \mathbb{E}[e^{s(X-(d+\varphi))}] \leq \frac{ds^2}{1-2s} + \frac{2\varphi s^2}{1-2s} = \frac{(d+2\varphi)s^2}{1-2s}. \tag{16}$$

According to the result in [68], if $\exists\, v, c > 0$, s.t. $\log \mathbb{E}[e^{uZ}] \leq \frac{vu^2}{2(1-cu)}$, then for $\forall\, t > 0$, the following inequality holds:

$$P(Z \geq ct + \sqrt{2vt}) \leq e^{-t}.$$

Applying this result to Eqn. (16), we set $Z = X - (d + \varphi)$, $v = 2(d + 2\varphi)$ and $c = 2$, then

$$P\left\{X - (d + \varphi) \geq 2\sqrt{(d + 2\varphi)t} + 2t\right\} \leq e^{-t}.$$

For $-1/2 < s < 0$, we have

$$-s - \frac{1}{2}\log(1-2s) \leq s^2, \tag{17}$$

which holds because the function $h(s) = -s - \frac{1}{2}\log(1-2s) - s^2$ satisfies $h'(s) = -1 + \frac{1}{1-2s} - 2s = \frac{4s^2}{1-2s} \geq 0$, implying $\max_{-1/2<s<0} \psi(s) < \psi(0^-) = 0$, *i.e.*, $h(s) \leq 0$.

Substituting Eqn. (17) into Eqn.(14), we get

$$\log \mathbb{E}[e^{s(X-(d+\varphi))}] \leq ds^2 + \frac{2\varphi s^2}{1-2s} = (d + \frac{2\varphi}{1-2s})s^2 \leq (d+2\varphi)s^2. \tag{18}$$

According to the result in [68], if $\exists\, v > 0$, s.t., $\log \mathbb{E}[e^{sZ}] \leq \frac{vs^2}{2}$, then for $\forall\, t > 0$, the following inequality holds:

$$P\left(Z \leq -\sqrt{2vt}\right) \leq e^{-t}.$$

Applying this result to Eqn. (18), we set $Z = X - (d + \varphi)$, $v = 2(d + 2\varphi)$, then

$$P\left\{X - (d + \varphi) \leq -2\sqrt{(d + 2\varphi)t}\right\} \leq e^{-t}.$$

$\square$

**Corollary 1.** *Given $X \sim \chi^2(d, \varphi)$, a noncentralchi-squared distribution with $d$ degrees of freedom and the noncentrality parameter $\varphi$, with probability at least $1 - \delta$, we have*

$$|X| \leq d + \varphi + \sqrt{4(d + 2\varphi)\log\left(\frac{2}{\delta}\right) + 2\log\left(\frac{2}{\delta}\right)}.$$

*Proof.* By Theorem 2, setting $e^{-t} = \frac{\delta}{2}$ yields the following inequalities:

$$P\left\{X - (d + \varphi) \geq 2\sqrt{(d + 2\varphi)\log\left(\frac{2}{\delta}\right) + 2\log\left(\frac{2}{\delta}\right)}\right\} \leq \frac{\delta}{2},$$

$$P\left\{X - (d + \varphi) \leq -2\sqrt{(d + 2\varphi)\log\left(\frac{2}{\delta}\right)}\right\} \leq \frac{\delta}{2}.$$

Combining these two inequalities, we obtain:

$$P\left\{X \geq d + \varphi + 2\sqrt{(d + 2\varphi)\log\left(\frac{2}{\delta}\right)} + 2\log\left(\frac{2}{\delta}\right) \text{ or } X \leq d + \varphi - 2\sqrt{(d + 2\varphi)\log\left(\frac{2}{\delta}\right)}\right\} \leq \delta.$$

Taking the complement of the above event, we have:

$$P\left\{d + \varphi - 2\sqrt{(d + 2\varphi)\log\left(\frac{2}{\delta}\right)} \leq X \leq d + \varphi + 2\sqrt{(d + 2\varphi)\log\left(\frac{2}{\delta}\right)} + 2\log\left(\frac{2}{\delta}\right)\right\} \geq 1 - \delta.$$

By relaxing the lower bound of $X$, we conclude

$$P\left\{|X| \leq d + \varphi + 2\sqrt{(d + 2\varphi)\log\left(\frac{2}{\delta}\right)} + 2\log\left(\frac{2}{\delta}\right)\right\} \geq 1 - \delta.$$

$\square$

## A.2 Proof of Proposition 1

**Proposition 1.** *Under the brightness constancy assumption $p(\mathbf{x} + \Delta\mathbf{x}, t + \Delta t) \approx p(\mathbf{x}, t)$ with small inter-frame motion ($\Delta t \to 0$) and inter-frame displacement ($\Delta\mathbf{x} \to 0$), we have*

$$\partial_t \log p(\mathbf{x}, t) \approx -\frac{\nabla_\mathbf{x} \log p(\mathbf{x}, t) \cdot \Delta\mathbf{x}}{\Delta t}. \tag{19}$$

*Proof.* We apply the Taylor expansion $\log p(\mathbf{x} + \Delta\mathbf{x}, t + \Delta t)$ around $(\mathbf{x}, t)$ to first order:

$$\log p(\mathbf{x} + \Delta\mathbf{x}, t + \Delta t) = \log p(\mathbf{x}, t) + \nabla_\mathbf{x} \log p(\mathbf{x}, t) \cdot \Delta\mathbf{x} + \partial_t \log p(\mathbf{x}, t) \cdot \Delta t + o(\|\Delta\mathbf{x}\|^2 + \Delta t^2),$$

where $o(\|\Delta\mathbf{x}\|^2 + \Delta t^2)$ represents higher-order infinitesimal terms.

By assumption, $\log p(\mathbf{x} + \Delta\mathbf{x}, t + \Delta t) \approx \log p(\mathbf{x}, t)$. Subtracting $\log p(\mathbf{x}, t)$ from both sides:

$$\nabla_\mathbf{x} \log p(\mathbf{x}, t) \cdot \Delta\mathbf{x} + \partial_t \log p(\mathbf{x}, t) \cdot \Delta t + o(\|\Delta\mathbf{x}\|^2 + \Delta t^2) \approx 0.$$

Under $\Delta t \to 0$ and $\Delta\mathbf{x} \to 0$, we obtain:

$$\nabla_\mathbf{x} \log p(\mathbf{x}, t) \cdot \Delta\mathbf{x} + \partial_t \log p(\mathbf{x}, t) \cdot \Delta t \approx 0.$$

Rearranging terms gives:

$$\partial_t \log p(\mathbf{x}, t) \approx -\frac{\nabla_\mathbf{x} \log p(\mathbf{x}, t) \cdot \Delta\mathbf{x}}{\Delta t}.$$

$\square$

## A.3 Derivations of Chain Rule to the Conservation of Probability Mass

For $\frac{\partial p}{\partial t} + \nabla_{\mathbf{x}} \cdot \mathbf{J} = 0$, we substitute $\mathbf{J} = p\mathbf{v}$ and then divide the entire equation by $p$ (which is strictly positive everywhere in its support), yielding:

$$\frac{1}{p} \partial_t p + \frac{1}{p} \nabla_{\mathbf{x}} \cdot (p\mathbf{v}) = 0.$$

Applying the vector calculus product rule $\nabla_{\mathbf{x}} \cdot (p\mathbf{v}) = p(\nabla_{\mathbf{x}} \cdot \mathbf{v}) + \mathbf{v} \cdot (\nabla_{\mathbf{x}} p)$ and the chain rule of calculus, $\frac{1}{p} \frac{\partial p}{\partial t} = \partial_t \log p$ and $\frac{\nabla_{\mathbf{x}} p}{p} = \nabla_{\mathbf{x}} \log p$, we obtain Eqn.(3):

$$\partial_t \log p + \nabla_{\mathbf{x}} \cdot \mathbf{v} + \mathbf{v} \cdot \nabla_{\mathbf{x}} \log p = 0.$$

This transformation does not alter the underlying fluid constraint—it is a variable change making explicit how velocity couples to log-density's temporal and spatial gradients.

## A.4 Derivations of Gradients in NSG

In the following, we provide the specific forms of the terms $\nabla_{\mathbf{x}} \log p(\mathbf{x}, t)$, $-\partial_t \log p(\mathbf{x}, t) + \lambda$ and $\mathbf{g}(\mathbf{x}, t)$ when the data are from Gaussian distributions, which will be used in Appendix A.5, A.6, A.7.

**Proposition 3.** *Given the real video distribution $p(\mathbf{x}, t) = \mathcal{N}(\mathbf{0}, \sigma(t)^2 \mathbf{I}_d)$ and the generated video distribution $q(\mathbf{y}, t) = \mathcal{N}(\boldsymbol{\mu}, \sigma(t)^2 \mathbf{I}_d)$ with $\boldsymbol{\mu} \neq \mathbf{0}$ and $\sigma(t) \neq 0$, the gradients $\nabla_{\mathbf{x}} \log p(\mathbf{x}, t)$ and $\nabla_{\mathbf{y}} \log p(\mathbf{y}, t)$ are:*

$$\nabla_{\mathbf{x}} \log p(\mathbf{x}, t) = -\frac{\mathbf{x}}{\sigma(t)^2} \sim \mathcal{N}(\mathbf{0}, \frac{1}{\sigma(t)^2} \mathbf{I}_d),$$

$$\nabla_{\mathbf{y}} \log p(\mathbf{y}, t) = -\frac{\mathbf{y}}{\sigma(t)^2} \sim \mathcal{N}(-\frac{\boldsymbol{\mu}}{\sigma(t)}, \sigma(t)^2 \mathbf{I}_d).$$

*Proof.* Recall that for a Gaussian distribution $p(\mathbf{z}) = \mathcal{N}(\boldsymbol{\nu}, \sigma^2 \mathbf{I}_d)$, the probability density function is

$$p(\mathbf{z}) = \frac{1}{(2\pi\sigma^2)^{d/2}} \exp\left(-\frac{\|\mathbf{z} - \boldsymbol{\nu}\|^2}{2\sigma^2}\right).$$

The log-density is

$$\log p(\mathbf{z}) = -\frac{d}{2} \log(2\pi\sigma^2) - \frac{\|\mathbf{z} - \boldsymbol{\nu}\|^2}{2\sigma^2}.$$

Thus, we have

$$\nabla_{\mathbf{z}} \log p(\mathbf{z}) = -\frac{\mathbf{z} - \boldsymbol{\nu}}{\sigma^2}.$$

For the real video distribution $p(\mathbf{x}, t) = \mathcal{N}(\mathbf{0}, \sigma(t)^2 \mathbf{I}_d)$, we have $\boldsymbol{\nu} = \mathbf{0}$. Taking the gradient w.r.t. $\mathbf{x}$:

$$\nabla_{\mathbf{x}} \log p(\mathbf{x}, t) = \nabla_{\mathbf{x}} \left(-\frac{\|\mathbf{x}\|^2}{2\sigma(t)^2}\right) = -\frac{\mathbf{x}}{\sigma(t)^2} \sim \mathcal{N}\left(\mathbf{0}, \frac{1}{\sigma(t)^2} \mathbf{I}_d\right).$$

For the generated video distribution $q(\mathbf{y}, t) = \mathcal{N}(\boldsymbol{\mu}, \sigma(t)^2 \mathbf{I}_d)$, evaluated under $p(\mathbf{y}, t)$), the gradient w.r.t. $\mathbf{y}$ is

$$\nabla_{\mathbf{y}} \log p(\mathbf{y}, t) = -\frac{\mathbf{y}}{\sigma(t)^2} \sim \mathcal{N}\left(-\frac{\boldsymbol{\mu}}{\sigma(t)^2}, \frac{1}{\sigma(t)^2} \mathbf{I}_d\right).$$

$\square$

**Proposition 4.** *Given the real video distribution $p(\mathbf{x}, t) = \mathcal{N}(\mathbf{0}, \sigma(t)^2 \mathbf{I}_d)$ and the generated video distribution $q(\mathbf{y}, t) = \mathcal{N}(\boldsymbol{\mu}, \sigma(t)^2 \mathbf{I}_d)$ with $\boldsymbol{\mu} \neq \mathbf{0}$ and $\sigma(t) \neq 0$, the partial derivatives $-\partial_t \log p(\mathbf{x}, t)$ and $-\partial_t \log p(\mathbf{y}, t)$ are:*

$$-\partial_t \log p(\mathbf{x}, t) = \frac{d\dot{\sigma}(t)}{\sigma(t)} - \frac{\|\mathbf{x}\|^2 \dot{\sigma}(t)}{\sigma(t)^3} \sim \frac{d\dot{\sigma}(t)}{\sigma(t)} - \frac{\dot{\sigma}(t)}{\sigma(t)} \chi^2(d),$$

$$-\partial_t \log p(\mathbf{y}, t) = \frac{d\dot{\sigma}(t)}{\sigma(t)} - \frac{\|\mathbf{y}\|^2 \dot{\sigma}(t)}{\sigma(t)^3} \sim \frac{d\dot{\sigma}(t)}{\sigma(t)} - \frac{\dot{\sigma}(t)}{\sigma(t)} \chi^2(d, \varphi),$$

*where $\dot{\sigma}(t) \triangleq \frac{d}{dt} \sigma(t)$ and $\varphi = \frac{\|\boldsymbol{\mu}\|^2}{\sigma(t)^2}$. Here, $\chi^2(d)$ is the central chi-squared distribution and $\chi^2(d, \varphi)$ is the noncentral chi-squared distribution with noncentrality parameter $\varphi$ [61].*

*Proof.* We first derive the expression for the real video distribution $p(\mathbf{x}, t)$. The log-density is

$$\log p(\mathbf{x}, t) = -\frac{d}{2} \log(2\pi\sigma(t)^2) - \frac{\|\mathbf{x}\|^2}{2\sigma(t)^2}.$$

Taking the time derivative $\partial_t$ (denoted by dot notation):

$$\partial_t \log p(\mathbf{x}, t) = -\frac{d}{2} \cdot \frac{1}{2\pi\sigma(t)^2} \cdot 2\pi \cdot 2\sigma(t)\dot{\sigma}(t) + \frac{\|\mathbf{x}\|^2}{\sigma(t)^3}\dot{\sigma}(t)$$

$$= -\frac{d\dot{\sigma}(t)}{\sigma(t)} + \frac{\|\mathbf{x}\|^2\dot{\sigma}(t)}{\sigma(t)^3}.$$

Thus, we get

$$-\partial_t \log p(\mathbf{x}, t) = \frac{d\dot{\sigma}(t)}{\sigma(t)} - \frac{\|\mathbf{x}\|^2\dot{\sigma}(t)}{\sigma(t)^3} \sim \frac{d\dot{\sigma}(t)}{\sigma(t)} - \frac{\dot{\sigma}(t)}{\sigma(t)}\chi^2(d),$$

where the last formula is based on $\|\frac{\mathbf{x}}{\sigma(t)}\|^2 \sim \chi^2(d)$.

For the generated video distribution $q(\mathbf{y}, t) = \mathcal{N}(\boldsymbol{\mu}, \sigma(t)^2\mathbf{I}_d)$, under $p(\mathbf{y}, t)$ with $\varphi = \frac{\|\boldsymbol{\mu}\|^2}{\sigma(t)^2}$, we have

$$-\partial_t \log p(\mathbf{y}, t) = \frac{d\dot{\sigma}(t)}{\sigma(t)} - \frac{\|\mathbf{y}\|^2\dot{\sigma}(t)}{\sigma(t)^3} \sim \frac{d\dot{\sigma}(t)}{\sigma(t)} - \frac{\dot{\sigma}(t)}{\sigma(t)}\chi^2(d, \varphi),$$

where the last formula is based on $\|\frac{\mathbf{y}}{\sigma(t)}\|^2 \sim \chi^2(d, \varphi)$. $\qquad\square$

## A.5 Proof of Proposition 2

**Proposition 2.** *Let the real video distribution be $p(\mathbf{x}, t) = \mathcal{N}(\mathbf{0}, \sigma(t)^2\mathbf{I}_d)$ and the generated video distribution be $q(\mathbf{y}, t) = \mathcal{N}(\boldsymbol{\mu}, \sigma(t)^2\mathbf{I}_d)$, respectively, where $\mathbf{I}_d \in \mathbb{R}^{d\times d}$ is an identity matrix and $\boldsymbol{\mu} \neq \mathbf{0} \in \mathbb{R}^d$ is the distribution shift and $\sigma(t) \neq 0$, the NSG $\mathbf{g}(\mathbf{x}, t)$ and $\mathbf{g}(\mathbf{y}, t)$ satisfy:*

$$\mathbf{g}(\mathbf{x}, t) = -\frac{\mathbf{x}/\sigma(t)^2}{D_r(\mathbf{x})}, \quad -\frac{\mathbf{x}}{\sigma(t)^2} \sim \mathcal{N}\Big(\mathbf{0}, \sigma(t)^2\mathbf{I}_d\Big), \; D_r(\mathbf{x}) \sim \lambda + \frac{d\dot{\sigma}(t)}{\sigma(t)} - \frac{\dot{\sigma}(t)}{\sigma(t)}\chi^2(d);$$

$$\mathbf{g}(\mathbf{y}, t) = -\frac{\mathbf{y}/\sigma(t)^2}{D_f(\mathbf{y})}, \quad -\frac{\mathbf{y}}{\sigma(t)^2} \sim \mathcal{N}\Big(-\frac{\boldsymbol{\mu}}{\sigma(t)}, \sigma(t)^2\mathbf{I}_d\Big), \; D_f(\mathbf{y}) \sim \lambda + \frac{d\dot{\sigma}(t)}{\sigma(t)} - \frac{\dot{\sigma}(t)}{\sigma(t)}\chi^2(d, \varphi),$$

*where $D_r(\mathbf{x}) = \lambda + \frac{d\dot{\sigma}(t)}{\sigma(t)} - \frac{\|\mathbf{x}\|^2\dot{\sigma}(t)}{\sigma(t)^3}$, $D_f(\mathbf{y}) = \lambda + \frac{d\dot{\sigma}(t)}{\sigma(t)} - \frac{\|\mathbf{y}\|^2\dot{\sigma}(t)}{\sigma(t)^3}$, $\dot{\sigma}(t) \triangleq \frac{d}{dt}\sigma(t)$, and $\varphi = \frac{\|\boldsymbol{\mu}\|^2}{\sigma(t)^2}$, $\chi^2(d)$ is the central chi-squared distribution with $d$ degrees of freedom and $\chi^2(d, \varphi)$ is the noncentral chi-squared distribution with noncentrality parameter $\varphi$ and $d$ degrees of freedom [61].*

*Proof.* According to the definition of NSG,

$$\mathbf{g}(\mathbf{x}, t) = \frac{\nabla_{\mathbf{x}} \log p(\mathbf{x}, t)}{-\partial_t \log p(\mathbf{x}, t) + \lambda}, \tag{20}$$

we can substitute the results of $\nabla_{\mathbf{x}} \log p(\mathbf{x}, t)$ in Proposition 3 and $-\partial_t \log p(\mathbf{x}, t)$ in Proposition 4 into Eqn. (20) and directly contribute to the results. $\qquad\square$

## A.6 Derivations of Upper Bounds for Gradients

Next, we present some propositions on the upper bounds that will be used in Appendix A.7.

**Proposition 5.** *Given the real video distribution $p(\mathbf{x}, t) = \mathcal{N}(\mathbf{0}, \sigma(t)^2\mathbf{I}_d)$ and the generated video distribution $q(\mathbf{y}, t) = \mathcal{N}(\boldsymbol{\mu}, \sigma(t)^2\mathbf{I}_d)$ with $\boldsymbol{\mu} \neq \mathbf{0}$ and $\sigma(t) \neq 0$, let $D_r(\mathbf{x}) = \lambda - \partial_t \log p(\mathbf{x}, t)$ and $D_f(\mathbf{y}) = \lambda - \partial_t \log p(\mathbf{y}, t)$, with probability at least $1 - \delta$, we have*

$$|D_r(\mathbf{x}) - D_f(\mathbf{y})| \le \varphi + 2\sqrt{(d + 2\varphi)\log\frac{4}{\delta}} + 2\sqrt{d\log\frac{4}{\delta}} + 2\log\frac{4}{\delta},$$

*where $\varphi = \frac{\|\boldsymbol{\mu}\|^2}{\sigma(t)^2}$.*

*Proof.* From Proposition 4, we obtain

$$|D_r(\mathbf{x}) - D_f(\mathbf{y})| = \frac{|\dot\sigma(t)|}{\sigma(t)^3}\left|\|\mathbf{x}\|^2 - \|\mathbf{y}\|^2\right|.$$

where $\dot\sigma(t) \triangleq \frac{d}{dt}\sigma(t)$.

Let $Z = \frac{\|\mathbf{x}\|^2}{\sigma(t)^2} \sim \chi^2(d)$ and $W = \frac{\|\mathbf{y}\|^2}{\sigma(t)^2} \sim \chi^2(d,\varphi)$, where $\varphi = \frac{\|\boldsymbol{\mu}\|^2}{\sigma(t)^2}$. The difference becomes

$$|D_r(\mathbf{x}) - D_f(\mathbf{y})| = \frac{|\dot\sigma(t)|}{\sigma(t)}|W - Z|.$$

To bound $|W - Z|$, we use concentration inequalities for chi-squared distributions by Theorem 2. For $Z \sim \chi^2(d)$, we have

$$P\left\{Z - d \geq 2\sqrt{d\log\tfrac{4}{\delta}} + 2\log\tfrac{4}{\delta}\right\} \leq \frac{\delta}{4},$$

$$P\left\{Z - d \leq -2\sqrt{d\log\tfrac{4}{\delta}}\right\} \leq \frac{\delta}{4}.$$

Combining these two events, we obtain

$$P\left\{-2\sqrt{d\log\tfrac{4}{\delta}} \leq Z - d \leq 2\sqrt{d\log\tfrac{4}{\delta}} + 2\log\tfrac{4}{\delta}\right\} \geq 1 - \frac{\delta}{2}. \tag{21}$$

Similarly, for $W \sim \chi^2(d,\varphi)$, we have

$$P\left\{\varphi - 2\sqrt{(d+2\varphi)\log\tfrac{4}{\delta}} \leq W - d \leq \varphi + 2\sqrt{(d+2\varphi)\log\tfrac{4}{\delta}} + 2\log\tfrac{4}{\delta}\right\} \geq 1 - \frac{\delta}{2}. \tag{22}$$

Combining the bounds in Eqn. (22) and (21), we have with probability $1 - \delta$:

$$|W - Z| \leq \varphi + 2\sqrt{(d+2\varphi)\log\tfrac{4}{\delta}} + 2\sqrt{d\log\tfrac{4}{\delta}} + 2\log\tfrac{4}{\delta}.$$

Substituting this into the expression for $|D_r(\mathbf{x}) - D_f(\mathbf{y})|$ completes the proof. $\square$

**Proposition 6.** *Given the real video distribution $p(\mathbf{x},t) = \mathcal{N}(\mathbf{0}, \sigma(t)^2\mathbf{I}_d)$ and the generated video distribution $q(\mathbf{y},t)=\mathcal{N}(\boldsymbol{\mu}, \sigma(t)^2\mathbf{I}_d)$ with $\boldsymbol{\mu}\neq\mathbf{0}$ and $\sigma(t)\neq 0$, the following inequalities hold with probability at least $1 - \delta$:*

$$1)\quad \frac{\|\mathbf{x}\|^2}{\sigma(t)^2} \leq d + \sqrt{4d\log\left(\frac{2}{\delta}\right) + 2\log\left(\frac{2}{\delta}\right)},$$

$$2)\quad \frac{\|\mathbf{y}\|^2}{\sigma(t)^2} \leq d + \varphi + \sqrt{4(d+2\varphi)\log\left(\frac{2}{\delta}\right) + 2\log\left(\frac{2}{\delta}\right)},$$

$$3)\quad \frac{\|\mathbf{x}-\mathbf{y}\|^2}{2\sigma(t)^2} \leq d + \frac{\varphi}{2} + \sqrt{4(d+\varphi)\log\left(\frac{2}{\delta}\right) + 2\log\left(\frac{2}{\delta}\right)},$$

*where $\varphi = \frac{\|\boldsymbol{\mu}\|^2}{\sigma(t)^2}$.*

*Proof.* 1) Since $\mathbf{x} \sim \mathcal{N}(\mathbf{0}, \sigma(t)^2\mathbf{I}_d)$, $\frac{\|\mathbf{x}\|^2}{\sigma(t)^2}$ follows a central chi-squared distribution $\chi^2(d)$. By the concentration inequality for central chi-squared distributions (Corollary 1), with probability $1 - \delta$:

$$\frac{\|\mathbf{x}\|^2}{\sigma(t)^2} \leq d + \sqrt{4d\log\left(\frac{2}{\delta}\right) + 2\log\left(\frac{2}{\delta}\right)}.$$

2) For $\mathbf{y} \sim \mathcal{N}(\boldsymbol{\mu}, \sigma(t)^2\mathbf{I}_d)$, $\frac{\|\mathbf{y}\|^2}{\sigma(t)^2}$ follows a noncentral chi-squared distribution $\chi^2(d,\varphi)$, where $\varphi = \frac{\|\boldsymbol{\mu}\|^2}{\sigma(t)^2}$. By Corollary 1, with probability $1 - \delta$, we have

$$\frac{\|\mathbf{y}\|^2}{\sigma(t)^2} \leq d + \varphi + \sqrt{4(d+2\varphi)\log\left(\frac{2}{\delta}\right) + 2\log\left(\frac{2}{\delta}\right)}.$$

3) Since $\mathbf{x} \sim \mathcal{N}(\mathbf{0}, \sigma(t)^2 \mathbf{I}_d)$ and $\mathbf{y} \sim \mathcal{N}(\boldsymbol{\mu}, \sigma(t)^2 \mathbf{I}_d)$, their difference $\mathbf{z}$ satisfies:

$$\mathbf{x} - \mathbf{y} \sim \mathcal{N}(-\boldsymbol{\mu}, 2\sigma(t)^2 \mathbf{I}_d).$$

Thus, $\frac{\|\mathbf{x}-\mathbf{y}\|^2}{2\sigma(t)^2}$ follows a noncentral chi-squared distribution $\chi^2(d, \varphi/2)$, where $\varphi/2 = \frac{\|\boldsymbol{\mu}\|^2}{2\sigma(t)^2}$.

By Corollary 1, with probability $1 - \delta$, we have

$$\frac{\|\mathbf{x} - \mathbf{y}\|^2}{2\sigma(t)^2} \leq d + \frac{\varphi}{2} + \sqrt{4(d + \varphi) \log\left(\frac{2}{\delta}\right)} + 2\log\left(\frac{2}{\delta}\right).$$

$\square$

## A.7  Proof of Theorem 1

Given a video $\mathbf{x} \in \mathbb{R}^{T \times d}$, its NSG Feature is $\mathbf{G}(\mathbf{x}) = \{\mathbf{g}(\mathbf{x}, t)\}_{t=1}^{T}$, where $\mathbf{g}(\mathbf{x}, t)$ is defined as:

$$\mathbf{g}(\mathbf{x}, t) = \frac{\nabla_\mathbf{x} \log p(\mathbf{x}, t)}{-\partial_t \log p(\mathbf{x}, t) + \lambda}, \tag{23}$$

Here, $\lambda > 0$ and $p(\mathbf{x}, t)$ is the probability density of the real video parameterized by time $t$.

Note that Theorem 1 share a common lower bound $C$ on both $D_r(\mathbf{x}) = \lambda - \partial_t \log p(\mathbf{x}, t)$ and $D_f(\mathbf{y}) = \lambda - \partial_t \log p(\mathbf{y}, t)$. We first derive the conditions for $D_r(\mathbf{x}) > C > 0$ and $D_f(\mathbf{y}) > C > 0$ to hold in Proposition 7. The same analytical approach can be extended to examine other cases.

**Proposition 7.** *Let the real video distribution be* $\mathbf{x} \sim \mathcal{N}(\mathbf{0}, \sigma(t)^2 \mathbf{I}_d)$ *and the generated video distribution be* $\mathbf{y} \sim \mathcal{N}(\boldsymbol{\mu}, \sigma(t)^2 \mathbf{I}_d)$, *respectively, where* $\mathbf{I}_d \in \mathbb{R}^{d \times d}$ *is an identity matrix and* $\boldsymbol{\mu} \neq \mathbf{0} \in \mathbb{R}^d$ *is the distribution shift, we have* $D_r(\mathbf{x}) > C > 0$ *and* $D_f(\mathbf{y}) > C > 0$ *with probability at least* $1 - \delta$, *provided* $C$ *and* $\lambda$ *meet the following conditions:*

1)  *Case 1* $(\frac{\dot{\sigma}(t)}{\sigma(t)} > 0)$:

$$C = \lambda - \frac{\dot{\sigma}(t)}{\sigma(t)} \cdot \varphi + \frac{2\dot{\sigma}(t)}{\sigma(t)} \sqrt{d \log\left(\frac{2}{\delta}\right)} + \frac{2\dot{\sigma}(t)}{\sigma(t)} \log\left(\frac{2}{\delta}\right),$$

$$\lambda > \frac{\dot{\sigma}(t)}{\sigma(t)}(d + \varphi) - \frac{2\dot{\sigma}(t)}{\sigma(t)} \sqrt{d \log\left(\frac{2}{\delta}\right)} - \frac{2\dot{\sigma}(t)}{\sigma(t)} \log\left(\frac{2}{\delta}\right).$$

*where* $\varphi = \frac{\|\boldsymbol{\mu}\|^2}{\sigma(t)^2}$.

2)  *Case 2* $(\frac{\dot{\sigma}(t)}{\sigma(t)} < 0)$:

$$C = \lambda - \frac{2\dot{\sigma}(t)}{\sigma(t)} \sqrt{d \log\left(\frac{2}{\delta}\right)},$$

$$\lambda > \frac{2\dot{\sigma}(t)}{\sigma(t)} \sqrt{d \log\left(\frac{2}{\delta}\right)}.$$

*Proof.* Let $W = \frac{\|\mathbf{y}\|^2}{\sigma(t)^2} \sim \chi^2(d, \varphi)$, where $\varphi = \frac{\|\boldsymbol{\mu}\|^2}{\sigma(t)^2}$. From Theorem 2, we have

$$P\left\{ W - (d + \varphi) \geq 2\sqrt{(d + 2\varphi) \log\left(\frac{2}{\delta}\right)} + 2\log\left(\frac{2}{\delta}\right) \right\} \leq \frac{\delta}{2}, \tag{24}$$

$$P\left\{ W - (d + \varphi) \leq -2\sqrt{(d + 2\varphi) \log\left(\frac{2}{\delta}\right)} \right\} \leq \frac{\delta}{2}. \tag{25}$$

1)  **Case 1** $(\frac{\dot{\sigma}(t)}{\sigma(t)} > 0)$:

Substituting $D_f(\mathbf{y}) = \lambda - \frac{\dot{\sigma}(t)}{\sigma(t)}(W - d)$ into the bound Eqn. (24):

$$P\left\{D_f(\mathbf{y}) \leq \lambda - \frac{\dot{\sigma}(t)}{\sigma(t)} \cdot \varphi + \frac{2\dot{\sigma}(t)}{\sigma(t)}\sqrt{(d + 2\varphi)\log\left(\frac{2}{\delta}\right)} + \frac{2\dot{\sigma}(t)}{\sigma(t)}\log\left(\frac{2}{\delta}\right)\right\} \leq \frac{\delta}{2}.$$

Thus, the following inequalities hold with probability at least $1 - \delta/2$:

$$D_f(\mathbf{y}) \geq \lambda - \frac{\dot{\sigma}(t)}{\sigma(t)} \cdot \varphi + \frac{2\dot{\sigma}(t)}{\sigma(t)}\sqrt{(d + 2\varphi)\log\left(\frac{2}{\delta}\right)} + \frac{2\dot{\sigma}(t)}{\sigma(t)}\log\left(\frac{2}{\delta}\right),$$

$$D_r(\mathbf{x}) \geq \lambda + \frac{2\dot{\sigma}(t)}{\sigma(t)}\sqrt{d\log\left(\frac{2}{\delta}\right)} + \frac{2\dot{\sigma}(t)}{\sigma(t)}\log\left(\frac{2}{\delta}\right).$$

To ensure $D_r(\mathbf{x}) > C > 0$ and $D_f(\mathbf{y}) > C > 0$, we can select $C$ and $\lambda$ as:

$$C = \lambda - \frac{\dot{\sigma}(t)}{\sigma(t)} \cdot \varphi + \frac{2\dot{\sigma}(t)}{\sigma(t)}\sqrt{d\log\left(\frac{2}{\delta}\right)} + \frac{2\dot{\sigma}(t)}{\sigma(t)}\log\left(\frac{2}{\delta}\right),$$

$$\lambda > \frac{\dot{\sigma}(t)}{\sigma(t)}(d + \varphi) - \frac{2\dot{\sigma}(t)}{\sigma(t)}\sqrt{d\log\left(\frac{2}{\delta}\right)} - \frac{2\dot{\sigma}(t)}{\sigma(t)}\log\left(\frac{2}{\delta}\right).$$

2) **Case 2** $(\frac{\dot{\sigma}(t)}{\sigma(t)} < 0)$:

Substituting $D_f(\mathbf{y}) = \lambda - \frac{\dot{\sigma}(t)}{\sigma(t)}(W - d)$ into the bound Eqn. (25):

$$P\left\{D_f(\mathbf{y}) \leq \lambda - \frac{\dot{\sigma}(t)}{\sigma(t)} \cdot \varphi - \frac{2\dot{\sigma}(t)}{\sigma(t)}\sqrt{(d + 2\varphi)\log\left(\frac{2}{\delta}\right)}\right\} \leq \frac{\delta}{2}.$$

Thus, the following inequalities hold with probability at least $1 - \delta/2$:

$$D_f(\mathbf{y}) \geq \lambda - \frac{\dot{\sigma}(t)}{\sigma(t)} \cdot \varphi - \frac{2\dot{\sigma}(t)}{\sigma(t)}\sqrt{(d + 2\varphi)\log\left(\frac{2}{\delta}\right)},$$

$$D_r(\mathbf{x}) \geq \lambda - \frac{2\dot{\sigma}(t)}{\sigma(t)}\sqrt{d\log\left(\frac{2}{\delta}\right)}.$$

To ensure $D_r(\mathbf{x}) > C > 0$ and $D_f(\mathbf{y}) > C > 0$, we can select $C$ and $\lambda$ as:

$$C = \lambda - \frac{2\dot{\sigma}(t)}{\sigma(t)}\sqrt{d\log\left(\frac{2}{\delta}\right)},$$

$$\lambda > \frac{2\dot{\sigma}(t)}{\sigma(t)}\sqrt{d\log\left(\frac{2}{\delta}\right)}.$$

$\square$

Building upon the established Propositions 2, 5, 6 and 7, we next prove Theorem 1.

**Theorem 1**. *Let the real video distribution be* $\mathbf{x} \sim \mathcal{N}(\mathbf{0}, \sigma(t)^2 \mathbf{I}_d)$ *and the generated video distribution be* $\mathbf{y} \sim \mathcal{N}(\boldsymbol{\mu}, \sigma(t)^2 \mathbf{I}_d)$, *respectively, where* $\mathbf{I}_d \in \mathbb{R}^{d \times d}$ *is an identity matrix and* $\boldsymbol{\mu} \neq \mathbf{0} \in \mathbb{R}^d$ *is the distribution shift. Given* $\mathbf{G}(\mathbf{x}) = \{\mathbf{g}(\mathbf{x}, t)\}_{t=1}^T$, *denote* $\varphi = \|\boldsymbol{\mu}\|^2 / \sigma(t)^2$ *and assume* $|-\partial_t \log p(\mathbf{x}, t) + \lambda| \geq C > 0$ *and* $|-\partial_t \log p(\mathbf{y}, t) + \lambda| \geq C > 0$, *with probability at least* $1 - \delta$, *we have*

$$\|\mathbf{G}(\mathbf{x}) - \mathbf{G}(\mathbf{y})\|^2 \leq \mathcal{O}\left(\frac{T}{C^4 \sigma(t)^2}\left[\varphi d + d^2 + \varphi + \log\frac{T}{\delta} \cdot (\varphi + d) + \log^2\frac{T}{\delta}\right]\right).$$

*Proof.* Based on the definition of $\mathbf{G}(\mathbf{x})$, we have

$$\|\mathbf{G}(\mathbf{x}) - \mathbf{G}(\mathbf{y})\|^2 = \sum_{t=1}^{T} \|\mathbf{g}(\mathbf{x}, t) - \mathbf{g}(\mathbf{y}, t)\|^2. \tag{26}$$

Next, we focus on the bound of $\mathbf{g}(\mathbf{x}, t) - \mathbf{g}(\mathbf{y}, t)$.

Let $D_r(\mathbf{x}) = \lambda - \partial_t \log p(\mathbf{x}, t)$ and $D_f(\mathbf{y}) = \lambda - \partial_t \log p(\mathbf{y}, t)$, by Propositions 3 and 4, we have

$$\begin{aligned}
&\|\mathbf{g}(\mathbf{x}, t) - \mathbf{g}(\mathbf{y}, t)\|^2 \\
&= \left\| \frac{\mathbf{x}/\sigma(t)^2}{D_r(\mathbf{x})} - \frac{\mathbf{y}/\sigma(t)^2}{D_f(\mathbf{y})} \right\|^2 \\
&= \left\| \frac{\mathbf{x}/\sigma(t)^2}{D_r(\mathbf{x})} - \frac{\mathbf{x}/\sigma(t)^2}{D_f(\mathbf{y})} + \frac{\mathbf{x}/\sigma(t)^2}{D_f(\mathbf{y})} - \frac{\mathbf{y}/\sigma(t)^2}{D_f(\mathbf{y})} \right\|^2 \\
&\leq 2 \left\| \frac{\mathbf{x}/\sigma(t)^2}{D_r(\mathbf{x})} - \frac{\mathbf{x}/\sigma(t)^2}{D_f(\mathbf{y})} \right\|^2 + 2 \left\| \frac{\mathbf{x}/\sigma(t)^2}{D_f(\mathbf{y})} - \frac{\mathbf{y}/\sigma(t)^2}{D_f(\mathbf{y})} \right\|^2 \\
&= 2 \left( \frac{1}{D_r(\mathbf{x})} - \frac{1}{D_f(\mathbf{y})} \right)^2 \cdot \frac{\|\mathbf{x}\|^2}{\sigma(t)^4} + 2 \left( \frac{1}{D_f(\mathbf{y})} \right)^2 \cdot \frac{\|\mathbf{x} - \mathbf{y}\|^2}{\sigma(t)^4} \\
&= 2 \left( \frac{D_f(\mathbf{y}) - D_r(\mathbf{x})}{D_r(\mathbf{x}) D_f(\mathbf{y})} \right)^2 \cdot \frac{\|\mathbf{x}\|^2}{\sigma(t)^4} + 2 \left( \frac{1}{D_f(\mathbf{y})} \right)^2 \cdot \frac{\|\mathbf{x} - \mathbf{y}\|^2}{\sigma(t)^4} \\
&\leq 2 \frac{|D_f(\mathbf{y}) - D_r(\mathbf{x})|^2}{C^4} \cdot \frac{\|\mathbf{x}\|^2}{\sigma(t)^4} + \frac{2}{C^2} \cdot \frac{\|\mathbf{x} - \mathbf{y}\|^2}{\sigma(t)^4}. \tag{27}
\end{aligned}$$

For the first term in Eqn. (27), according to Proposition 5, with probability at least $1 - 2\delta/3$, we have

$$\begin{aligned}
&2 \frac{|D_f(\mathbf{y}) - D_r(\mathbf{x})|^2}{C^4} \cdot \frac{\|\mathbf{x}\|^2}{\sigma(t)^4} \\
&\leq \frac{2}{C^4 \sigma(t)^2} \left( \varphi + 2\sqrt{(d + 2\varphi) \log \tfrac{12}{\delta}} + 2\sqrt{d \log \tfrac{12}{\delta}} + 2 \log \tfrac{12}{\delta} \right) \cdot \left( d + \sqrt{4d \log \left( \tfrac{6}{\delta} \right)} + 2 \log \left( \tfrac{6}{\delta} \right) \right) \\
&\leq \frac{2}{C^4 \sigma(t)^2} \left( \varphi + 4\sqrt{(d + 2\varphi) \log \tfrac{12}{\delta}} + 2 \log \tfrac{12}{\delta} \right) \cdot \left( d + \sqrt{4d \log \left( \tfrac{6}{\delta} \right)} + 2 \log \left( \tfrac{6}{\delta} \right) \right). \tag{28}
\end{aligned}$$

For the second term in Eqn. (27), applying Proposition 6, with probability at least $1 - \delta/3$, we have

$$\frac{2}{C^2} \cdot \frac{\|\mathbf{x} - \mathbf{y}\|^2}{\sigma(t)^4} \leq \frac{4}{C^2 \sigma(t)^2} \left( d + \frac{\varphi}{2} + \sqrt{4(d + \varphi) \log \left( \frac{6}{\delta} \right)} + 2 \log \left( \frac{6}{\delta} \right) \right). \tag{29}$$

For simplicity, let $L = \log \frac{12}{\delta}$. Since $\log \frac{6}{\delta} = L - \log 2 < L$, we have

$$\begin{aligned}
2 \frac{|D_f(\mathbf{y}) - D_r(\mathbf{x})|^2}{C^4} \cdot \frac{\|\mathbf{x}\|^2}{\sigma(t)^4} &\leq \frac{2}{C^4 \sigma(t)^2} \left( \varphi + 4\sqrt{(d + 2\varphi)L} + 2L \right) \cdot \left( d + 2\sqrt{dL} + 2L \right) \\
&\leq \frac{2}{C^4 \sigma(t)^2} \left( \varphi + 2(d + 2\varphi) + L + 2L \right) \cdot \left( d + d + L + 2L \right) \\
&= \frac{2}{C^4 \sigma(t)^2} (5\varphi + 2d + 3L)(2d + 3L) \\
&= \frac{2}{C^4 \sigma(t)^2} \left( 10\varphi d + 4d^2 + 3L \cdot (5\varphi + 4d) + 9L^2 \right). \tag{30}
\end{aligned}$$

$$\begin{aligned}
\frac{2}{C^2} \cdot \frac{\|\mathbf{x} - \mathbf{y}\|^2}{\sigma(t)^4} &\leq \frac{4}{C^2 \sigma(t)^2} \left( d + \frac{\varphi}{2} + 2\sqrt{(d + \varphi)L} + 2L \right) \\
&\leq \frac{4}{C^2 \sigma(t)^2} \left( d + \frac{\varphi}{2} + (d + \varphi)L + 2L \right) \\
&= \frac{4}{C^2 \sigma(t)^2} \left( d + \frac{\varphi}{2} + L(d + \varphi + 2) \right). \tag{31}
\end{aligned}$$

Combining Eqn. (30) and (31) and (27), and substituting $L = \log \frac{12}{\delta}$, with probability at least $1 - \delta$, we have

$$\|\mathbf{g}(\mathbf{x}, t) - \mathbf{g}(\mathbf{y}, t)\|^2$$
$$\leq \frac{2}{C^4 \sigma(t)^2} \left(10\varphi d + 4d^2 + 3L \cdot (5\varphi + 4d) + 9L^2\right) + \frac{4}{C^2 \sigma(t)^2} \left(d + \frac{\varphi}{2} + L(d + \varphi + 2)\right).$$

For further simplicity, note that $\frac{1}{C^2} \leq \frac{1}{C^4}$, we obtain

$$\|\mathbf{g}(\mathbf{x}, t) - \mathbf{g}(\mathbf{y}, t)\|^2$$
$$\leq \frac{2}{C^4 \sigma(t)^2} \left(10\varphi d + 4d^2 + 2d + \varphi + L \cdot (17\varphi + 14d + 4) + 9L^2\right)$$
$$= \frac{2}{C^4 \sigma(t)^2} \left(10\varphi d + 4d^2 + 2d + \varphi + \log \frac{12}{\delta} \cdot (17\varphi + 14d + 4) + 9\log^2 \frac{12}{\delta}\right). \qquad (32)$$

Summing over time steps $t = 1, \ldots, T$, and replacing $\delta$ in Eqn. (32) into $\delta/T$, we get:

$$\|\mathbf{G}(\mathbf{x}) - \mathbf{G}(\mathbf{y})\|^2 \leq \frac{2T}{C^4 \sigma(t)^2} \left(10\varphi d + 4d^2 + 2d + \varphi + \log \frac{12T}{\delta} \cdot (17\varphi + 14d + 4) + 9\log^2 \frac{12T}{\delta}\right).$$

Thus, we obtain the final results

$$\|\mathbf{G}(\mathbf{x}) - \mathbf{G}(\mathbf{y})\|^2 \leq \mathcal{O}\left(\frac{T}{C^4 \sigma(t)^2} \left[\varphi d + d^2 + \varphi + \log \frac{T}{\delta} \cdot (\varphi + d) + \log^2 \frac{T}{\delta}\right]\right).$$

$\square$

# B More Related Work

**Maximum Mean Discrepancy (MMD).** Maximum Mean Discrepancy (MMD) is an effective metric for two-sample testing to assess whether two samples originate from the same distribution [69, 33, 70, 34, 71, 72, 73]. Originally introduced by Müller et al. [72] as an instance of an integral probability metric, MMD admits several sample-based estimators. Particularly, Gretton et al. [33] introduce the U-statistic estimator, which is unbiased for the squared MMD and achieves near-minimal variance among all unbiased alternatives. Further, Tolstikhin et al. [74] derive lower bounds on the estimation error of MMD under finite samples when employing a radial universal kernel.

Building upon the traditional formulation of MMD, recent advancements incorporate learnable kernels to enhance its discriminative capability. Liu et al. [34] develop a data-splitting strategy for kernel optimization and selection, effectively addressing the kernel adaptation challenges for complex-data scenarios. Kim et al. [75] propose an adaptive two-sample test designed for comparing two Hölder densities supported on the $d$-dimensional unit ball. In addition, Zhang et al. [48] introduce MMD-MP, a multi-population aware optimization framework to further improve the stability of kernel-based MMD training. At present, MMD has been extensively applied to distributional measurement and discrepancy detection tasks across both textual and visual modalities [45, 48, 46].

# C More Details for Experiment Settings

## C.1 More Details on Datasets

**GenVideo** [25] is a large-scale benchmark for AI-generated video detection, comprising 1.22 million real videos and 1.05 million AI-generated videos. The real video collection aggregates content from three established datasets: MSR-VTT (web video clips) [65], Kinetics-400 (human action videos) [62], and Youku-mPLUG (diverse online videos captured from Youku.com) [76]. The AI-generated portion features videos produced by 19 distinct generation models, including both open-source implementations (ZeroScope [77], I2VGen-XL [78], SVD [1], VideoCrafter [7], DynamiCrafter [8], Stable Diffusion(SD) [79], SEINE [63], Latte [80], OpenSora [81], ModelScope [82], HotShot [83], Show-1 [84], Gen2 [85], Crafter [86], Lavie [87]) and commercial closed-source systems (Pika, Sora, MoonValley, MorphStudio). This dataset spans multiple generation paradigms, including text-to-video and image-to-video synthesis. Throughout all experiments, we filter videos with less than 8 frames and only uniformly sample 8 frames for each video during training and testing.

## C.2 More Details on Evaluation Metrics

Video generation detection is inherently a binary classification task. Here, we introduce four fundamental evaluation metrics of binary classification: **True Positive** (TP) means correctly predicted positive instances (ground truth is positive, prediction is positive).**True Negative** (TN) means correctly predicted negative instances (ground truth is negative, prediction is negative). **False Positive** (FP) means incorrectly predicted positive instances (ground truth is negative, prediction is positive). **False Negative** (FN) means incorrectly predicted negative instances (ground truth is positive, prediction is negative).

**AUROC** denotes the Area Under the Receiver Operating Characteristic Curve [67, 88], which is a widely used statistic for assessing the discriminatory capacity of distribution models. Formally,

$$AUROC = \int TP(t)FP(t)dt,$$

where $TP(t) = TP(t)/(TP(t) + FN(t))$ is the true positive rate and $FP(t) = FP(t)/(FP(t) + TN(t))$ is false positive rate with a threshed $t$.

**Accuracy** (ACC) measures the model's overall correctness in classification by calculating the ratio of correctly predicted instances (both true positive and true negative) to the total instances.

$$Accuracy = \frac{TP + TN}{TP + TN + FP + FN}.$$

**Recall** [66] evaluates the model's ability to identify all relevant instances of a class, measuring the proportion of true positives among all actual positive instances. It emphasizes minimizing false

negatives, ensuring comprehensive coverage of positive cases.

$$\text{Recall} = \frac{TP}{TP + FN}.$$

**Precision** [66] quantifies the model's capability to avoid false positives by calculating the proportion of true positives among all predicted positive instances. It ensures reliability in positive predictions.

$$\text{Precision} = \frac{TP}{TP + FP}.$$

**F1-score** [66] balances Precision and Recall using their harmonic mean, providing a robust metric for scenarios with imbalanced class distributions. It penalizes extreme biases toward either precision or recall.

$$\text{F1-score} = 2 \times \frac{\text{Precision} \times \text{Recall}}{\text{Precision} + \text{Recall}}.$$

### C.3 Implementation Details on NSG-VD

In our NSG-VD, we employ the pre-trained diffusion model $s_\theta$ of Guided Diffusion using the $256 \times 256$ unconditional checkpoint from the guided-diffusion library [3] following [56]. For a given video $\mathbf{x}$ at $t$-th frame, we compute its score feature $\nabla_{\mathbf{x}} \log p(\mathbf{x}, t)$ by diffusing $\mathbf{x}_t$ at diffusion timestep $5/1,000$ and passing it through $s_\theta$. For the deep kernel $\phi_{\mathbf{G}}$, we employ a single-layer of Swin transformer [89], mapping input features of dimension $8 \times 224 \times 224$ to a 300-dimensional output.

We conduct our experiments on a server with 1× NVIDIA RTX 3090 GPU using Python 3.10.17 and Pytorch 2.7.0. We use Adam optimizer [90] to optimize the kernel parameters $\omega$ with batchsize 24, learning rate 0.0001, weight decay 0.1, $\sigma_\phi = 0.1$ and $\sigma_\Phi = 100$. For the testing, we set the decision threshold $\tau = 1$ in Eqn. (11). The overall algorithms for training and testing are in Alg. 1 and 2.

### C.4 Pseudo Code of NSG-VD

---
**Algorithm 1** Training deep kernel of MMD
---
**Input:** Real and generated videos $S_{\mathbb{P}}^{tr}$, $S_{\mathbb{Q}}^{tr}$;
$\omega \leftarrow \omega_0$; $\lambda \leftarrow 10^{-10}$; learning rate $\eta$;
**for** $r = 1, 2, \ldots, r_{max}$ **do**
  $k_\omega \leftarrow$ kernel function using Eqn. (12);
  $M(\omega) \leftarrow \widehat{\text{MPP}}_u(S_{\mathbb{P}}^{tr}, S_{\mathbb{Q}}^{tr}; k_\omega)$;
  $V_\lambda(\omega) \leftarrow \hat{\sigma}^2(S_{\mathbb{P}}^{tr}, S_{\mathbb{Q}}^{tr}; k_\omega)$ using Eqn. (13);
  $\hat{J}_\lambda(\omega) \leftarrow M(\omega)/\sqrt{V_\lambda(\omega)}$;
  $\omega \leftarrow \omega + \eta \nabla_{\text{Adam}} \hat{J}_\lambda(\omega)$;
**end for**
**Output:** $k_\omega^*$

---
**Algorithm 2** Detecting videos with NSG-VD
---
**Input:** Referenced videos $S_{\mathbb{P}}^{re}$, testing videos $S_{\mathbb{Q}}^{te}$;
decision $f(\cdot)$; deep kernel $k_\omega$; threshold $\tau$;
**for** $\mathbf{x}_i$ in $S_{\mathbb{Q}}^{te}$ **do**
  $Q_i \leftarrow \widehat{\text{MMD}}_b^2(S_{\mathbb{P}}^{re}, \{\mathbf{x}_i\}; k_\omega)$ using Eqn. (10);
  $f(\mathbf{x}_i; S_{\mathbb{P}}^{re}, k_\omega, \tau) = \mathbb{I}(Q_i > \tau)$;
  Obtaining $f(\mathbf{x}_i)$ using Eqn. (11);
**end for**
**Output:** Predictions $\{f(\mathbf{x}_i)\}$ of the testing set

---

[3] https://github.com/openai/guided-diffusion

# D   More Experimental Results

## D.1   More Results on Standard Evaluation

To demonstrate the statistical robustness and reproducibility of our proposed NSG-VD method, we report standard deviations of four metrics across 10 datasets with three different seeds (Table 5). The table shows that our NSG-VD achieves consistently high performance with minimal variance (*e.g.*, $0.41\%$ for Recall, $0.87\%$ for Accuracy), indicating strong reliability and repeatability of our methods.

Table 5: Standard deviations of NSG-VD with three different random seeds (%), where we train all models with $10,000$ real and generated videos from Kinetics-400 and SEINE, respectively.

| Dataset | Recall | Accuracy | F1 | AUROC |
|---|---|---|---|---|
| ModelScope | $92.78 \pm 0.48$ | $84.31 \pm 1.88$ | $85.54 \pm 1.54$ | $92.49 \pm 3.07$ |
| MorphStudio | $99.44 \pm 0.96$ | $86.53 \pm 2.64$ | $88.10 \pm 2.14$ | $97.08 \pm 0.92$ |
| MoonValley | $100.00 \pm 0.00$ | $87.64 \pm 0.64$ | $89.00 \pm 0.50$ | $99.05 \pm 0.30$ |
| HotShot | $100.00 \pm 0.00$ | $88.06 \pm 2.30$ | $89.35 \pm 1.83$ | $98.64 \pm 0.49$ |
| Show1 | $100.00 \pm 0.00$ | $88.89 \pm 2.55$ | $90.03 \pm 2.08$ | $96.96 \pm 0.93$ |
| Gen2 | $98.61 \pm 1.73$ | $87.08 \pm 2.20$ | $88.43 \pm 1.86$ | $95.27 \pm 2.17$ |
| Crafter | $99.72 \pm 0.48$ | $88.89 \pm 1.05$ | $89.98 \pm 0.86$ | $98.07 \pm 0.41$ |
| Lavie | $98.61 \pm 0.96$ | $88.75 \pm 2.17$ | $89.77 \pm 1.85$ | $95.32 \pm 2.92$ |
| Sora | $94.05 \pm 1.03$ | $84.12 \pm 1.97$ | $83.85 \pm 1.56$ | $93.10 \pm 1.36$ |
| WildScrape | $91.67 \pm 2.21$ | $85.41 \pm 2.60$ | $86.29 \pm 2.37$ | $92.75 \pm 0.31$ |
| Avg. | $97.49 \pm 0.41$ | $86.97 \pm 0.87$ | $88.04 \pm 0.74$ | $95.87 \pm 0.87$ |

## D.2   More Results on Impact of Spatial Gradients and Temporal Derivatives

To comprehensively analyze how spatial gradients and temporal derivatives contribute to detection performance across diverse generative paradigms, we include detailed results across 10 diverse generative paradigms in Table 6. The spatial gradients achieve strong performance across most generated models (*e.g.*, $81.67\%$ Recall on ModelScope, $97.50\%$ Recall on MorphStudio), with only minor performance gaps on models like HotShot ($72.23\%$ Recall) and Show1 ($74.17\%$ Recall).

In contrast, the temporal derivatives show complementary strengths and relatively better detection capabilities on these challenging cases, notably achieving $75.40\%$ Recall on HotShot and $77.60\%$ Recall on Show1, where temporal dynamics (*e.g.*, rapid motion transitions in HotShot dataset) may play a more pronounced role in exposing synthetic anomalies.

Notably, our proposed NSG-VD demonstrates superior reliability by integrating both components, achieving an average F1-score of $90.87\%$, a significant improvement over individual features. This highlights the complementary nature of spatiotemporal dynamics modeling, enabling consistent detection performance even when individual cues exhibit dataset-specific limitations.

Table 6: Impact of spatial gradients and temporal derivatives across 10 generated models (%), where we train all models with $10,000$ real and generated videos from Kinetics-400 and Pika, respectively.

| Method | Metric | Model Scope | Morph Studio | Moon Valley | HotShot | Show1 | Gen2 | Crafter | Lavie | Sora | Wild Scrape | Avg. |
|---|---|---|---|---|---|---|---|---|---|---|---|---|
| Spatial Gradients | Recall | 81.67 | 97.50 | 100.00 | 72.23 | 74.17 | 98.33 | 98.13 | 77.12 | 98.21 | 82.50 | 87.99 |
| | Accuracy | 77.50 | 89.58 | 87.08 | 75.00 | 76.25 | 87.92 | 88.75 | 76.67 | 88.39 | 81.25 | 82.84 |
| | F1 | 78.40 | 90.35 | 88.56 | 74.36 | 75.74 | 89.06 | 89.73 | 76.86 | 89.43 | 81.48 | 83.40 |
| | AUROC | 87.32 | 98.43 | 99.99 | 78.76 | 82.72 | 98.98 | 98.64 | 84.15 | 99.01 | 90.52 | 91.85 |
| Tmporal Derivative | Recall | 52.60 | 40.20 | 58.40 | 75.40 | 77.60 | 49.00 | 72.40 | 47.20 | 60.71 | 70.00 | 60.35 |
| | Accuracy | 67.40 | 61.20 | 70.30 | 78.80 | 79.90 | 65.60 | 77.30 | 64.70 | 69.64 | 76.10 | 71.09 |
| | F1 | 61.74 | 50.89 | 66.29 | 78.05 | 79.43 | 58.75 | 76.13 | 57.21 | 66.67 | 74.55 | 66.97 |
| | AUROC | 72.97 | 68.64 | 81.46 | 87.09 | 87.80 | 74.48 | 84.24 | 72.97 | 76.28 | 83.62 | 78.95 |
| NSG-VD (Ours) | Recall | 68.33 | 98.33 | 100.00 | 92.50 | 87.50 | 80.00 | 98.33 | 94.17 | 78.57 | 82.50 | 88.02 |
| | Accuracy | 81.67 | 98.33 | 96.67 | 91.67 | 90.83 | 88.33 | 95.83 | 94.17 | 88.39 | 88.75 | 91.46 |
| | F1 | 78.85 | 98.33 | 96.77 | 91.74 | 90.52 | 87.27 | 95.93 | 94.17 | 87.13 | 88.00 | 90.87 |
| | AUROC | 92.26 | 98.66 | 98.15 | 94.45 | 96.38 | 94.83 | 98.16 | 97.41 | 96.40 | 94.73 | 96.14 |

# E   More Discussions on NSG-VD

## E.1   Efficiency of NSG-VD

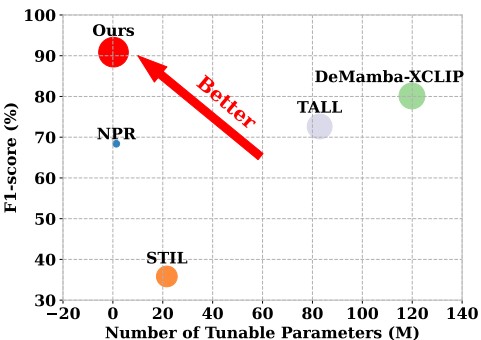

| Method | All Params(M) ↓ | Tun. Params(M) ↓ | F1(%) ↑ |
|---|---|---|---|
| DeMamba | 119.89 | 119.89 | 80.12 |
| TALL | 82.89 | 82.89 | 72.63 |
| STIL | 21.63 | 21.63 | 35.82 |
| NPR | **1.37** | 1.37 | 68.39 |
| NSG-VD (Ours) | 527.45 | **0.25** | **90.87** |

Figure 4: Comparisons with baselines in terms of training costs and performance (%), where we train all models with $10,000$ real and generated videos from Kinetics-400 and Pika, respectively.

**Performance vs. Training Cost Analysis.** We compare the number of trainable parameters and detection performance of NSG-VD against state-of-the-art baselines. As shown in Figure 4, our NSG-VD achieves a 90.87% F1-score with only 0.25 M trainable parameters, demonstrating superior parameter efficiency compared to methods like DeMamba (80.12% F1-score, 119.89 M parameters) and TALL (72.63% F1-score, 82.89 M parameters). This demonstrates that NSG-VD's physics-driven design enables high accuracy through minimal parameter tuning.

Existing baselines struggle to balance parameter scale and performance. NPR achieves only 68.39% F1-score despite its minimal trainable parameters (1.37 M), while STIL requires 21.63 M parameters to attain 35.82% F1-score—a suboptimal trade-off compared to NSG-VD's superior performance with 100× fewer parameters. These results underscore the limitations of conventional artifact-driven frameworks in effective parameter budget utilization, further validating the importance of our physics-guided spatiotemporal modeling paradigm for AI-generated video detection.

**Performance vs. Inference Time Analysis.** We evaluate the efficiency of our NSG-VD by analyzing both detection performance and computational overhead under the same setting as Table 1. As shown in Table 7, our NSG-VD achieves superior detection performance (*e.g.*, 97.13% Recall, 87.45% F1-score) with a practical inference latency of $0.3605s$ per video, which remains viable for non-real-time applications (*e.g.*, judicial video evidence analysis) despite being slower than other baselines. This latency stems from our current implementation of pre-trained diffusion models for gradient estimation—a design choice prioritizing theoretical validation over computational optimization.

Importantly, this current implementation prioritizes accuracy over speed to establish the theoretical and empirical validity of physics-guided spatiotemporal modeling. Empirically, we observe that the inference speed can be greatly enhanced with minimal performance degradation by scaling the resolution of pre-trained diffusion models, *e.g.*, reducing resolution to $128 \times 128$ and $64 \times 64$ cuts inference time by 67.73% and 91.73%, respectively, while retaining over 98.63% and 94.57% of the original AUROC (Table 7). This trade-off underscores the flexibility of our approach in balancing accuracy and efficiency according to application needs. Future work may further boost efficiency via diffusion model compression [91, 92, 93] or efficient architecture design [94, 95], highlighting NSG-VD's potential for practical deployment as video generation and detection requirements advance.

Table 7: Comparisons with baselines in terms of Inference time and performance, where we train all models with $10,000$ real and generated videos from Kinetics-400 and SEINE, respectively.

| Method | Recall(%) ↑ | Accuracy(%) ↑ | F1(%) ↑ | AUROC(%) ↑ | Infer. Time (s) ↓ |
|---|---|---|---|---|---|
| DeMamba | 71.25 | 84.54 | 80.87 | 94.42 | 0.0311 |
| NPR | 59.31 | 77.95 | 71.33 | 89.92 | **0.0036** |
| TALL | 61.47 | 80.20 | 74.05 | 95.14 | 0.0044 |
| STIL | 42.35 | 71.08 | 55.81 | 94.94 | 0.0108 |
| NSG-VD (64x64) | 78.29 | 83.26 | 81.99 | 90.27 | 0.0298 |
| NSG-VD (128x128) | 84.93 | **86.99** | 86.25 | 94.15 | 0.1163 |
| NSG-VD (256x256) | **97.13** | 86.05 | **87.45** | **95.46** | 0.3605 |

### E.2 Numerical Stability of NSG

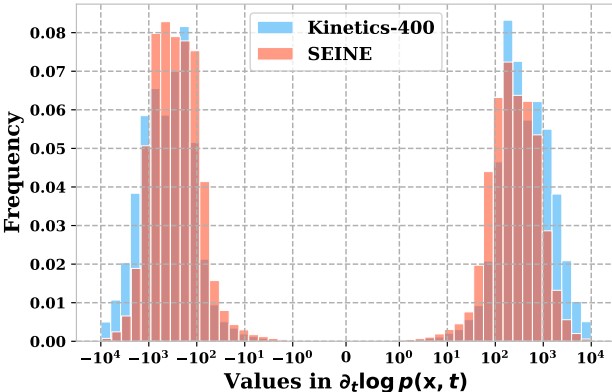

Figure 5: Distribution of the values of temporal derivatives $\partial_t \log p(\mathbf{x}, t)$ in the NSG statistic across $10,000$ real and generated videos from Kinetics-400 and SEINE, respectively.

To ensure the numerical stability of the NSG statistic with the temporal derivatives $\partial_t \log p(\mathbf{x}, t)$ in its denominator, we examine the distribution of values in $\partial_t \log p(\mathbf{x}, t)$ across $10,000$ real and generated videos from Kinetics-400 and SEINE, respectively. From Figure 5, nearly all values lie outside the critical near-zero range $[-0.1, 0.1]$. This indicates that almost no value in $\partial_t \log p(\mathbf{x}, t)$ approaches zero in practice, effectively mitigating instability risks from division by vanishingly small values.

The observed distribution aligns with the physical intuition that temporal density changes in real or synthetic videos are inherently non-stationary, resulting in measurable temporal derivatives. Additionally, the regularization term $\lambda > 0$ in the NSG denominator (Eqn. 6) further safeguards against edge cases where $\partial_t \log p(\mathbf{x}, t)$ might marginally approach zero. These design choices collectively ensure robust numerical stability for NSG across diverse video distributions.

### E.3 Impact of MMD for NSG-VD

To validate the inherent superiority of the NSG statistic independent of the training objective, we compare our framework trained with both Maximum Mean Discrepancy (NSG-VD) and standard binary cross-entropy loss (NSG-BCE) against baselines using BCE. From Table 8, NSG-BCE achieves superior performance across all metrics compared to state-of-the-art baselines, even when adopting a conventional training paradigm. For example, it achieves $77.67\%$ average Recall and $82.70\%$ F1-score, significantly outperforming Demamba by $6.42\% \uparrow$ in Recall and $1.83\% \uparrow$ in F1-score, and TALL by $16.20\% \uparrow$ in Recall and $8.65\% \uparrow$ in F1-score. This demonstrates that the NSG statistic's ability to capture spatiotemporal dynamics remains effective regardless of the training objective.

Notably, NSG-BCE excels in challenging scenarios where other methods struggle. For instance, it achieves $64.29\%$ Recall on Sora (vs. $42.86\%$ for Demamba) and $63.60\%$ Recall on WildScrape (vs. $48.00\%$ for Demamba), highlighting its ability to generalize beyond superficial artifacts. The performance gap widens further in NSG-VD ($97.13\%$ Recall), where MMD explicitly models distributional shifts by the NSG feature in a producing kernel Hilbert space and enables more precise separation between real and synthetic videos. These results confirm that the NSG statistic's physics-driven design captures fundamental spatiotemporal dynamics, providing an intrinsic advantage over conventional features regardless of the training strategy.

Table 8: Impact of MMD in our NSG-VD across 10 generated paradigms (%), where we train all models with 10,000 real and generated videos from Kinetics-400 and SEINE, respectively.

| Method | Metric | Model Scope | Morph Studio | Moon Valley | HotShot | Show1 | Gen2 | Crafter | Lavie | Sora | Wild Scrape | Avg. |
|---|---|---|---|---|---|---|---|---|---|---|---|---|
| DeMamba | Recall | 47.40 | 87.80 | 88.20 | 77.40 | 75.00 | 85.60 | 91.60 | 68.60 | 42.86 | 48.00 | 71.25 |
| | Accuracy | 72.80 | 93.00 | 93.20 | 87.80 | 86.60 | 91.90 | 94.90 | 83.40 | 68.75 | 73.10 | 84.54 |
| | F1 | 63.54 | 92.62 | 92.84 | 86.38 | 84.84 | 91.36 | 94.73 | 80.52 | 57.83 | 64.09 | 80.87 |
| | AUROC | 88.29 | 98.39 | 98.76 | 97.84 | 96.89 | 98.76 | 99.35 | 96.87 | 80.93 | 88.11 | 94.42 |
| NPR | Recall | 46.40 | 76.40 | 69.80 | 63.80 | 56.00 | 75.00 | 83.80 | 58.80 | 35.71 | 27.40 | 59.31 |
| | Accuracy | 71.40 | 86.40 | 83.10 | 80.10 | 76.20 | 85.70 | 90.10 | 77.60 | 66.96 | 61.90 | 77.95 |
| | F1 | 61.87 | 84.89 | 80.51 | 76.22 | 70.18 | 83.99 | 89.43 | 72.41 | 51.95 | 41.83 | 71.33 |
| | AUROC | 85.73 | 96.01 | 93.79 | 91.44 | 89.96 | 95.13 | 96.87 | 89.46 | 84.15 | 76.66 | 89.92 |
| TALL | Recall | 58.60 | 75.00 | 79.40 | 60.20 | 62.00 | 77.80 | 88.20 | 43.80 | 33.93 | 35.80 | 61.47 |
| | Accuracy | 78.80 | 87.00 | 89.20 | 79.60 | 80.50 | 88.40 | 93.60 | 71.40 | 66.07 | 67.40 | 80.20 |
| | F1 | 73.43 | 85.23 | 88.03 | 74.69 | 76.07 | 87.02 | 93.23 | 60.50 | 50.00 | 52.34 | 74.05 |
| | AUROC | 97.10 | 98.12 | 98.63 | 96.37 | 96.45 | 97.76 | 99.38 | 94.80 | 83.35 | 89.45 | 95.14 |
| STIL | Recall | 28.60 | 57.40 | 78.40 | 46.80 | 18.80 | 66.40 | 69.00 | 24.80 | 14.29 | 19.00 | 42.35 |
| | Accuracy | 64.20 | 78.60 | 89.10 | 73.30 | 59.30 | 83.10 | 84.40 | 62.30 | 57.14 | 59.40 | 71.08 |
| | F1 | 44.41 | 72.84 | 87.79 | 63.67 | 31.60 | 79.71 | 81.56 | 39.68 | 25.00 | 31.88 | 55.81 |
| | AUROC | 95.53 | 97.91 | 99.40 | 96.49 | 92.79 | 98.06 | 98.86 | 91.00 | 92.79 | 86.58 | 94.94 |
| NSG-BCE (Ours) | Recall | 53.40 | 96.40 | 94.80 | 90.60 | 77.60 | 79.40 | 83.20 | 73.40 | 64.29 | 63.60 | 77.67 |
| | Accuracy | 72.70 | 94.20 | 93.40 | 91.30 | 84.80 | 85.70 | 87.60 | 82.70 | 74.11 | 77.80 | 84.43 |
| | F1 | 66.17 | 94.32 | 93.49 | 91.24 | 83.62 | 84.74 | 87.03 | 80.93 | 71.29 | 74.13 | 82.70 |
| | AUROC | 84.67 | 98.79 | 97.77 | 96.90 | 92.69 | 93.00 | 93.86 | 91.32 | 83.58 | 87.99 | 92.06 |
| NSG-VD (Ours) | Recall | 91.67 | 100.00 | 100.00 | 100.00 | 100.00 | 98.33 | 100.00 | 97.50 | 94.64 | 89.17 | **97.13** |
| | Accuracy | 82.50 | 88.33 | 89.58 | 84.58 | 86.25 | 87.08 | 86.67 | 87.92 | 89.29 | 78.33 | **86.05** |
| | F1 | 83.97 | 89.55 | 90.57 | 86.64 | 87.91 | 88.39 | 88.24 | 88.97 | 89.83 | 80.45 | **87.45** |
| | AUROC | 90.67 | 97.62 | 98.38 | 95.88 | 96.69 | 97.87 | 97.64 | 95.09 | 96.14 | 88.65 | **95.46** |

## E.4 Impact of Size of Reference Set for NSG-VD

We investigate the impact of reference set size by evaluating subsets containing between 10 and 500 samples, with other settings remaining consistent with Table 1. Performance is assessed using comprehensive criteria, including AUROC, Accuracy, F1 Score, and Recall. Intuitively, a larger reference set enables more accurate estimation of the underlying distribution of real videos, thereby supporting more stable and reliable detection. In contrast, smaller reference sets may introduce substantial sampling and estimation biases. As shown in Figure 6, our NSG-VD demonstrates consistently robust performance across varying reference set sizes, with the exception of extreme cases involving very limited samples (*e.g.*, $n = 10$). Consequently, we set $n = 100$ for all experiments.

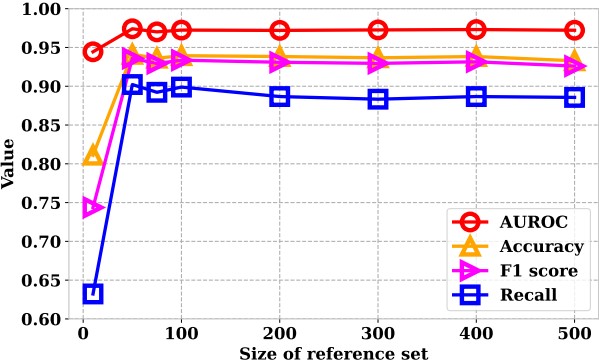

Figure 6: Impact of reference set size for NSG-VD, where we train all models with 10,000 real and generated videos from Kinetics-400 and Pika, respectively.

### E.5 Impact of Diversity of Real Videos in the Reference Set

We conduct additional ablation studies on *real-domain mixed* reference sets, revealing a key strength of NSG-VD: strong generalization to unseen generated video domains when most real test samples are covered by the reference distribution. Specifically, we train on Kinetics-400 (real) and SEINE (generated) videos, and test on MSR-VTT (real) and 10 generated videos using reference sets with varying ratios of MSR-VTT and Kinetics-400. From Table 9, even a small proportion (3 : 7) yields satisfactory performance (84.19% of Accuray, 81.12% of F1-Score) compared with baselines, which quickly saturates. This confirms that NSG-VD needs only modest in-domain real coverage, while the fake side can remain highly heterogeneous. These results will be included in our revision.

Table 9: Performance under different domain coverage ratios between MSR-VTT and Kinetics-400.

| Domain Coverage (MSR-VTT : Kinetics-400) | 0:10 (Low) | 3:7 (Medium) | 5:5 (Balanced) | 7:3 (High) | 10:0 (Full) | DeMamba | TALL |
|---|---|---|---|---|---|---|---|
| Average Accuracy (%) | 77.82 | 84.19 | 85.57 | **87.06** | 86.05 | 84.21 | 80.20 |
| Average F1-Score (%) | 75.68 | 81.12 | 83.36 | 85.41 | **87.45** | 80.87 | 74.05 |

### E.6 Discussions on Assumption of the Divergence Term

We assume $\nabla_{\mathbf{x}} \cdot \mathbf{v}$ is subdominant in smoothly varying video distributions for three reasons: **First**, its direct estimation is ill-posed in high-dimensional video data. Solving $\partial_t \mathbf{x} = \mathbf{v}(\mathbf{x}, t)$ is an underdetermined inverse problem, and video noise (*e.g.*, blur or compression) further amplifies estimation errors, making explicit divergence computation unstable and infeasible [32, 29]. **Second**, many physical flows approximate incompressibility ($\nabla_{\mathbf{x}} \cdot \mathbf{v} \approx 0$), a simplification grounded in fluid dynamics [29] and quantum mechanics [55] that preserves physical interpretability. **Third**, our NSG remains robust even if $\nabla_{\mathbf{x}} \cdot \mathbf{v} \neq 0$, as it captures cumulative spatiotemporal inconsistencies across all terms in Eqn. (3). Experiments confirm the resilience of NSG-VD to deviations from this assumption.

## F  Limitations and Future Directions

While our proposed NSG-VD method demonstrates strong performance across diverse AI-generated video detection scenarios, several limitations and opportunities for future work remain:

**Limitations.** First, the current formulation of the NSG statistic relies on simplified physical assumptions (*e.g.*, the incompressible flow approximation in continuity equations), which may fail to capture highly dynamic or discontinuous motion patterns in complex real-world scenarios. Second, the effectiveness of NSG-VD critically depends on the quality of pre-trained diffusion models used for score estimation; domain shifts or limited training data may degrade the reliability of estimated NSG features. Third, while NSG-VD achieves competitive performance, its reliance on diffusion models introduces computational overhead, making it less suitable for large-scale real-time detection tasks. Lastly, while our deep kernel design improves detection performance, its architecture could be further optimized to better adapt to heterogeneous spatiotemporal patterns.

**Future Directions.** To address these limitations, future work could explore more sophisticated physical models that account for compressible flows or discontinuous motion dynamics [54], enhancing the NSG statistic's adaptability to complex scenarios. Additionally, developing effective domain-specific fine-tuning strategies [96, 97, 98, 99] could improve the reliability of score estimation under distribution shifts. For real-time deployment, investigating lightweight diffusion model compression techniques (*e.g.*, pruning [91, 100], quantization [92, 93]) would reduce computational costs. Finally, advancing the design of the deep kernel network—such as incorporating attention mechanisms [101] or hierarchical feature fusion [89]—could further optimize the MMD-based detection framework, enabling better performance for AI-generated video detection.

# G  Broader Impacts

The development of AI-generated video detection methods like NSG-VD has significant societal, ethical, and technical implications. Our work contributes to mitigating the risks of malicious deepfake content, such as misinformation, identity theft, and political manipulation, by enabling more reliable verification of video authenticity. By leveraging physics-informed principles, NSG-VD provides a reliable framework for detecting synthetic videos that may otherwise evade traditional artifact-based detection methods. This could strengthen trust in digital media, support content moderation efforts, and aid legal or journalistic investigations involving video evidence.

This research aligns with broader efforts to establish trustworthy multimedia ecosystems. By bridging physics principles with machine learning, NSG-VD advances interpretable detection mechanisms—a critical step toward auditing AI-generated content while fostering public awareness of synthetic media risks. We encourage interdisciplinary collaboration among researchers, ethicists, and legislators to ensure such technologies serve as safeguards rather than instruments of control.

# H  Visualizations

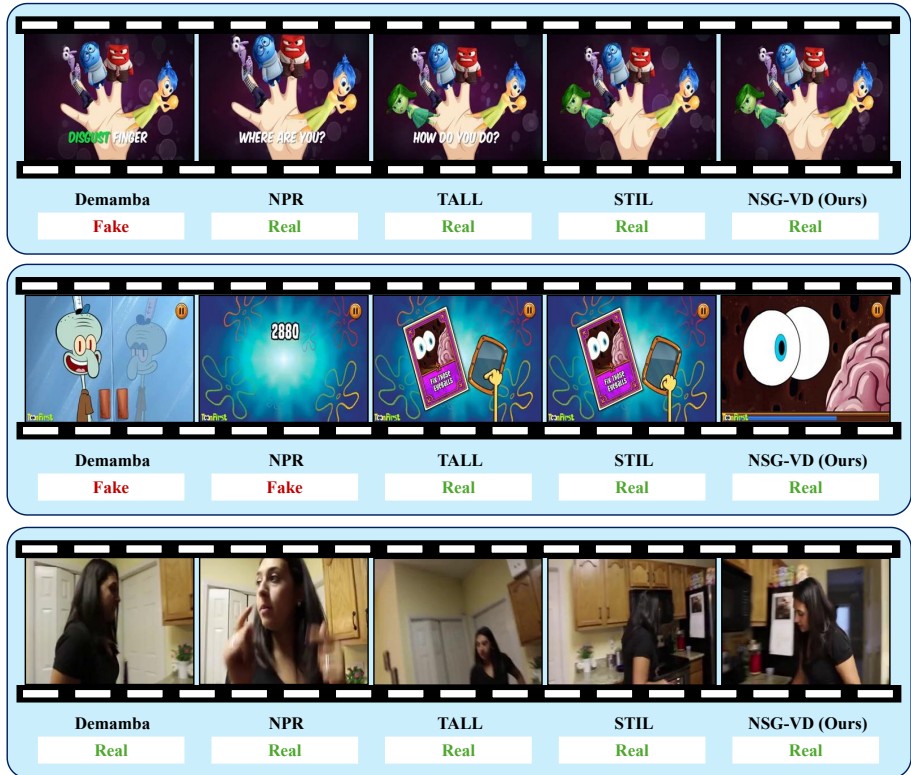

Figure 7: Results of the detection on *real* videos from the MSR-VTT dataset.

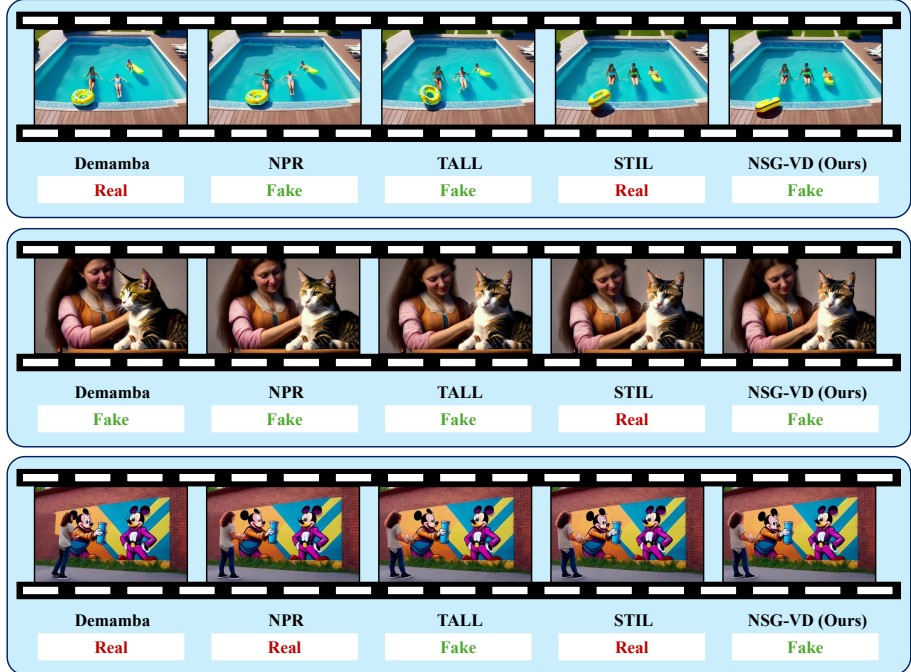

Figure 8: Results of the detection on *generated* videos from the Crafter dataset.

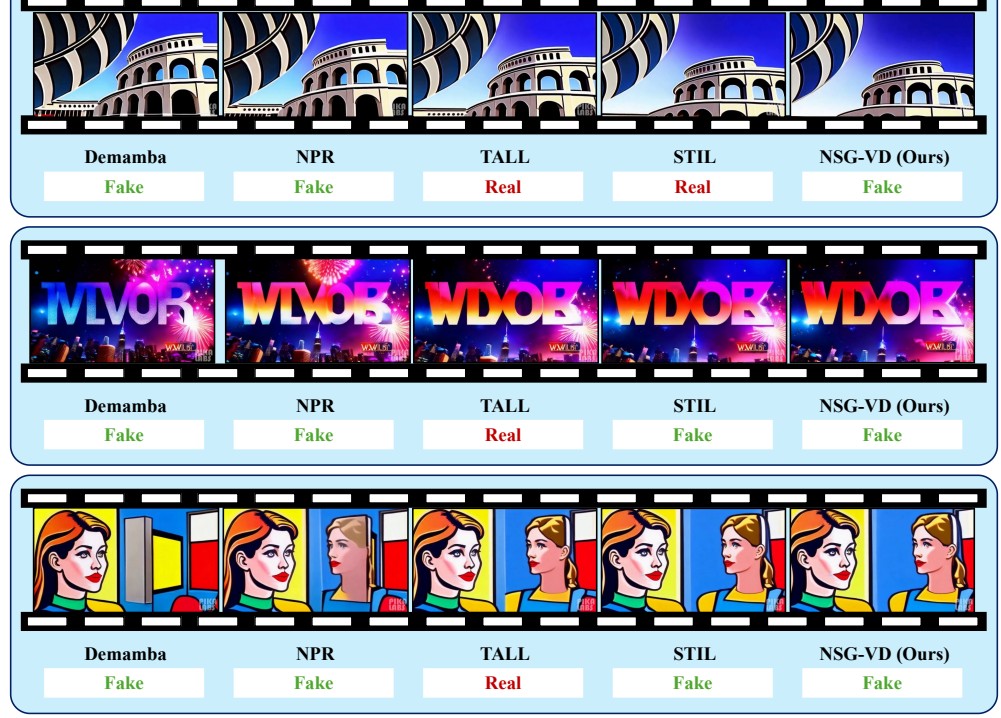

Figure 9: Results of the detection on *generated* videos from the Gen2 dataset.

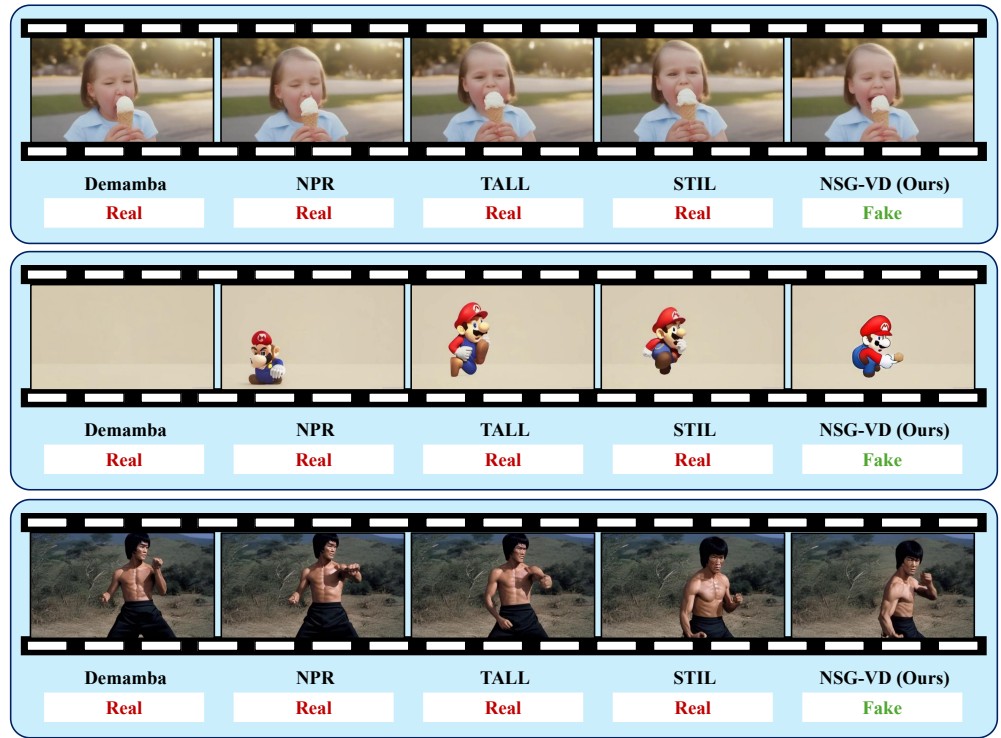

Figure 10: Results of the detection on *generated* videos from the HotShot dataset.

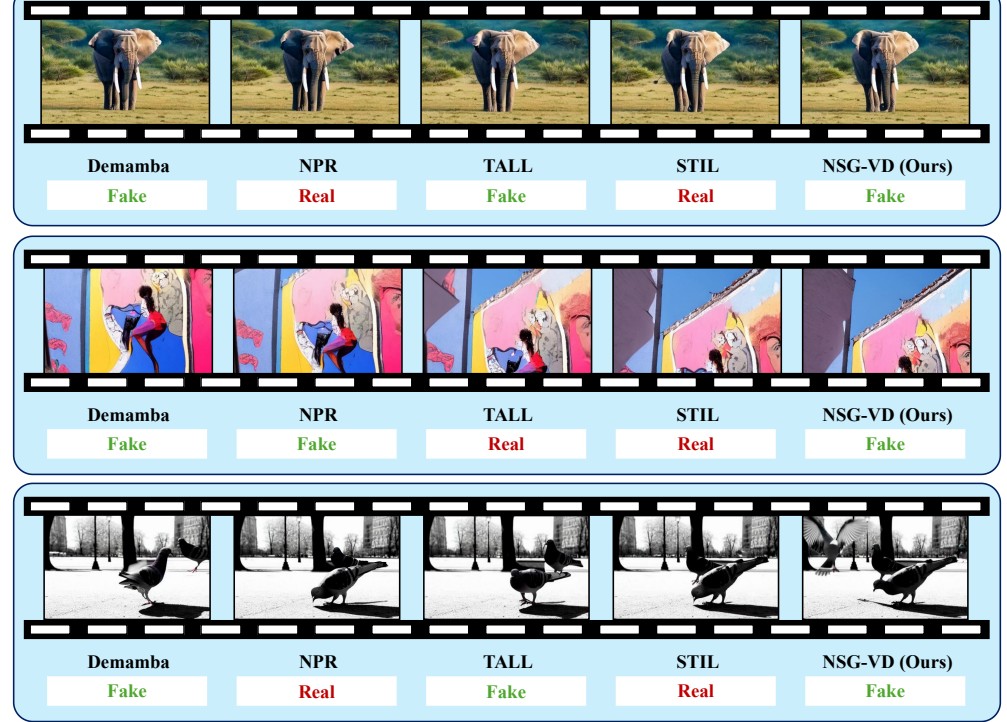

Figure 11: Results of the detection on *generated* videos from the Lavie dataset.

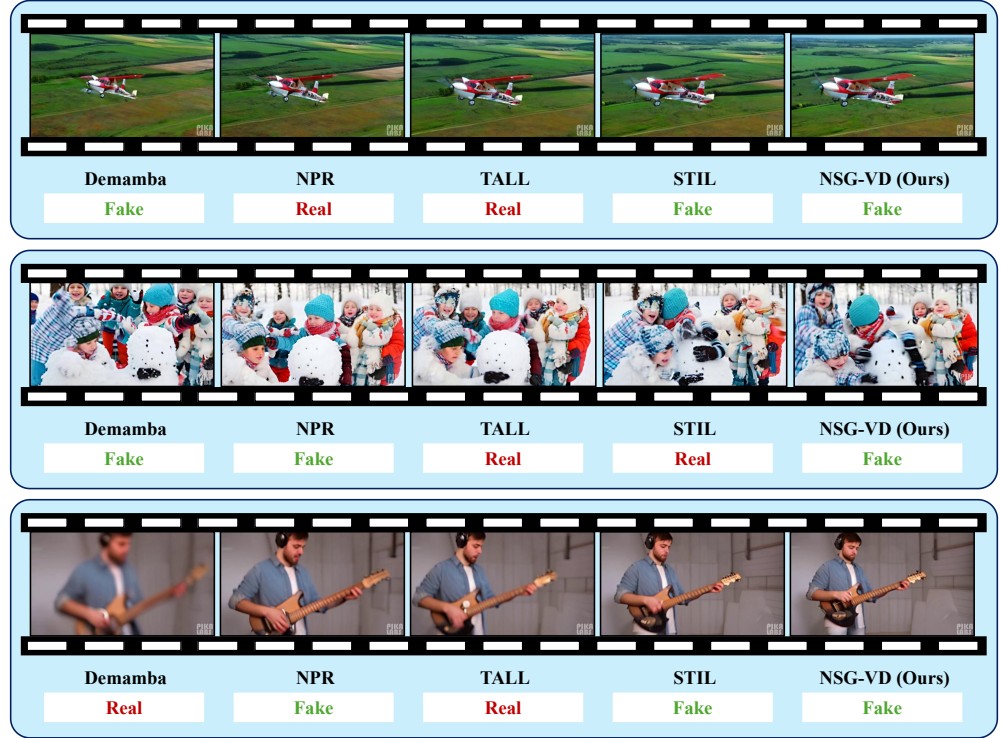

Figure 12: Results of the detection on *generated* videos from the ModelScope dataset.

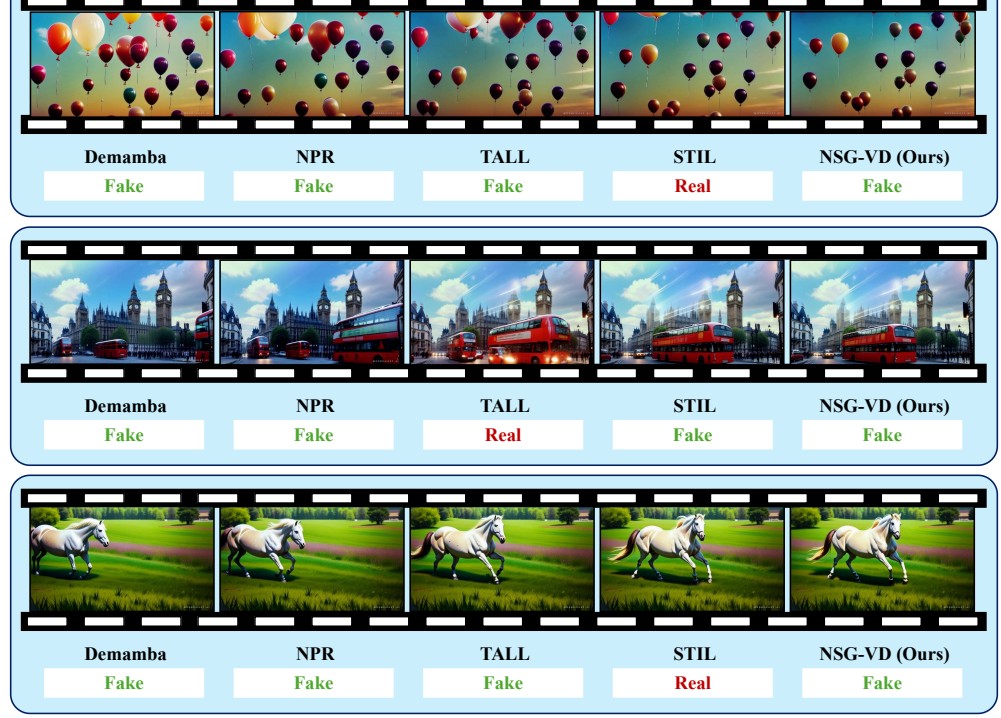

Figure 13: Results of the detection on *generated* videos from the MoonValley dataset.

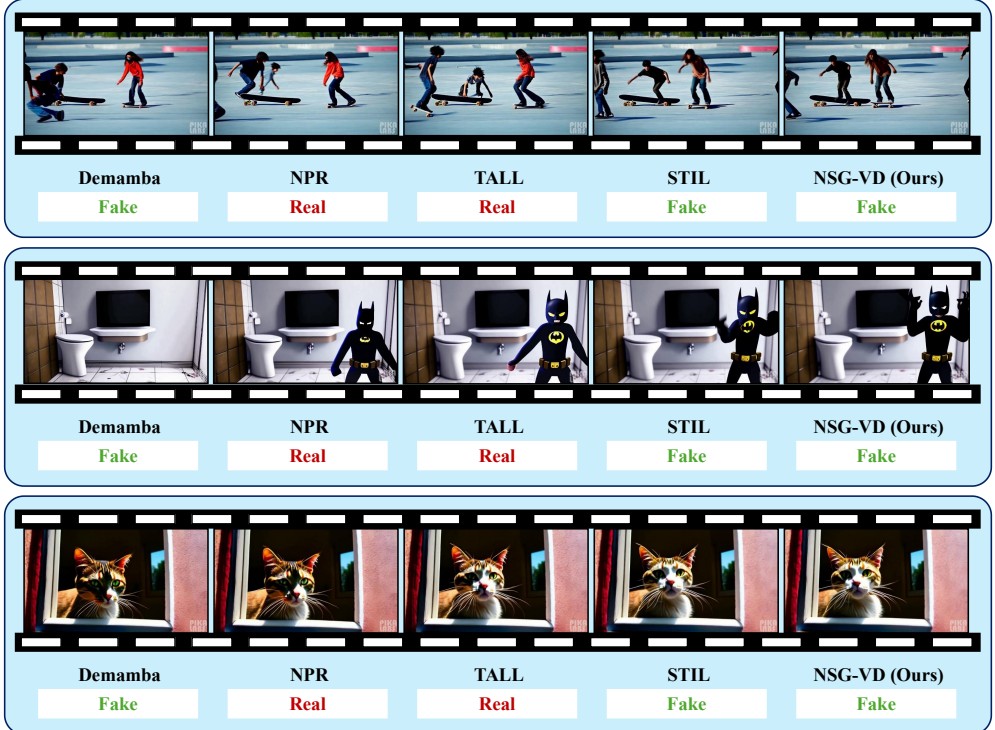

Figure 14: Results of the detection on *generated* videos from the MorphStudio dataset.

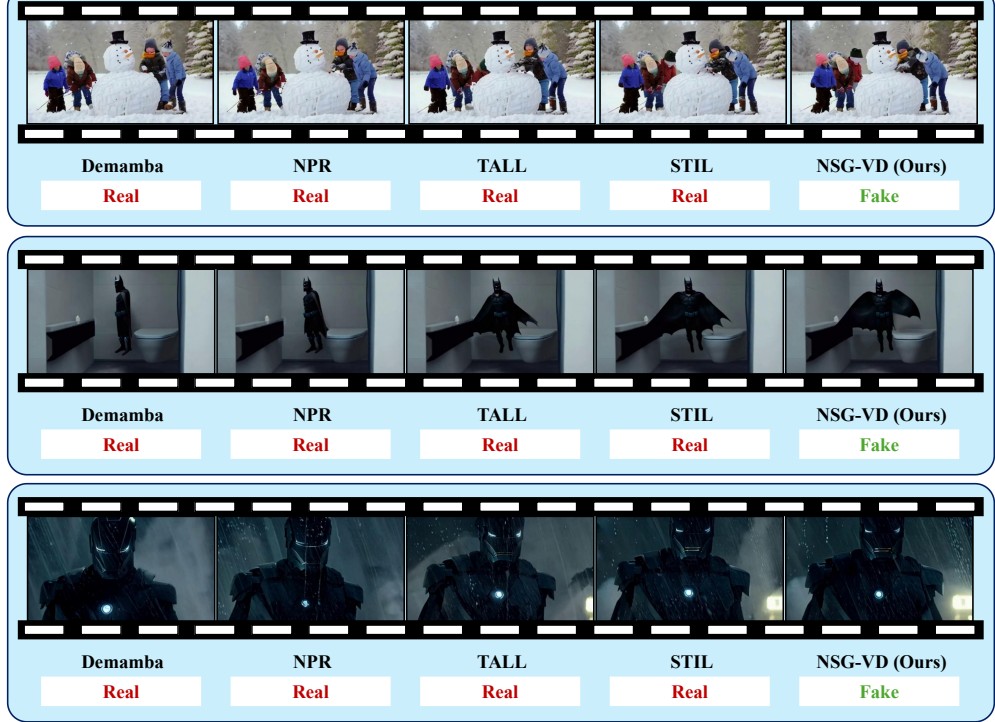

Figure 15: Results of the detection on *generated* videos from the Show1 dataset.

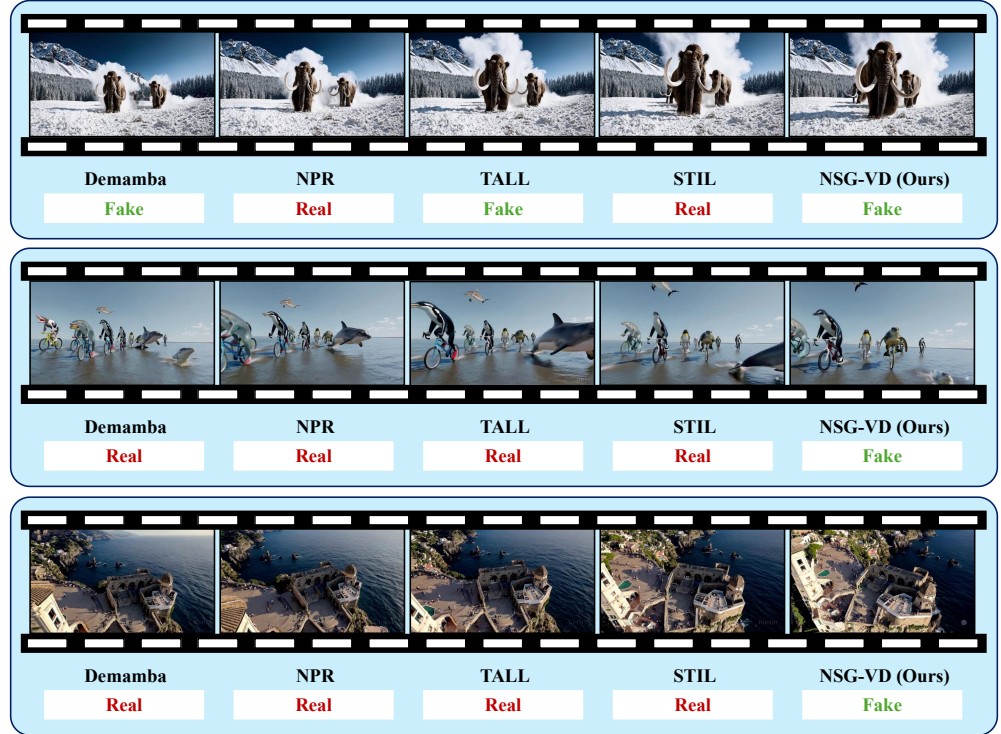

Figure 16: Results of the detection on *generated* videos from the Sora dataset.

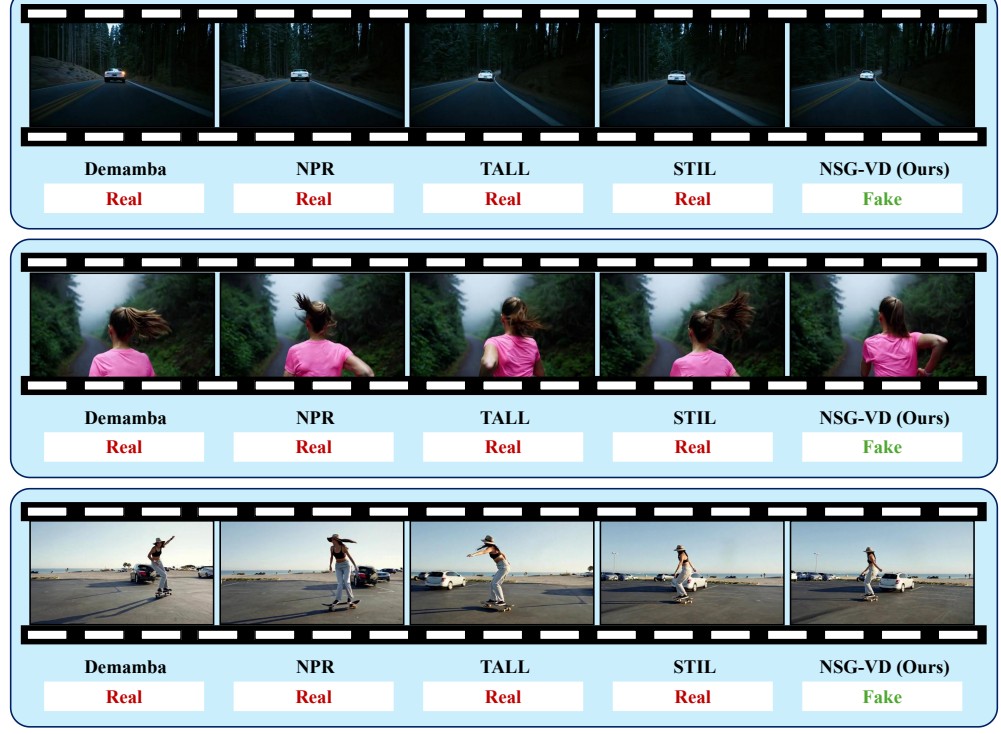

Figure 17: Results of the detection on *generated* videos from the Seaweed dataset.

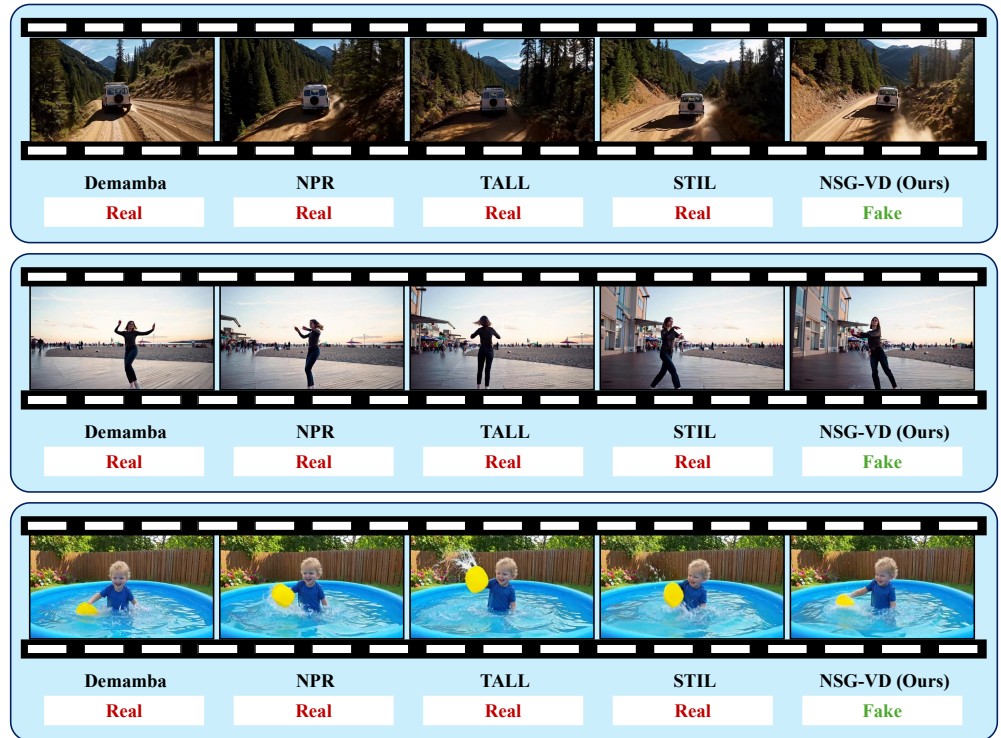

Figure 18: Results of the detection on *generated* videos from the Seaweed dataset.

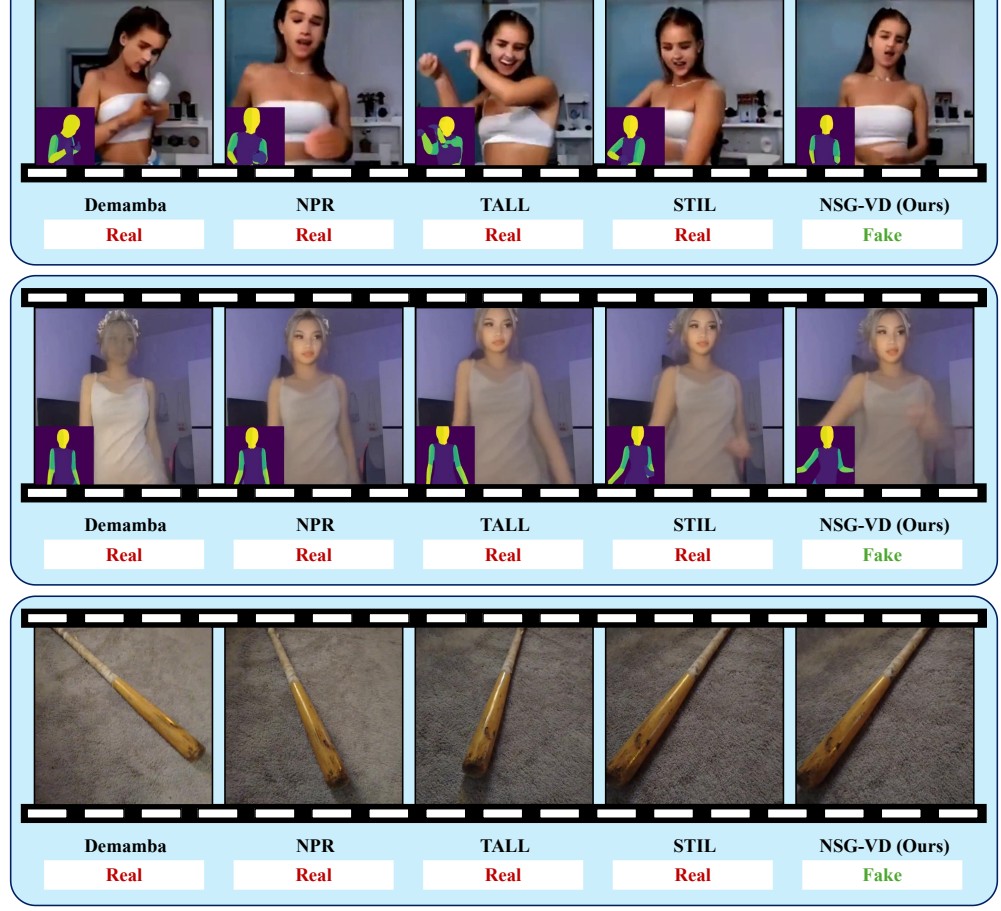

Figure 19: Results of the detection on *generated* videos from the WildScrape dataset.

To further demonstrate the excellent performance of our NSG-VD, we present visual detection results on both real and generated videos across all 10 datasets. As illustrated in Figures 7-19, both the baselines and NSG-VD demonstrate satisfactory detection on real video samples. For generated videos, the existing baselines achieve reasonable performance on early generation models (e.g., Crafter, Gen2, and MoonValley), but exhibit significant performance degradation when applied to more advanced generative models (e.g., Show1, Sora, and WildScrape). In contrast, NSG-VD consistently achieves strong detection performance across all generation levels.

On this basis, we consider the recently proposed Seaweed [102] method (as shown in Figures 17, 18), which generates highly realistic long-form videos. All four baselines exhibit near-complete failure on this dataset, whereas NSG-VD continues to deliver effective detection performance.

