# OpenReview forum: "Physics-Driven Spatiotemporal Modeling for AI-Generated Video Detection"
_NeurIPS.cc/2025/Conference — NeurIPS 2025 spotlight_

### Official Review · Reviewer_iZEA · 2025-06-26

**Clarity:** 2
**Significance:** 3
**Originality:** 3
**Rating:** 4
**Confidence:** 3

**Summary:**

The paper proposes a physics-driven AI-generated video detection paradigm. First, it designs Normalized Spatiotemporal Gradient (NSG), a statistic based on probability flow conservation principles. Then, it proposes an NSG estimator by spatial gradients approximation and motion-aware temporal dynamics modeling using pre-trained diffusion models. Last, it proposes an NSG-based video detection model (NSG-VD) with theoretical and empirical justifications.

**Questions:**

1. The estimation of spatial gradients and temporal derivations heavily relies on the capabilities of the pretrained diffusion model. The potential impact of the pretrained model's performance ceiling and biases on the final effectiveness of NSG-VD needs discussion.
2. The impact of the reference set's size and distribution on detection performance remains unclear. Additional experiments are needed to assess the robustness of NSG-VD to different reference set configurations.
3. A comparative analysis between the model's performance and human-level discrimination accuracy for AI-generated versus natural videos would provide further validation of the method's effectiveness.

**Ethical Concerns:**

["NO or VERY MINOR ethics concerns only"]

**Final Justification:**

Thanks for the detailed response. My concerns about the ablation study, the distinction between natural and AI-generated videos, and the potential impact of the pretrained model have been addressed. Hope the authors can integrages relevant discussion into the final version. I will keep my rate as "Borderline Accept".

**Limitations:**

Yes.

**Quality:**

3

**Strengths And Weaknesses:**

Strengths:
1. The paper is well-motivated, as AI-generated videos become increasingly realistic and it is critically urgent to develop effective video detection methods for preserving societal trust in digital media.
2. The theoretical analysis for NSG-VD further guarantees the effectiveness of NSG-VD.
3. The comparisons are comprehensive, evaluating the method on both standard and data-imbalanced scenarios.

Weaknesses:
1. The ablation study is not sufficient. The paper analyzes the impact of spatial gradients and temporal derivatives for NSG-VD, but some key components (e.g., the reference set and diffusion model) lack systematic analysis.
2. The distinction between natural and AI-generated videos is not illustrated clearly or intuitively.

---

> ### Author Rebuttal · Authors · 2025-07-29
>
> We thank the reviewer for the encouraging comments and detailed suggestions. Responses are below:
>
> >Q1. The ablation study is not sufficient. The paper analyzes the impact of spatial gradients and temporal derivatives for NSG-VD, but some key components (e.g., the reference set) lack systematic analysis.
>
> **A1.** We sincerely thank the reviewer for this valuable suggestion regarding the reference set analysis. We conduct additional ablation studies on **real‑domain mixed reference sets**, revealing a key strength of NSG-VD: **strong generalization to unseen generated video domains** when most real test samples are covered by the reference distribution.
>
> Specifically, we train our detetor on Kinetics-400 (real) and SEINE (generated) videos, and test on MSR-VTT (real) and other 10 generated videos using **reference sets with varying ratios of MSR-VTT and Kinetics-400** (0:10, 3:7, 5:5, 7:3, 10:0). As shown in Table I, **even a small proportion (3:7)** of in-domain real samples **yields satisfactory performance** (84.19% of Accuray and 81.12% of F1-Score and then performance quickly saturates) compared with baselines. This confirms that **NSG‑VD needs only modest in-domain real coverage** to establish the detector, while the fake side can remain highly heterogeneous. We will include these results in our revised manuscript.
>
> Table I. Effect of reference set composition, trained on Kinetics-400 (real) and SEINE (generated) videos, and tested on MSR-VTT (real) and the other 10 generated videos (%).
> | **MSR-VTT: Kinetics-400** **(Domain Coverage)**|0:10 (Low)|3:7 (Medium)|5:5 (Balanced)|7:3 (High)|10:0 (Full)|DeMamba (Baseline)|TALL (Baseline)
> |-|-|-|-|-|-|-|-|
> |**Average Accuray**|77.82|84.19|85.57|87.06|86.05|84.21|80.20|
> |**Average F1-Score**|75.68|81.12|83.36|85.41|87.45|80.87|74.05|
>
> >Q2. The key components diffusion model lack systematic analysis. The estimation of spatial gradients and temporal derivations heavily relies on the capabilities of the pretrained diffusion model. The potential impact of the pretrained model's performance ceiling and biases on the final effectiveness of NSG-VD needs discussion.
>
> **A2**. We thank the reviewer for this valuable suggestion regarding the pre-trained diffusion models. We discuss this with three key insights: 1) **framework agnosticism to score function sources**, 2) **diffusion model selection criteria**, and 3) **empirical consistency** across different pretrained diffusion models.
>
> * **Architecture‑agnostic framework**: Our NSG‑VD pipeline requires only access to the score function $\nabla_{\mathbf{x}}\log p(\mathbf{x},t)$. Once this gradient is obtained, regardless of the underlying diffusion architecture, we compute the NSG features $\mathbf{g}(\mathbf{x},t)$ (Eqn. 4) and train a **lightweight deep‑kernel** MMD detector on these features. This design decouples the detection mechanism from any specific diffusion implementation.
> * **Diffusion model selection criteria**: We prioritize **unconditional, pixel-space diffusion** models because they directly estimate $\nabla_{\mathbf{x}} \log p(\mathbf{x}, t)$ in the pixel domain—critical for capturing spatiotemporal dynamics in videos. In contrast, recent latent-space models (e.g., Stable Diffusion [r3], EDMv2 [r4]) perform diffusion in compressed latents, requiring additional decoding and Jacobian projections to recover pixel-space scores, introducing additional complexity and potential approximation errors. We view this as compelling future work.
> * **Empirical validation across diffusion models**: In our main experiments, we use an ImageNet‑pretrained Guided Diffusion model [r1] and achieve strong detection performance (Tables 1–2). To further validate generalizability, we supplement additional experiments with an Improved DDPM-based pre-trained diffusion model [r2] on the same dataset, obtaining **consistent performance** of 1.79% average Accuracy and 5.46% F1-score (Table II). These results confirm that NSG‑VD’s efficacy stems from the **universal gradient‑estimation capability** of diffusion models, not model‑specific design and that our current pixel‑space diffusion model is **sufficient to capture the spatiotemporal dynamics needed** for robust detection.
>
> We will include these findings in the revised manuscript to highlight the framework’s generalizability and its current practical scope.
>
> Table II. Effect of different diffusion Models, with the same setting as Table 2 (%).
> Table II. Effect of different diffusion Models, with the same setting as Table 2 (%).
> |Method|Accuracy|F1‑Score|
> |-|-|-|
> |DeMamba (Baseline)|84.21|80.12|
> |NSGVD (Improved DDPM, 64x64)| 84.26|81.99|
> |NSGVD (Guided Diffusion, 256x256)|86.05|87.45|
>
> [r1] Diffusion Models Beat GANs on Image Synthesis.NeurIPS, 2021.
>
> [r2] Improved Denoising Diffusion Probabilistic Models. ICML, 2021.
>
> [r3] Scaling Rectified Flow Transformers for High-Resolution Image Synthesis. ICML, 2024.
>
> [r4] Guiding a Diffusion Model with a Bad Version of Itself. NeurIPS, 2024.
>
>
> >Q3. The distinction between natural and AI-generated videos is not illustrated clearly or intuitively.
>
> **A3**. We appreciate the reviewer's insightful comment. Natural videos typically exhibit **rich, coherent, and physically plausible details**. In contrast, current AI-generated videos, particularly those from diffusion models, often display subtle artifacts in high-frequency regions, such as **oversmoothing, localized blurring, or physically inconsistent lighting and shadows, as well as temporal incoherence**. These discrepancies induce implicit distributional shifts in the feature space, enabling their discrimination. In the revised manuscript, we will further elaborate on the core discriminative features underpinning our method and their representational differences, complemented by visualizations of the underlying distributional divergence.
>
> >Q4. The impact of the reference set's size and distribution on detection performance remains unclear. Additional experiments are needed to assess the robustness of NSG-VD to different reference set configurations.
>
> **A4**. We thank the reviewer for this suggestion. We have already analyzed the impact of the reference set’s distribution in our response to Q1-A1. In Appendix E.4 in the paper, we evaluated NSG‑VD using reference sets of varying sizes (Table III, below). From the table, NSG‑VD maintains consistently high detection accuracy and F1‑Score across varying reference set sizes.
>
> Table III. Effect of reference set size (%).
> |size|10|50|75|100|200|300|400|500|
> |-|-|-|-|-|-|-|-|-|
> |Recall|63.22|90.22|89.22|89.89|88.67|88.33|88.67|88.56|
> |AUROC|94.44|97.41|97.01|97.24|97.20|97.26|97.31|97.22|
> |Accuracy|81.11|94.11|93.61|93.94|93.83|93.67|93.83|93.28|
> |F1|74.37|93.59|92.93|93.36|93.09|92.95|93.15|92.60|
>
> >Q5. A comparative analysis between the model's performance and human-level discrimination accuracy for AI-generated versus natural videos would provide further validation of the method's effectiveness.
>
> **A5**. We thank the reviewer's comments. As discussed in our work, early generative models produced visible artifacts such as **distortions and temporal inconsistencies**, which are easily detectable by both humans and algorithms. However, with the rapid advancement of SOTA generative models, modern AI-generated videos attain near-photorealistic fidelity, where residual **artifacts are subtle and often imperceptible to the human eye**. In such cases, an AI detection model leverages fine-grained, frame-level analysis to detect nuanced deviations beyond human perceptual limits. Furthermore, human observers can outperform algorithms in cases involving semantic or commonsense inconsistencies, even when no visual artifacts are present. We acknowledge the importance of this perspective and will include a human-AI comparative evaluation in the revised manuscript.
>
> ---
> **We hope our response has clarified the confusion and addressed your concerns. We would greatly appreciate it if you could kindly reconsider your score. Thank you.**

---

> > ### Comment · Reviewer_iZEA · 2025-08-05
> >
> > Thanks for the detailed response. My concerns about the ablation study, the distinction between natural and AI-generated videos, and the potential impact of the pretrained model have been addressed. Hope the authors can integrages relevant discussion into the final version.

---

### Official Review · Reviewer_1Wda · 2025-07-02

**Clarity:** 3
**Significance:** 4
**Originality:** 4
**Rating:** 5
**Confidence:** 3

**Summary:**

The paper addresses the challenge of detecting realistic AI-generated videos. The authors argue that existing methods that often rely on specific visual artifacts are becoming less effective. They propose a novel, physics-driven paradigm that models video evolution based on the principle of probability flow conservation, analogous to fluid dynamics.

The core contribution is a new statistic called the Normalized Spatiotemporal Gradient, which quantifies the ratio between spatial probability gradients and temporal density changes. The goal is to capture subtle violations of physical laws that are common in synthetic videos but absent in real ones. To practically implement this, the authors develop the NSG-based Video Detection method. This method uses pre-trained diffusion models to estimate the spatial gradients and approximates temporal derivatives using a motion-aware model based on the brightness constancy assumption. The final detection is performed by calculating the MMD between the NSG features of a test video and a reference set of real videos. A larger discrepancy suggests the video is AI-generated.

**Questions:**

1. Physical assumptions: Could you elaborate on the failure modes of the incompressible flow and brightness constancy assumptions in the context of modern generative models? Do you observe performance degradation on specific types of ``real'' and generated videos (e.g., those with fantasy elements, non-rigid object transformations, or explosive visual effects) where these physical approximations might be less valid?

2. Diffusion model: How sensitive is NSG-VD's performance to the choice of the pre-trained diffusion model? Would using a different or more powerful model (like Stable Diffusion) as the score estimator further improve results, or is the current model sufficient to capture the necessary dynamics?

3. Failure cases: The method shows impressive average performance. Could you provide some analysis of the most common failure cases? Understanding when NSG-VD fails (i.e., misclassifying a real video as fake, or vice-versa) would offer valuable insights into its limitations and areas for future improvement.

4. MMD reference set: Your analysis shows that performance stabilizes with a reference set size greater than 10. In a practical setting, how does the _diversity_ of the real videos in the reference set affect detection performance, in addition to its size?

**Ethical Concerns:**

["NO or VERY MINOR ethics concerns only"]

**Final Justification:**

I appreciated the rebuttal to my and the other reviewers' concerns. I remain convinced of the paper's merits and will maintain my positive rating.

**Limitations:**

Yes.

**Paper Formatting Concerns:**

None.

**Quality:**

4

**Strengths And Weaknesses:**

### Strengths

- Originality: The paper introduces a fundamentally new approach to video deepfake detection. Moving from artifact-based detection to a physics-informed model that exploits spatiotemporal dynamics is a significant contribution, especially with the rise of physically implausible but realistic video generators.

- Theoretical foundation: The method is well-grounded in physical principles, like the probability flow conservation. The authors support their approach with a theoretical analysis, which, despite its simplifying assumptions, provides a solid justification for why NSG features should be discriminative for real vs. fake videos.

- Experiments: The method is tested on a large-scale, recent benchmark against multiple baselines. The results demonstrate superior performance not just on average, but specifically on challenging, closed-source models and in difficult data-imbalanced settings, which mimic real-world challenges.

- Parameter Efficiency: The proposed NSG-VD method is efficient in terms of trainable parameters. As shown in Figure 3, it achieves state-of-the-art results by fine-tuning only 0.25 million parameters, a fraction of what competing baselines require for full training.

### Weaknesses

- Computational overhead: A practical limitation is the inference time. The authors honestly report that NSG-VD takes approximately 0.36s per video, which is an order of magnitude slower than the baselines. While the authors position it for non-real-time applications and suggest future work, this currently limits its practical deployment.

- Physical assumptions: The methodology rests on key assumptions, such as the incompressible flow approximation ($\nabla x \cdot v \approx 0$) and the brightness constancy assumption. While standard in other fields, the paper could benefit from a deeper discussion of scenarios where these assumptions might not hold for real and AI-generated videos (e.g., videos with chaotic motion, rapid lighting effects, or object dissolution), and how that would impact performance.

- Pre-trained model: The quality of the NSG features is directly tied to the score estimation from the pre-trained diffusion model. The experiments use a single diffusion model, therefore the robustness of the method to different diffusion model architectures or to a domain mismatch between the model's training data and the test videos is not fully explored.

---

> ### Author Rebuttal · Authors · 2025-07-30
>
> We thank the reviewer for the encouraging comments and detailed suggestions. Responses are below:
> >Q1. "...a **fundamentally new approach** to video deepfake detection. ... exploits spatiotemporal dynamics is a **significant contribution**...", "...**well-grounded** in physical principles...provides a **solid justification**...", " ...demonstrate **superior performance**...", "...is **efficient** in terms of trainable parameters...".
>
> **A1.** We are deeply appreciative of your constructive comments. Your recognition of NSG-VD as a new approach, acknowledgment of our contribution in moving from artifact-based to physics-informed detection, and noting its solid physical grounding, superior performance on large benchmarks motivate us to advance this research.
>
> >Q2. Computational overhead: The authors report that NSG-VD takes approximately 0.36s per video, which is an order of magnitude slower than the baselines. While intended for non-real-time use, this limits practical deployment.
>
> **A2.** We appreciate the reviewer’s concern regarding inference latency. We fully agree that the inference time is important for deployment while our current NSG‑VD prioritizes detection accuracy to validate the physics‑driven paradigm. We will clarify and extend our discussion as follows:
>
> * **Core contribution focus**: Our primary goal is establishing the **physics-driven paradigm's effectiveness** for AI-generated video detection. This requires high-fidelity gradient estimation from diffusion models, e.g., the 256×256 Guided Diffusion model [r1], which prioritizes theoretical validation over speed optimization. This achieves SOTA detection with higher latency.
> * **Acceleration via resolution scaling**: To explore speedups, we replace the 256×256 diffusion model with a 64×64 Improved DDPM [r2] and 128x128 Guided Diffusion recompute NSG features at lower resolution. From Table I, inference time reduces by over **10×** (0.36 s→0.0298 s) with only modest drops in Accuracy (86.05%→83.20%) and F1‑Score (87.45%→84.90%) at 64×64 resolution, showing a favorable trade-off for near-offline use.
> * **Future work**: As discussed in the manuscript, further acceleration is possible via diffusion model compression using lightweight architectures for score estimation, e.g., pruning [r3], quantization [r4]. These are important directions for future work.
>
> We will include these acceleration results in the revised version.
>
> Table I. Comparisons with baselines on Inference time and performance (Table 2 settings(%)).
> |Method|Accuracy↑|F1‑Score↑|Infer. Time (s)↓|
> |-|-|-|-|
> |DeMamba (Baseline)|84.21|80.12|0.0311|
> |NSGVD (Improved DDPM, 64x64)| 84.26|81.99|0.0298|
> |NSGVD (Guided Diffusion, 128x128)|86.20|86.25|0.1163|
> |NSGVD (Guided Diffusion, 256x256)|86.05|87.45|0.3605|
>
> [r1] Diffusion Models Beat GANs on Image Synthesis.NeurIPS, 2021.
>
> [r2] Improved Denoising Diffusion Probabilistic Models. ICML, 2021.
>
> [r3] Structural pruning for diffusion models. NeurIPS, 2023.
>
> [r4] Ptq4dit: Post-training quantization for diffusion transformers. NeurIPS, 2024.
>
> >Q3. The method relies on incompressible flow and brightness constancy. When these fail in real or AI‑generated videos (e.g., chaotic motion, rapid lighting changes, non‑rigid deformations) and any observed performance drops?
>
> **A3.** We thank the reviewer's insightful comment on physical assumptions. While NSG-VD leverages established approximations (incompressible flow and brightness constancy) for tractability, it is designed to **remain robust even when these assumptions are imperfect**.
>
> * **Incompressibility approximation**. Unlike methods relying strictly on the divergence term $\nabla_{\mathbf{x}} \cdot \mathbf{v}$, our NSG formulation $\partial_t\log p+\nabla_{\mathbf{x}}\cdot\mathbf{v}+\mathbf{v}\cdot\nabla_{\mathbf{x}}\log p=0$ aggregates multiple physical inconsistencies, including temporal change $\partial_t \log p$, spatial divergence $\nabla_{\mathbf{x}} \cdot \mathbf{v}$, and directional advection. Even if $\nabla_{\mathbf{x}} \cdot \mathbf{v} \ne 0$, the NSG’s ratio $\mathbf{g} = \frac{\nabla_{\mathbf{x}} \log p}{-\partial_t \log p + \lambda}$ still exposes distributional shifts.
> * **Brightness constancy**. This is a **mild assumption**—pixel intensities change smoothly with motion. While real videos may violate it under lighting/specular changes, these are often **spatiotemporally coherent** (e.g., a lamp sweeping across the frame). In contrast, generative videos often exhibit **disconnected inconsistencies** (e.g., misaligned shadows), causing larger NSG violations. Moreover, estimating $\partial_t\log p$ via **multi-frame finite differences** and combining it with score‐based $\nabla_{\mathbf{x}}\log p$ helps regularize such effects.
> * **Empirical validation**. We test NSG‑VD on challenging subsets with possible assumption violations: Sora (contains fluid/smoke), ModelScope（contains fast motion), MSR-VTT-Sports (contains rapid non-rigid motion), MSR-VTT-Vehicle (contains fast vehicle motion). From Table II, the average accuracy drops by 6.74% on these “stress” sets versus the full average, yet **remains competitive**, showing robustness under physically non-ideal cases.
>
> We will include these analyses in our revised manuscripts.
>
> Table II. Effect of "assumption-stress" datasets (Table 1 settings (%)).
> |Dataset|Subset|Accuracy|
> |-|-|-|
> |Sora|Fluid/Smoke|88.39|
> |ModelScope|Fast motion|81.67|
> |MSR-VTT-Sports|Sports|84.80|
> |MSR-VTT-Vehicle|Vehicle|84.00|
> |All Subsets (avg.)|—|91.46|
>
> >Q4. Pre-trained model: The NSG feature quality depends on the score from a pre-trained diffusion model. As only one is used, robustness to different architectures or domain gaps isn’t fully explored. How sensitive is NSG-VD to the model choice? Would a stronger one (e.g., Stable Diffusion) help, or is the current model sufficient?
>
> **A4.** We thank the reviewer for the valuable suggestion on pre-trained diffusion models. We discuss this with three key insights: 1) **framework agnosticism to score function sources**, 2) **diffusion model selection criteria**, and 3) **empirical consistency** across different diffusion models.
>
> * **Architecture‑agnostic framework**: NSG‑VD only requires the score function $\nabla_{\mathbf{x}}\log p(\mathbf{x},t)$. Once it is obtained, regardless of the diffusion architecture, we compute NSG features $\mathbf{g}(\mathbf{x},t)$ and train a **lightweight deep‑kernel** MMD detector. This decouples the detection from any specific diffusion designs.
> * **Diffusion model selection criteria**: We prioritize **unconditional, pixel-space diffusion** models that directly estimate $\nabla_{\mathbf{x}} \log p(\mathbf{x}, t)$ in the pixel space—critical for capturing video dynamics. In contrast, recent latent-space models (e.g., Stable Diffusion [r3], EDMv2 [r4]) operate in compressed latents and require additional decoding and Jacobian projections to recover pixel-space scores, adding complexity and potential approximation errors. We view this as compelling future work.
> * **Empirical validation across diffusion models**: In the main experiments, we use an ImageNet‑pretrained Guided Diffusion model [r1] and achieve strong detection performance (Tables 1–2). We also test with an Improved DDPM model [r2] on the same dataset, yielding **consistent performance** of 84.26% average Accuracy and 81.99% F1-score (Table I). This confirms that NSG‑VD’s efficacy stems from **general gradient estimation**, not model‑specific design and that our current diffusion model is **sufficient to capture key spatiotemporal dynamics** for detection.
>
> We will include these findings in the revised manuscript.
>
> [r5] Scaling Rectified Flow Transformers for High-Resolution Image Synthesis. ICML, 2024.
>
> [r6] Guiding a Diffusion Model with a Bad Version of Itself. NeurIPS, 2024.
>
> >Q5. Failure cases: The method shows impressive performance. Could you analyze when NSG-VD fails? This would offer valuable insights into its limitations and future improvement.
>
> **A5.** We agree that understanding failure modes is important. NSG‑VD commonly fails in scenarios mentioned in A2, e.g., videos with rapid lighting changes or high-speed motion, which may violate physical assumptions, e.g., brightness‑constancy or incompressibility approximations. We will include this analysis with representative examples in the revised manuscript.
>
> >Q6. How does the diversity of real videos in the reference set affect detection performance?
>
> **A6.** We sincerely thank the reviewer's valuable suggestion regarding the diversity of the reference set. We conduct additional ablation studies on **real‑domain mixed reference sets**, revealing a key strength of NSG-VD: **strong generalization to unseen generated video domains** when most real test samples are covered by the reference distribution.
>
> Specifically, we train on Kinetics-400 (real) and SEINE (generated) videos, and test on MSR-VTT (real) and 10 generated videos using **reference sets with varying ratios of MSR-VTT and Kinetics-400**. From Table III, **even a small proportion (3:7) yields satisfactory performance** (84.19% of Accuray, 81.12% of F1-Score) compared with baselines, which quickly saturates. This confirms that **NSG‑VD needs only modest in‐domain real coverage**, while the fake side can remain highly heterogeneous. These results will be included in our revision.
>
> Table III. Effect of reference set. Training: Kinetics-400 (real) and SEINE (generated); Testing: MSR-VTT (real) and 10 generated videos (%).
> |MSR-VTT: Kinetics-400 (Domain Coverage)|0:10 (Low)|3:7 (Medium)|5:5 (Balanced)|7:3 (High)|10:0 (Full)|DeMamba (Baseline)|TALL (Baseline)
> |-|-|-|-|-|-|-|-|
> |**Average Accuray**|77.82|84.19|85.57|87.06|86.05|84.21|80.20|
> |**Average F1-Score**|75.68|81.12|83.36|85.41|87.45|80.87|74.05|
>
> ---
> **We hope our response has clarified the confusion and addressed your concerns. Please feel free to reach out if you have any further questions or require additional details. Thank you.**

---

> > ### Comment · Reviewer_1Wda · 2025-08-04
> >
> > Thank you for the detailed response. I remain convinced of the paper's merits and will maintain my positive rating. I would appreciate the authors' perspective on one specific result: in Table I, the diffusion model operating on 256x256 underperforms (in accuracy) the 128x128 one. Could the authors elaborate on the potential reasons for this? Intuitively, one might expect the higher-resolution data to enhance the model's performance, so this finding is particularly interesting.

---

> > > ### Author Response · Authors · 2025-08-04
> > >
> > > We are deeply grateful for the reviewer's continued support and insightful engagement throughout the review process. Your recognition of the paper's core merits provides strong validation of our approach's scientific contribution, particularly **the novel physics-driven modeling paradigm** and **theoretically grounded detection framework**. We sincerely appreciate your constructive perspective in evaluating this work, which exemplifies the collaborative spirit of peer review.
> > >
> > > Regarding the 128×128 vs. 256×256 guided-diffusion models in Table I (see AUROC in Table IV), the 256×256 variant exhibits **a marginal 0.15% accuracy drop while achieving notable gains of 1.25% AUROC and 1.30% F1 Score**. We attribute this to two factors: 1) using a fixed decision threshold (τ=1) across all resolutions may not be optimal for the 256×256 model, slightly underestimating its accuracy (more experiments are in Reviewer oZqK-A3); and 2) the 256×256 setup was not exhaustively tuned, further hyperparameter optimization (e.g., adjusting learning rates) would likely recover or exceed the 128×128 accuracy while preserving its AUROC and F1 improvements.
> > >
> > > Table IV. Results of different-resolution diffusion model (Table 2 settings(%)).
> > > |Method|Accuracy|F1‑Score|AUROC|
> > > |-|-|-|-|
> > > |NSGVD (Guided Diffusion, 128x128)|**86.20**|86.25|94.16|
> > > |NSGVD (Guided Diffusion, 256x256)|86.05|**87.45**|**95.46**|
> > >
> > > Should you have any further questions, we are happy to provide additional clarifications. Thank you again for your time and consideration.
> > >
> > > Sincerely,
> > >
> > > The authors

---

### Official Review · Reviewer_74ip · 2025-07-02

**Clarity:** 4
**Significance:** 4
**Originality:** 3
**Rating:** 5
**Confidence:** 3

**Summary:**

The paper addresses the challenge of detecting AI-generated videos generated using diffusion models. Inspired from the principle of probability flow conservation in physics, the authors propose Normalized Spatiotemporal Gradient (NSG) statistics, which quantifies the ratio of spatial probability gradients to temporal density variations, enabling effective modeling of a video's spatiotemporal dynamics. Spatial gradients are estimated using a pretrained diffusion model, while temporal dynamics are captured through motion-aware techniques based on brightness constancy assumption avoiding the explicit flow computation.

The authors introduce NSG-based Video Detection (NSG-VD), which employs Maximum Mean Discrepancy (MMD) between features of real and test videos as a detection metric. Extensive evaluations are conducted on real videos from Kinetics-400 and generated videos from Pika and SEINE, using metrics such as Recall, Accuracy, F1-score, and AUROC, under both balanced and imbalanced data settings. The approach is compared against existing state-of-the-art methods.

A theoretical justification is also provided, showing that NSG feature distances between real and generated videos are upper-bounded, and that generated videos exhibit amplified discrepancies under distribution shifts.

**Questions:**

Please refer to the points listed in the weaknesses above.

**Ethical Concerns:**

["NO or VERY MINOR ethics concerns only"]

**Final Justification:**

Most of my concerns regarding change in the spatiotemporal dynamics of the videos induced through camera motion, analysis of different pre-trained diffusion models for computing NSG statistics and applicability of NSG-VD to fully unsupervised scenarios have been satisfactorily resolved. Therefore, I will raise my current score and support its acceptance.

**Limitations:**

Yes

**Quality:**

4

**Strengths And Weaknesses:**

Strengths

1. The paper identifies a relevant research problem that AI-generated videos often exhibit near-perfect visual realism while violating physical laws, making their reliable detection essential. The proposed approach is intuitive, drawing on the principles of probability flow conservation.
2. The paper is well-motivated and clearly articulated, backed by extensive evaluations, theoretical analysis, efficiency studies, and visualizations. The proposed NSG-VD approach demonstrates strong performance in detecting AI-generated videos, achieving high Recall, Accuracy, F1-score, and AUROC under both standard and data imbalanced evaluation settings.

Weaknesses

1.  Does the proposed NSG-VD also take into account the camera motion along with the object motion in the videos? Does change in the spatiotemporal dynamics of the videos induced through camera motion accounted for in Equation 6 and Equation 7?
2. An analysis of the types of pretrained diffusion models used to compute NSG
statistics and their impact on detection performance would provide valuable insight
into the generalizability of the proposed method.
3. Can NSG-VD be applied to fully unsupervised scenarios?

---

> ### Author Rebuttal · Authors · 2025-07-29
>
> We thank the reviewer for the encouraging comments and suggestions. Responses are below:
> >Q1. "...**intuitive**, drawing on the principles of probability flow conservation...", "the paper is **well-motivated and clearly articulated**, backed by **extensive evaluations, theoretical analysis, efficiency studies, and visualizations**", "...demonstrates **strong performance** in detecting AI-generated videos...".
>
> **A1.** We sincerely appreciate your thoughtful and encouraging comments on our work. Your acknowledgment of the strong performance of NSG-VD and your recognition of our paper as well-motivated and clearly articulated are deeply appreciated. We are also grateful for highlighting our work backed by extensive evaluations, theoretical analysis, efficiency studies, and visualizations. The comments that remark on our approach are intuitive, drawing on the principles of probability flow conservation is especially motivating, encouraging us to further advance our research in this domain.
>
> >Q2. Does the proposed NSG-VD also take into account the camera motion along with the object motion in the videos? Does change in the spatiotemporal dynamics of the videos induced through camera motion accounted for in Equation 6 and Equation 7?
>
> **A2.** We thank the reviewer's insightful comment. The proposed NSG-VD inherently accounts for **both object motion and camera motion through its physics-driven formulation**. Here's how our approach handles camera-induced dynamics:
> * **Unified physical modeling**: Our physics‐driven formulation $\frac{\partial p}{\partial t} + \nabla_{\mathbf{x}} \cdot \mathbf{J} = 0$ models the *net* spatiotemporal evolution of pixel probability density, which **naturally incorporates all motion components**--whether arising from objects, cameras, or their interaction. In this view, camera motion simply appears as a global coordinate transformation in the velocity field $\mathbf{v}(\mathbf{x}, t)$, which is absorbed into the probability flow $\mathbf{J} = p\mathbf{v}$ and preserved in continuity constraints.
> * **NSG's motion-agnostic property**: The NSG statistic $\mathbf{g} = \frac{\nabla_{\mathbf{x}} \log p}{-\partial_t \log p + \lambda}$ quantifies **relative changes** in probability density rather than absolute motion. Since camera-induced motion tends to produce a coherent, approximately uniform shift in the spatial gradient $\nabla_{\mathbf{x}}\log p$ across the frame, NSG remains **invariant** to such global flows. Meanwhile, subtle violations of conservation, e.g., those introduced by AI synthesis, still manifest as localized deviations in this ratio, allowing NSG to remain sensitive to the anomalies.
> * **Implementation-level robustness**: Both components of our estimator naturally incorporate camera motion. The pre-trained diffusion model’s score function $\nabla_\mathbf{x} \log p$ captures underlying texture geometry regardless of motion source, and our brightness-constancy–based temporal derivative $(\mathbf{x}_{t+\Delta t}-\mathbf{x}_t)/\Delta t$ reflects all observed pixel displacements from objects or the camera itself. Together, these ensure that NSG‑VD robustly handles combined object and camera motions without requiring any separate stabilization or motion‐segmentation preprocessing.
>
> We will clarify this in the revised manuscript, emphasizing that NSG-VD’s strength lies in its ability to model generic spatiotemporal constraints that hold regardless of motion origin.
>
> >Q3. An analysis of the types of pretrained diffusion models used to compute NSG statistics and their impact on detection performance would provide valuable insight into the generalizability of the proposed method.
>
> **A3.** We thank the reviewer for this valuable suggestion regarding the pre-trained diffusion models. We discuss this with three key insights: 1) **framework agnosticism to score function sources**, 2) **diffusion model selection criteria**, and 3) **empirical consistency** across different pretrained diffusion models.
>
> * **Architecture‑agnostic framework**: Our NSG‑VD pipeline requires only access to the score function $\nabla_{\mathbf{x}}\log p(\mathbf{x},t)$. Once this gradient is obtained, regardless of the underlying diffusion architecture, we compute the NSG features $\mathbf{g}(\mathbf{x},t)$ (Eqn. 4) and train a **lightweight deep‑kernel** MMD detector on these features. This design decouples the detection mechanism from any specific diffusion implementation.
> * **Diffusion model selection criteria**: We prioritize **unconditional, pixel-space diffusion** models because they directly estimate $\nabla_{\mathbf{x}} \log p(\mathbf{x}, t)$ in the pixel domain—critical for capturing spatiotemporal dynamics in videos. In contrast, recent latent-space models (e.g., Stable Diffusion [r3], EDMv2 [r4]) perform diffusion in compressed latents, requiring additional decoding and Jacobian projections to recover pixel-space scores, introducing additional complexity and potential approximation errors. We view this as compelling future work.
> * **Empirical validation across diffusion models**: In our main experiments, we use an ImageNet‑pretrained Guided Diffusion model [r1] and achieve strong detection performance (Tables 1–2). To further validate generalizability, we supplement additional experiments with an Improved DDPM-based pre-trained diffusion model [r2] on the same dataset, obtaining **consistent performance** of 84.26% average Accuracy and 81.99% F1-score. These results confirm that NSG‑VD’s efficacy stems from the **universal gradient‑estimation capability** of diffusion models, not model‑specific design and that our current pixel‑space diffusion model is **sufficient to capture the spatiotemporal dynamics needed** for robust detection.
>
> We will include these findings in the revised manuscript to highlight the framework’s generalizability and its current practical scope.
>
> Table I. Effect of different diffusion Models, with the same setting as Table 2 (%).
> |Method|Accuracy|F1‑Score|
> |-|-|-|
> |DeMamba (Baseline)|84.21|80.12|
> |NSGVD (Improved DDPM, 64x64)| 84.26|81.99|
> |NSGVD (Guided Diffusion, 256x256)|86.05|87.45|
>
> [r1] Diffusion Models Beat GANs on Image Synthesis.NeurIPS, 2021.
>
> [r2] Improved Denoising Diffusion Probabilistic Models. ICML, 2021.
>
> [r3] Scaling Rectified Flow Transformers for High-Resolution Image Synthesis. ICML, 2024.
>
> [r4] Guiding a Diffusion Model with a Bad Version of Itself. NeurIPS, 2024.
>
> >Q4. Can NSG-VD be applied to fully unsupervised scenarios?
>
> **A4.** We appreciate the reviewer’s insightful question regarding fully unsupervised scenarios. In principle, NSG‑VD can be applied in a fully unsupervised setup: 1）**using a Gaussian‑kernel MMD** on raw NSG features without any deep‑kernel training, and 2) **integrating unsupervised adaptation techniques** to learn discriminative mappings at test time.
>
> * **Unsupervised Gaussian-kernel MMD.** Our NSG‑VD framework can be run without labels by directly computing a Gaussian‑kernel MMD on the extracted NSG features. In this case, the detector still exploits spatiotemporal continuity violations but must operate on high‑dimensional features without a learned mapping, leading to reduced separability. From Table II,  Gaussian‑kernel MMD obtains 57.86% Accuracy and 61.67% F1‑Score, which is consistent with prior observations that untrained MMD can detect distribution shifts but underperforms learned kernels \[r5, r6].
> * **Unsupervised adaptation for enhanced separability.** To bridge this gap in truly label‑free settings, one promising direction is to combine NSG‑VD with unsupervised domain adaptation [r7] or test‑time adaptation (TTA) methods [r8], e.g., self‑supervised alignment or contrastive tuning of the deep kernel on unlabeled video streams could automatically discover the most discriminative projections of NSG features. We consider this an exciting avenue for future work and will clarify these supervision requirements in the revised manuscript.
>
> Table II. Result of Gaussian‑kernel MMD with NSG feature, where the test setting is the same as Table 2 (%).
> |Diffusion Model|Accuracy|F1‑Score|
> |-|-|-|
> |Gaussian‑kernel MMD|57.86|61.67|
> |NSG-VD (ours)| 86.05|87.45|
>
> [r5] Detecting machine-generated texts by multi-population aware optimization for maximum mean discrepancy. ICLR, 2024.
>
> [r6] Maximum Mean Discrepancy Test is Aware of Adversarial Attacks. ICML, 2021.
>
> [r7] Do We Really Need to Access the Source Data? Source Hypothesis Transfer for Unsupervised Domain Adaptation. ICML, 2020.
>
> [r8] Tent: Fully Test-time Adaptation by Entropy Minimization. ICLR, 2021.
>
> ---
>
> **We hope our response has clarified the confusion and addressed your concerns. We would greatly appreciate it if you could kindly reconsider your score. Thank you.**

---

> > ### Comment · Reviewer_74ip · 2025-08-05
> >
> > I thank the authors for addressing my questions in the rebuttal. Most of my concerns regarding change in the spatiotemporal dynamics of the videos induced through camera motion, analysis of different pretrained diffusion models for computing NSG statistics and applicability of NSG-VD to fully unsupervised scenarios have been satisfactorily resolved. Kindly integrate the justification provided in the rebuttal in the final version of the paper. Therefore, I will raise my current score and support its acceptance.

---

> > > ### Author Response · Authors · 2025-08-05
> > >
> > > Dear Reviewer 74ip,
> > >
> > > Thank you for your thoughtful review and your decision to raise your score following our rebuttal. We sincerely appreciate your positive feedback and support for the acceptance of our paper.
> > >
> > > We will carefully integrate the clarifications and justifications provided in our rebuttal into the final manuscript, as you suggested.
> > >
> > > Best regards,
> > >
> > > The Authors

---

### Official Review · Reviewer_oZqK · 2025-07-03

**Clarity:** 2
**Significance:** 3
**Originality:** 3
**Rating:** 4
**Confidence:** 3

**Summary:**

The paper proposes a metric termed Normalized Spatiotemporal Gradient (NSG) and defines it as the ratio of spatial probability gradients to temporal density changes. This metric is used to capture deviations from natural video dynamics. The motivation for this metric is the insight that "natural video dynamics preserve the product between the velocity field and the ratio of spatial probability gradients to temporal density changes."

An NSG estimator is designed using pretrained diffusion models. NSG-VD for video detection is then proposed using MMD between NSG features of test and real videos. It is claimed that generated videos have amplified discrepancies due to distribution shifts.

The performance of the proposed NSG-VD is shown to be competitive across numerous datasets using several quantitative metrics.

**Questions:**

- Are you assuming a Newtonian fluid in this work? Please clarify.

- Why and when is log introduced in the flow equation? If you are replacing p(x, t) with log (p(x, t)), does it impact the basic fluid-flow assumption? A basic review of fluid flow field constraints does not contain logarithmic terms. Please explain.

- Have you considered and compared your approach with the standard optical flow constrained equation? Please consider a comparison with optical flow given that the typical constraint imposed on it is the smoothness condition combined with the intensity motion.

- Why is Figure 1-b chosen to convey the intuition for this approach? Firstly, it is synthetic. Secondly, the reader cannot possibly compute the terms on the left hand side and check if it is indeed zero. Please provide better illustrations since the proposed methods fully rests on this idea.

- Can you please provide justification for the following assumption? "Assuming that the divergence term is subdominant
in smoothly varying distributions "

- Why is equation (7) referring to definitions made after it ((8) and (9))? Please rearrange the order.

**Ethical Concerns:**

["NO or VERY MINOR ethics concerns only"]

**Final Justification:**

Based on the responses and clarifications, the rating has been revised upwards. I still think that the overall idea, which is interesting, can be conveyed more clearly.

**Limitations:**

Yes

**Paper Formatting Concerns:**

There are no formatting concerns.

**Quality:**

3

**Strengths And Weaknesses:**

**Strengths:**
- The premise that natural (real) videos have a unique spatiotemporal distribution and that the flow of probability mass is governed by fluid-type physics is interesting.

- The results are encouraging.

**Weaknesses:**
- While the premise in indeed interesting, it is not sufficiently substantiated either empirically or analytically. It is not intuitive as to why natural videos are expected to obey equation (2). Without this clarity, readers will not be able to appreciate the contribution.

- Further, the logarithm of the distribution is used to go from (2) to (3). This is again not justified. Specifically, since the work is physics-inspired, this transition must be justified in terms of its impact on modeling fluid flow.

- The success of this method is dependent on the threshold $\tau$. As with any threshold-based approach, the choice of the threshold has a direct impact on performance. There is no discussion of this. Further, the choice of the threshold is given in the supplementary material.

- Please check for a typo in Table 4 ("gredients").

---

> ### Author Rebuttal · Authors · 2025-07-29
>
> We thank the reviewer for the detailed comments. Responses are below:
> >Q1. While the premise in indeed interesting, it is not sufficiently substantiated either empirically or analytically. It is not intuitive as to why natural videos are expected to obey equation (2).
>
> **A1.** We appreciate the reviewer's insightful comment regarding the foundation of the continuity Equation (2), i.e., $\frac{\partial p}{\partial t}+\nabla_{\mathbf{x}}\cdot\mathbf{J}=0$. This is not a video-specific assumption but a **universal mathematical formulation of probability mass conservation**, which holds for **any time-evolving probability density** $p(\mathbf{x}, t)$. It is **general and well-established** in statistical physics ([r1], Eqn.(1.16), Page 12) and Fokker-Planck equation（[r2], Eqn.(4.103), Page 84), where $\mathbf{J}=p \mathbf{v}$ ensures global conservation of the total probability over time. This principle is **universally valid without video-specific assumptions**. In fact, other reviewers recognize this strong physical grounding:
> * The proposed approach is **intuitive**, drawing on the principles of probability flow conservation [*Reviewer 74ip*].
> * The method is **well-grounded in physical principles**, like the probability flow conservation; The authors support their approach with a theoretical analysis, which, despite its simplifying assumptions, provides a **solid justification** [*Reviewer 1Wda*].
> * The paper is **well-motivated**; A statistic based on probability flow conservation principles [*Reviewer iZEA*].
>
> To clarify its relevance to videos, we highlight two perspectives:
> * **Analoging fluid mechanics to videos**. Just as the continuity equation describes local density change equals net inflow/outflow, we treat pixel probability density $p(\mathbf{x},t)$ as a "visual mass" and motion as a flow field $\mathbf{v}$. Here, $\mathbf{J}=p\cdot\mathbf{v}$ quantifies probability flux driven by inter-frame motion. This analogy grounds foundational video models like optical flow ([r3], Sec. 4, page 3).
> * **Intuitive interpretation for video dynamics.** Natural videos are inherently continuous—objects move coherently and pixels change smoothly over time. This implies that "visual probability mass" redistributes conservatively along motion trajectories rather than appearing or vanishing abruptly. Equation (2) mathematically encodes this intuition from spatial redistribution $\nabla_{\mathbf{x}}\cdot\mathbf{J}$ and temporal evolution $\frac{\partial p}{\partial t}$.
>
> We will include this discussion in our revised version.
>
> [r1] Fluid dynamics: an introduction. Springer, 2014.
>
> [r2] Fokker-planck equation. Springer Berlin Heidelberg, 1996.
>
> [r3] Determining optical flow. Artificial intelligence, 1981.
>
> >Q2. Logarithm use in (2)→(3) is unjustified. Why introduce it? Does it impact fluid-flow assumptions?
>
> **A2.** The logarithm is **not an artificial introduction but a direct mathematical consequence** of applying the chain rule to the continuity equation, without altering the underlying physics. To clarify, for $\frac{\partial p}{\partial t}+\nabla_{\mathbf{x}}\cdot\mathbf{J}=0$, we substitute $\mathbf{J}=p\mathbf{v}$ and then divide the entire equation by $p$ (which is strictly positive everywhere in its support), yielding: $$\frac{1}{p}\partial_t p+\frac{1}{p}\nabla\cdot(p\mathbf{v})=0.$$ Applying the vector calculus product rule $\nabla_{\mathbf{x}}\cdot(p\mathbf{v})=p(\nabla_{\mathbf{x}} \cdot \mathbf{v})+\mathbf{v}\cdot(\nabla_{\mathbf{x}}p)$ and the chain rule of calculus, $\frac{1}{p}\frac{\partial p}{\partial t}=\partial_t\log p$ and $\frac{\nabla_{\mathbf{x}} p}{p}=\nabla_{\mathbf{x}}\log p$, we obtain Eqn.(3)： $$\partial_t\log p+\nabla_{\mathbf{x}}\cdot\mathbf{v}+\mathbf{v}\cdot\nabla_{\mathbf{x}}\log p=0.$$ Moreover, this transformation does **not alter the underlying fluid-flow constraint**—it is a variable change making explicit how velocity couples to log-density's temporal and spatial gradients.
>
> >Q3. Threshold τ affects performance, but its choice is undiscussed and in supplements.
>
> **A3.** Thanks for pointing out the importance of threshold selection. Our NSG-VD maintains **remarkably stable performance across a wide range of τ values** without requiring fine-grained tuning. As shown in Tables I, II, NSG-VD consistently high detection performance as $τ\in[0.7, 1.2]$ for average Accuracy and F1-Score across diverse generators compared with DeMamba. These results indicate that NSG features create a **clear separation between real and fake distributions**. In practice, we set $τ=1.0$ as the default. We will add these results to the main text.
>
> Table I. Effect of τ on average metrics (Table 1 settings (%)) (vs. DeMamba: 84.21 Acc / 80.12 F1).
> |τ|0.4|0.5|0.6|0.7|0.8|0.9|1.0|1.1|1.2|1.3|
> |-|-|-|-|-|-|-|-|-|-|-|
> |Accuracy|78.16|84.95|88.63|90.62|91.03|91.60|91.46|90.48|89.44|87.55|
> |F1|81.80|86.59|89.37|90.82|90.88|91.21|90.87|89.43|87.82|85.17|
>
> Table II. Effect of τ on average metrics (Table 2 settings (%)) (vs. DeMamba: 84.54 Acc / 80.87 F1).
> |τ|0.4|0.5|0.6|0.7|0.8|0.9|1.0|1.1|1.2|1.3|
> |-|-|-|-|-|-|-|-|-|-|-|
> |Accuracy|50.45|60.81|69.04|75.11|79.05|83.31|86.05|88.97|89.36|78.08|
> |F1|66.87|71.82|76.29|79.97|82.52|85.47|87.45|89.60|89.12|70.64|
>
> >Q4. Are you assuming a Newtonian fluid?.
>
> **A4.** No, **our method does not assume Newtonian fluids** (which invloves  linear stress-strain relationship). We only leverage the continuity equation, a universal mass/probability conservation that holds for any continuous medium (**Newtonian/non-Newtonian** fluids, or abstract probability flows). This equation governs the fundamental relationship between density evolution ($\partial_t p$) and spatial flux ($\nabla\cdot\mathbf{J}$), independent of material-specific properties like viscosity.
>
> >Q5. Have you considered and compared your approach with the standard optical flow constrained equation?
>
> **A5.** We do not rely on the standard optical-flow constrained equation [r3] for two key reasons: **fundamental limitations in exposing AI-generated anomalies** and **computational/robustness concerns**. Moreover, our experiments show NSG-VD's superiority over optical-flow-based methods.
> * **Limitations of Optical Flow Modeling:**
>     - **Local intensity focus**: Optical flow focuses on local pixel intensity changes under a smoothness prior; as a low‐level motion estimator, it often misses the subtle distributional anomalies from modern generative models.
>     - **Cost and sensitivity**: Dense flow estimation is computationally costly and sensitive to noise and minor misalignments, exposing the limitation that the generated models exploit.
> * **Empirical superiority over optical-flow baselines:** We compare NSG‑VD with an SOTA optical flow method [r4] trained on the same Kinetics‑400+Pika splits. From Table III, NSG‑VD outperforms [r4] by **3.6%** in Accuracy and **8.81%** in F1‑Score. This substantial margin demonstrates the advantages of our physics‑driven NSG in capturing spatiotemporal distribution discrepancies over traditional optical-flow constraints.
>
> Table III. Comparisons with [r4] on average metrics (Table 2 settings (%)).
> |Method|Accuracy|F1‑Score|
> |-|-|-|
> |AIGVDet|82.45|78.64|
> |**NSG-VD (ours)**|**86.05**|**87.45**|
>
> [r4] AI-Generated Video Detection via Spatio-Temporal Anomaly Learning. PRCV, 2024.
>
> >Q6. Why is Figure 1-b chosen to convey the intuition? Provide better illustrations.
>
> **A6.** Figure 1-b  was intended purely as a conceptual schematic of probability-flow conservation (our theoretical foundation) **not as a direct validation**. To avoid confusion, we will  **connect the theoretical foundation with our practical detection mechanism**, e.g., placing the conservation equation alongside video frames to link theory to video dynamics and highlighting the NSG feature as the key discriminative metric derived from this principle. We hope this can bridge the abstract physics-inspired model to our detection mechanism, and welcome further suggestions to strengthen this intuition.
>
> >Q7. Can you please provide justification for the assumption of the divergence term?
>
> **A7.** We assume $\nabla_{\mathbf{x}}\cdot\mathbf{v}$ is subdominant in smoothly varying video distributions based on three considerations: 1) **its estimation is ill-posed** in high dimensions, 2) **many physical flows approximate incompressibility**, and 3) our NSG **remains robust** even if $\nabla_{\mathbf{x}}\cdot \mathbf{v}\neq0$.
> * **Ill‑posed estimation:** Directly estimating $\nabla_{\mathbf{x}}\cdot\mathbf{v}$ is **provably ill-posed** for high-dimensional video data. Solving $\partial_t\mathbf{x}=\mathbf{v}(\mathbf{x},t)$ is an underdetermined inverse problem with infinite solutions. Video noise (e.g., blur, compression) further amplifies errors, making explicit divergence estimation unstable and computationally infeasible [r1, r3].
> * **Theoretical precedence**: The $\nabla_{\mathbf{x}}\cdot\mathbf{v}\approx 0$ is well-established physical and theoretical grounding in fluid dynamics [r1] and quantum mechanics [r5], simplifying estimation while preserving physical interpretability.
> * **Framework robustness:** Unlike methods relying strictly on $\nabla_{\mathbf{x}}\cdot\mathbf{v}=0$, NSG-VD captures cumulative violations across all terms in Eqn. (3). Even if $\nabla_{\mathbf{x}} \cdot \mathbf{v}\neq0$, its contribution is subsumed by NSG’s integration of spatiotemporal inconsistencies, with experiments confirming resilience to this assumption.
>
> [r5] Quantum mechanics: foundations and applications. Springer Science & Business Media, 2013.
>
> >Q8. Why is equation (7) referring to definitions made after it ((8) and (9))? Please rearrange the order. Please check for a typo in Table 4 ("gredients").
>
> **A8.** We will move (8)-(9) before (7) for clarity and correct the typos.
>
> ---
> **We hope our response has clarified the confusion and addressed your concerns. We would greatly appreciate it if you could kindly reconsider your score. Thank you.**

---

> > ### Comment · Reviewer_oZqK · 2025-08-03
> > **Clarification questions**
> >
> > The responses and clarifications are much appreciated. These answer the questions to a large extent. However, there are still a few questions that need clarification.
> > 1. Have you empirically analyzed the estimated p(x, t) (or the spatial gradient/temporal derivate)?
> > 2. How close is p(x, t) to a Gaussian (if this was estimated)? The reason for these questions is the assumption of Gaussianity in the theoretical guarantee analysis.
> > 3. Specifically, is there a unique (and ideally a universal) signature for real videos?
> > 4. What is the impact of the choice of real video references? Specifically, the Kinetics dataset relates to human actions. However, a generative model like Sora is not limited to such data.
> > 5. A related question is on the choice of the real video set for classification. Can you provide guidance on this choice based on the input test video? Is the current analysis limited to the datasets chosen for training?

---

> ### Author Response · Authors · 2025-08-04
>
> We sincerely thank the reviewers for their constructive and insightful feedback, which has helped us clarify key aspects of our physics-driven detection paradigm and further strengthen the presentation of our work.
> >Q1. Have you empirically analyzed the estimated p(x, t) (or the spatial gradient/temporal derivate)?
>
> **A1.** Estimating $p(\mathbf{x}, t)$ of high-dimensional video data is computationally intractable. Instead, we rigorously analyze the **core components of our NSG statistic**, including the spatial gradient $\nabla_{\mathbf{x}} \log p(\mathbf{x}, t)$ (numerator), temporal derivative $\partial_t \log p(\mathbf{x}, t)$ (denominator) and their ratio (NSG). Below, we analyze empirical distributions and detection performance to validate their detection power.
> * **Statistical Analysis of NSG Components:** We compute frame-wise statistics for 100 real (Kinetics-400) and generated (SEINE) videos. From Table I, both the spatial score and the temporal derivative magnitude exhibit **clear shifts in mean and variance** between real and generated videos. The NSG ratio also shifts, confirming that synthetic videos violate the spatiotemporal priors.
>     - **Spatial Gradients**: Generated videos exhibit 8.1% higher spatial gradient magnitude ($\mathbb E_t \Vert\nabla_{\mathbf{x}} \log p\Vert_2$: 6,201.43 vs. 5,737.61), indicating amplified texture irregularities.
>     - **Temporal Derivatives**: Real videos show 54.4% stronger temporal dynamics ($\mathbb{E}_t \vert\partial_t \log p\vert$: 1,263.95 vs. 576.82), reflecting that generated videos optimized for perceptual smoothness exhibit muted temporal changes.
>     - **NSG Ratio**: The combined statistic shows 79.6% larger norm magnitude (68.28 vs. 38.01) in generated videos, highlighting NSG’s sensitivity to physical violations.
> * **Detection Performance of Individual Components:** We quantify each component’s contribution to detection performance. While spatial and temporal features show statistical differences (Table I), using only the spatial score yields moderate Accuracy (87.99%) and F1 (83.40%). Using only the temporal derivative further drops performance (71.09% Accuracy, 66.97% F1). In contrast, our full **NSG ratio combines complementary cues**, achieving substantial gains across all metrics (91.46% Accuracy, 90.87% F1).
>
> We will include these results in our revised version.
>
> Table I. Results (mean ± std) of statistics from Kinetics-400 and SEINE, respectively, where $\mathbf{g} = \frac{\nabla_{\mathbf{x}} \log p}{-\partial_t \log p + \lambda}$.
> |Statics|$\mathbb E_t \Vert\nabla_{\mathbf{x}} \log p(\mathbf{x},t)\Vert_2$|$\mathbb E_t \ mean_i \ [\nabla_{\mathbf{x}} \log p(\mathbf{x},t)]_i$|$\mathbb E_t \ std_i \ [\nabla_{\mathbf{x}} \log p(\mathbf{x},t)]_i$|$\mathbb E_t \lvert\partial_t \log p(\mathbf{x},t)\rvert$|$mean_t \ \partial_t \log p(\mathbf{x},t)$|$std_t \ [\partial_t \log p(\mathbf{x},t)]$|$\mathbb E_t \Vert\mathbf{g} (\mathbf{x},t)\Vert_2$|$\mathbb E_t \ mean_i \ [\mathbf{g}(\mathbf{x},t)]_i$|$\mathbb E_t \ std_i \ [\mathbf{g}(\mathbf{x},t)]_i$|
> |-|-|-|-|-|-|-|-|-|-|
> |Real|5,737.61±586.69|0.04±0.05|14.79±1.51|1,263.95±1,073.76|84.48±664.76|1,704.79±1,613.40|38.01±77.00|-0.0002±0.0016|0.10±0.20|
> |Generated|6,201.43±464.24|0.00±0.04|15.98±1.20|576.82±385.19|-69.68±368.86|683.85±502.12|68.28±103.51|0.0001±0.0005|0.18±0.27|
>
> Table II. Impact of spatial gradients and temporal derivatives on average metrics (%).
> |Method|Recall|Accuracy|F1|AUROC|
> |-|-|-|-|-|
> |Spatial Gredients|87.99|82.84|83.40|91.85|
> |Temporal Derivatives|60.35|71.09|66.97|78.95|
> |NSG-VD (Ours)|**88.02**|**91.46**|**90.87**|**96.14**|
>
> >Q2. How close is p(x, t) to a Gaussian (if this was estimated)? The reason for these questions is the assumption of Gaussianity in the theoretical guarantee analysis.
>
> **A2.** Estimating the true video density is highly challenging and often infeasible, especially due to complex spatialtemporal interactions in NSG. Nevertheless, we aim to gain **theoretical understanding** of NSG-VD under principled assumptions. Our primary goal is **not to claim that real videos are exactly Gaussian, but to extract analytical insights** into NSG’s behavior in a simplified setting.
> * **Closed-form and bounds under Gaussian assumption：** Under the Gaussian assumption, we derive closed-form expressions of the NSG numerator and denominator and a rigorous upper bound on the squared NSG distance (Theorem 1 in our paper) that highlights the distribution shift between real and generated videos.
> * **Common practice for theoretical insights:** The Gaussian approximation is a widely analytical tool for deriving theoretical insights, e.g., OOD detection [r1] and adversarial detection [r2], not an empirical claim about video distributions.
>
> [r1] Embedding trajectory for out-of-distribution detection in mathematical reasoning. NeurIPS, 2024.
>
> [r2] A Simple Unified Framework for Detecting Out-of-Distribution Samples and Adversarial Attacks. NeurIPS, 2018.

---

> > ### Author Response · Authors · 2025-08-04
> >
> > >Q3. Specifically, is there a unique (and ideally a universal) signature for real videos?
> >
> > **A3.** Our method is designed to uncover a **universal physical feature** for real videos by modeling spatiotemporal dynamics from **first principles**. Inspired by probability flow conservation principles, we formulate video evolution through a *probability flow continuity equation*.  This physics-driven approach yields the Normalized Spatiotemporal Gradient (NSG), a **statistic that intrinsically captures adherence to physical constraints** (e.g., motion coherence, texture continuity) without relying on superficial artifacts, any specific scene content or generator architecture, serving as a broadly applicable signature of natural video dynamics.
> >
> > * **Uniqueness via spatiotemporal gradient coupling:** NSG’s uniqueness stems from its joint modeling of spatial gradients and temporal derivatives.
> >     - **Spatial gradients** $\nabla_{\mathbf{x}}\log p$: Capture texture/structure irregularities.
> >     - **Temporal derivatives** $\partial_t \log p$: Encode motion plausibility.
> >     - **Ratio formulation** $\mathbf{g} = \frac{\nabla_{\mathbf{x}} \log p}{-\partial_t \log p + \lambda}$: Physically couples both terms, amplifying subtle inconsistencies *unique* to synthetic content.
> > * **Empirical validation of universality:** Tables 1, 2, 3 in our paper show the superior detection ability on cross-generator scenarios. For example, NSG-VD maintains 81%+ Accuracy on unseen 10 generated videos, demonstrating the universality of NSG features to detect novel generations.
> >
> > >Q4. What is the impact of the choice of real video references? Specifically, the Kinetics dataset relates to human actions. However, a generative model like Sora is not limited to such data.
> > >Q5. A related question is on the choice of the real video set for classification. Can you provide guidance on this choice based on the input test video? Is the current analysis limited to the datasets chosen for training?
> >
> > **A4 & A5.** Thanks to the reviewer for this valuable suggestion regarding the real reference set. We conduct additional ablation studies on **real‑domain mixed reference sets**, revealing a key strength of NSG-VD: **strong generalization to unseen generated video domains** and **robust performance even when reference sets may differ from the training data**. Below, we present both empirical results and practical recommendations.
> >
> > * **Ablation on mixed reference sets:** we train on Kinetics-400 (real) and SEINE (generated) videos, and test on MSR-VTT (real) and 10 generated videos using **reference sets with varying ratios of MSR-VTT and Kinetics-400**. From Table III, **even a small proportion (3:7) yields satisfactory performance** (84.19% of Accuray, 81.12% of F1-Score) compared with baselines, which quickly saturate. This demonstrates that NSG-VD is **not limited to the original training datasets**, but only requires **modest in-domain real coverage** to establish a reliable detector, while tolerating highly heterogeneous fake sources.
> > * **Practical guidance for the choice of reference sets:** 1）**Known test domain:** Include at least 30% real videos from the target domain in your reference set. 2）**Unknown or diverse domains:** Combine multiple real-video datasets to broadly cover expected scene types; NSG’s physics-driven prior remains effective under moderate domain shifts.
> >
> > We will include these discussions in our revised manuscript.
> >
> > Table II. Effect of reference set composition, trained on Kinetics-400 (real) and SEINE (generated) videos, and tested on MSR-VTT (real) and the other 10 generated videos (%).
> > | **MSR-VTT: Kinetics-400** **(Domain Coverage)**|0:10 (Low)|3:7 (Medium)|5:5 (Balanced)|7:3 (High)|10:0 (Full)|DeMamba (Baseline)|TALL (Baseline)
> > |-|-|-|-|-|-|-|-|
> > |**Average Accuray**|77.82|84.19|85.57|87.06|86.05|84.21|80.20|
> > |**Average F1-Score**|75.68|81.12|83.36|85.41|87.45|80.87|74.05|

---

> > > ### Author Response · Authors · 2025-08-05
> > >
> > > Dear Reviewer,
> > >
> > > Thank you sincerely for your insightful questions, constructive feedback, and for acknowledging that our responses were appreciated and largely addressed your concerns. Your inquiries have been invaluable in helping us clarify the core principles, strengthen the theoretical foundations, and highlight the practical significance of our work.
> > >
> > > As we are unsure whether all concerns have been fully resolved to your satisfaction given that we cannot see the final assessment, we would appreciate the opportunity to provide further clarification on our key innovations, which we believe make a significant contribution to the field:
> > >
> > > * **Physics-Driven Spatiotemporal Modeling Grounded in Fundamental Principles**：Our approach is rooted in probability mass conservation, a universal principle from statistical physics and fluid dynamics. By modeling video dynamics as "visual probability flow," we derive the Normalized Spatiotemporal Gradient (NSG) statistic to capture deviations from natural spatiotemporal coherence. Unlike heuristic or data-specific methods, this physics-driven foundation enables generalization across diverse content and generators, providing a robust basis for detecting AI-generated anomalies.
> > > * **The NSG Metric: A Unique Signature for Real vs. Generated Videos**: NSG physically couples spatial gradients (texture/structural irregularities) and temporal derivatives (motion plausibility) into a single ratio, amplifying subtle inconsistencies in synthetic content. Empirical results show generated videos have distinct shifts, e.g., stronger spatial gradients and muted temporal dynamics, validating its discriminative power.
> > > * **Robustness and Discriminative Power Across Scenarios**：The NSG metric intrinsically combines spatial texture irregularities and temporal motion implausibility into a physically coupled ratio, creating a clear separation between real and fake distributions. Crucially, NSG-VD maintains remarkably stable performance across a wide threshold range (τ ∈ [0.7, 1.2] in Tables I-II), which reduces reliance on precise calibration and ensures consistent efficacy across diverse generative sources.
> > > * **Generalizability Beyond Training Domains:** NSG-VD’s physics prior ensures strong cross-generator generalization (**81%+** Acc on unseen generators) and robustness to reference data composition: even partial coverage of real-domain data (e.g., 30% in-domain real videos) yields high detection performance (**>84% Acc**), highlighting its resilience to domain shifts. This robustness makes it applicable to diverse generated content without relying on dataset-specific biases.
> > >
> > > We believe these contributions advance the field of AI-generated video detection by providing a **principled, generalizable, and robust** solution grounded in physics. We hope that these innovations and the substantial empirical gains will merit a score improvement.
> > >
> > > Should you have any further questions, we are happy to provide additional clarifications.
> > >
> > > Sincerely,
> > >
> > > The Authors

---

> > > > ### Comment · Reviewer_oZqK · 2025-08-05
> > > > **Concluding remarks**
> > > >
> > > > The detailed responses are much appreciated. I hope some of these points can be reflected in the final version. I have raised my score by two levels.

---

> > > > > ### Author Response · Authors · 2025-08-06
> > > > >
> > > > > Dear Reviewer oZqK05,
> > > > >
> > > > > Thank you for your kind feedback and for raising your score by two levels. We're pleased that our detailed responses were helpful, and we will ensure that the discussed points are incorporated into the final version of the manuscript.
> > > > >
> > > > > Best regards,
> > > > >
> > > > > The Authors

---

### Decision · Program_Chairs · 2025-09-17

**Decision:**

Accept (spotlight)

**Comment:**

This paper proposes a physics-inspired approach to detecting AI-generated videos via the Normalized Spatiotemporal Gradient (NSG), derived from probability flow conservation, and introduces NSG-VD using pretrained diffusion models and MMD comparisons. The method is original, theoretically grounded, and shows strong empirical results across diverse datasets, including challenging imbalanced cases. Reviewers praised the motivation and novelty but raised concerns about clarity of the derivation, reliance on assumptions like brightness constancy, and limited ablations on diffusion models and reference sets. During rebuttal, the authors addressed most of the concerns. Overall, while some aspects of clarity and analysis could be improved, the paper is novel, well-motivated, and empirically strong. I recommend acceptance.